# Structural basis for RNA polymerase II ubiquitylation and inactivation in transcription-coupled repair

Goran Kokic[1,2], George Yakoub [3], Diana van den Heuvel[3], Annelotte P. Wondergem[3], Paula J. van der Meer [3], Yana van der Weegen[3], Aleksandar Chernev[4], Isaac Fianu [1], Thornton J. Fokkens[5], Sonja Lorenz [5], Henning Urlaub [4,6], Patrick Cramer [1]✉ & Martijn S. Luijsterburg [3]✉

During transcription-coupled DNA repair (TCR), RNA polymerase II (Pol II) transitions from a transcriptionally active state to an arrested state that allows for removal of DNA lesions. This transition requires site-specific ubiquitylation of Pol II by the CRL4$^{CSA}$ ubiquitin ligase, a process that is facilitated by ELOF1 in an unknown way. Using cryogenic electron microscopy, biochemical assays and cell biology approaches, we found that ELOF1 serves as an adaptor to stably position UVSSA and CRL4$^{CSA}$ on arrested Pol II, leading to ligase neddylation and activation of Pol II ubiquitylation. In the presence of ELOF1, a transcription factor IIS (TFIIS)-like element in UVSSA gets ordered and extends through the Pol II pore, thus preventing reactivation of Pol II by TFIIS. Our results provide the structural basis for Pol II ubiquitylation and inactivation in TCR.

Transcription is an essential cellular process in which RNA polymerase II (Pol II) synthesizes complementary copies of protein-coding and non-coding genes. Lesions on the template DNA strand cause stalling of elongating Pol II (ref. 1) and trigger a genome-wide transcriptional arrest[2,3]. Transcription-coupled DNA repair (TCR) is a subpathway of nucleotide excision repair (NER) that specifically removes transcription-blocking lesions from actively transcribed DNA strands[4]. Defects in TCR genes give rise to distinct clinical phenotypes, including Cockayne syndrome (CS) and UV-sensitive syndrome (UV$^S$S). Investigation of these clinical phenotypes revealed CSB, CSA and UVSSA as essential TCR factors[5–9].

Previous structural, functional and cell biology studies have shaped current mechanistic understanding of the stepwise assembly of TCR factors around lesion-arrested Pol II (refs. 10–14). Upon stalling at a lesion, Pol II is initially recognized by the DNA-dependent ATPase CSB[10,14,15]. The ATPase activity of CSB is enhanced by the recruitment

of CSA, which directly binds the CSB ATPase domain, as well as a conserved CSA-interacting motif (CIM) at the very C-terminus of CSB[10,14]. CSA acts as a substrate recognition subunit of the DDB1–CUL4A–RBX1 ubiquitin (Ub) ligase complex (CRL4$^{CSA}$) that ubiquitylates Pol II in response to DNA damage[11,16–18]. Activation of CRL4$^{CSA}$ requires conjugation with the Ub-like protein NEDD8 (ref. 19), and inhibition of CRL4 neddylation prevents Pol II ubiquitylation in vivo[11,17]. Pol II ubiquitylation promotes the integration of UVSSA into the TCR complex in which CSA interacts with the N-terminal VHS domain of UVSSA[10–14]. Previous work revealed that only CSB interacts with Pol II, whereas CSA and the VHS domain of UVSSA are held in close proximity but do not have direct contact with Pol II (ref. 14). Therefore, how the CRL4$^{CSA}$ Ub ligase specifically targets a single lysine residue (RPB1-K1268) on the surface of the multiprotein Pol II assembly remains to be elucidated. The recruitment of the transcription factor IIH (TFIIH) complex to DNA damage-stalled Pol II is mediated by an interaction with UVSSA[9–11,20–29]. Additionally,

[1]Department of Molecular Biology, Max Planck Institute for Multidisciplinary Sciences, Göttingen, Germany. [2]Division of Structural Biology and Protein Therapeutics, Odyssey Therapeutics GmbH, Frankfurt am Main, Germany. [3]Department of Human Genetics, Leiden University Medical Center, Leiden, The Netherlands. [4]Bioanalytical Mass Spectrometry, Max Planck Institute for Multidisciplinary Sciences, Göttingen, Germany. [5]Ubiquitin Signaling Specificity, Max Planck Institute for Multidisciplinary Sciences, Göttingen, Germany. [6]Bioanalytics Group, University Medical Center Göttingen, Institute of Clinical Chemistry, Göttingen, Germany. ✉e-mail: patrick.cramer@mpinat.mpg.de; s.m.luijsterburg@lumc.nl

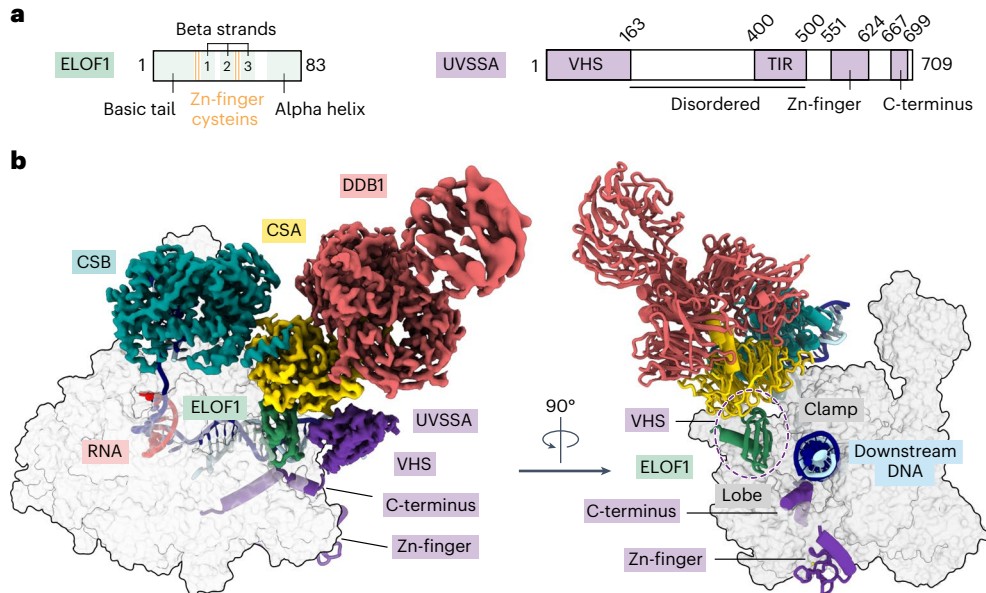

**Fig. 1 | Structure of the Pol II–TCR complex with ELOF1. a**, The scheme depicts the domain composition for ELOF1 and UVSSA featuring novel elements. **b**, Structure of the Pol II–TCR–ELOF1 complex. Cryo-EM density (left) and ribbon model (right) of TCR factors bound to Pol II.

TFIIH recruitment is impaired in cells lacking the RPB1-K1268 site[11]. Thus, both Pol II ubiquitylation and UVSSA interactions drive TFIIH binding to lesion-stalled Pol II.

The transition from transcription initiation to processive elongation involves evolutionary conserved Pol II-bound transcription elongation factors, such as DSIF (SPT4/SPT5)[30,31], transcription factor IIS (TFIIS)[32] and ELOF1 (refs. 12,13,32). Structural studies on the yeast Pol II elongation complex revealed that Spt4/Spt5 encircles the upstream DNA, constituting a DNA exit tunnel, whereas Elf1 is close to the downstream DNA and bridges the Pol II central cleft, completing a DNA entry tunnel[32]. It is becoming clear that elongation factors have functional connections with TCR. For example, stalling of Pol II at a DNA lesion triggers the replacement of DSIF by CSB[14]. Moreover, human ELOF1 not only stimulates transcription elongation in human cells but is also a Pol II-bound TCR factor that facilitates Pol II ubiquitylation and TFIIH recruitment in an unknown way[12,13]. The pausing of Pol II triggered by various obstacles, including small base damages, is typically overcome by TFIIS-dependent RNA cleavage and Pol II reactivation[3,33]. How the TCR machinery suppresses Pol II re-activation to lock Pol II in an arrested state that allows DNA repair of transcription-blocking lesions is unknown.

## Results

### ELOF1 anchors TCR factors to Pol II
To investigate the role of the elongation factor ELOF1 in TCR, we formed a complex among RNA Pol II, recombinant human ELOF1 and canonical TCR factors CSB, CRL4$^{CSA}$ and UVSSA and analyzed it by cryogenic electron microscopy (cryo-EM). We solved the structure of the complex at an overall resolution of 2.6 Å, which allowed the placement of side chains and refinement of the structure with good stereochemistry (Fig. 1, Extended Data Fig. 1 and Table 1). CSB binding to Pol II and upstream DNA remains largely unchanged in the presence of ELOF1 (refs. 14,15). We visualized all structured parts of ELOF1, including the beta-sheet comprising three anti-parallel beta-strands and associated zinc (Zn)-finger and a C-terminal helix (Fig. 1a,b). ELOF1 uses the helix to dock on the RPB2 lobe and projects the beta-sheet toward the RPB1 clamp, thereby completing the Pol II DNA entry tunnel, as observed for the yeast ELOF1 orthologue Elf1 (Fig. 1b)[32]. Such ELOF1 positioning explains how this factor stimulates transcription elongation[12,13].

Remarkably, the Zn-finger of ELOF1 inserts into the bottom face of the CSA beta-propeller, where it is grasped by residues protruding from the loops between the blades of CSA. Formation of this tight interface drives the repositioning of TCR factors closer to Pol II (Supplementary Video 1). The structure reveals that ELOF1 is the key factor for stable recruitment of the TCR machinery to DNA damage-arrested Pol II.

### ELOF1 binds CSA and positions the CRL4$^{CSA}$ Ub ligase
Identification of ELOF1 as the adaptor for the CRL4$^{CSA}$ Ub ligase allows us to analyze how the ligase integrates into a multi-protein complex containing its native substrate. In the absence of ELOF1, CSA is recruited to Pol II via two contact points with CSB: one with the ATPase lobe 1 of CSB and the other with the CIM located in the C-terminus of CSB[10,14] (Fig. 2a). Without other stabilizing interactions, the two contacts are insufficient to fix the position of CSA in three-dimensional (3D) space. This can be visualized by two-body refinement that shows the large-scale movement of CSA–DDB1 relative to Pol II (Fig. 2a and Supplementary Video 2). In the presence of ELOF1, CSA is fully engulfed in interactions that stabilize its position within the Pol II–TCR assembly: (1) two contact points with CSB are maintained while CSB is further stabilized by a new contact with the Pol II protrusion; (2) pincer helices on the Pol II clamp were a docking site for CSB in the absence of ELOF1, but, in the rearranged complex, the helices insert at the CSB–CSA interface; (3) ELOF1 connects CSA to the Pol II lobe; and (4) the VHS domain of UVSSA binds CSA and ELOF1 and is anchored to Pol II and downstream DNA via the C-terminus of UVSSA (Fig. 2b). The two-body refinement on the Pol II–TCR–ELOF1 complex shows a marked reduction of CSA mobility within the complex (Fig. 2b and Supplementary Video 2). Thus, ELOF1 restructures the interaction network within the Pol II–TCR complex and positions the CRL4$^{CSA}$ ligase in relation to its substrate, which may be important to prevent off-target ubiquitylation events after E3 ligase activation.

### ELOF1$^{Δdock}$ supports transcription but is deficient in TCR
Our structure reveals that ELOF1 interacts with CSA through a region close to its Zn-finger, including the conserved cysteine C29 and neighboring residues N30, H31 and E32. These residues interact with a pocket at the CSA surface lined with the aromatic residues Y100, F120 and Y145. In addition, the ELOF1–CSA interaction may be stabilized by two

**Table 1 | Model validation using Molprobity**

| Structure | Pol II –TCR–ELOF1 | C$^N$RL4$^{CSA}$–E2–Ub | C$^N$RL4$^{CSA}$–E2–Ub–UVSSA |
|---|---|---|---|
| PDB / EMDB | PDB: 8B3D; EMDB: EMD-15825 | PDB: 8B3I; EMDB: EMD-15829 | PDB: 8B3G; EMDB: EMD-15827 |
| Magnification | 81,000 | 81,000 | 120,000 |
| Voltage (kV) | 300 | 300 | 200 |
| Electron exposure (e/Å$^2$) | 42 | 42 | 40 |
| Defocus range (µm) | 0.5–2.5 | 0.5–2.5 | 0.5–2.5 |
| Pixel size (Å) | 1.05 | 1.05 | 1.23 |
| Symmetry imposed | C1 | C1 | C1 |
| Initial particle images (no.) | 2,807,926 | 1,288,908 | 1,201,107 |
| Final particle images (no.) | 189,593 | 82,975 | 67,365 |
| Map resolution (Å) | 2.6 | 3.5 | 4.4 |
| FSC threshold | 0.143 | 0.143 | 0.143 |
| Map resolution range (Å) | 2.3–5.2 | 3.5–5.9 | 3.7–5.9 |
| Refinement | | | |
| Initial model used (PDB ID) | 7OO3 | 7OO3, AlphaFold | 7OO3, AlphaFold |
| Model resolution (Å) | 2.6 | 3.7 | 8.4 |
| FSC threshold | 0.5 | 0.5 | 0.5 |
| Model composition | | | |
| Non-hydrogen atoms | 95,814 | 40,379 | 42,862 |
| Protein residues | 5,876 | 2,537 | 2,685 |
| Nucleotide | 102 | 0 | 0 |
| Ligands | 10 Zn, 2 Mg, ADP:BF3 | 3 Zn | 4 Zn |
| B factors (Å) | | | |
| Protein | 21.71 | 59.10 | 370.14 |
| Nucleotide | 51.62 | NA | NA |
| Ligand | 37.75 | 92.08 | 735.83 |
| Root-mean-square deviations | | | |
| Bond lengths (Å) | 0.01 | 0.005 | 0.005 |
| Bond angles (°) | 1.663 | 0.646 | 0.681 |
| Validation | | | |
| MolProbity score | 1.67 | 1.80 | 1.88 |
| Clashscore | 6.70 | 5.99 | 7.70 |
| Poor rotamers (%) | 0.08 | 0 | 0.13 |
| Ramachandran plot | | | |
| Favored (%) | 95.5 | 92.5 | 92.7 |
| Allowed (%) | 4.5 | 7.5 | 7.3 |
| Disallowed (%) | 0 | 0 | 0 |

glutamic acid residues (E55 and E79) in ELOF1, which interact with opposing arginine residues (R354 and R92) in CSA (Fig. 3a). To test the importance of these interactions, we stably expressed GFP-tagged versions of ELOF1$^{WT}$, ELOF1$^{N30A–H31A–E32A}$ (hereafter ELOF1$^{\Delta dock}$), ELOF1$^{E55A–E79A}$ or an ELOF1$^{5A}$ allele in ELOF1-deficient RPE1 cells (Extended Data Fig. 2a–c). ELOF1-knockout (KO) cells have approximately 50% reduced basal transcription levels[12,13]. Re-expression of all ELOF1 alleles rescued the transcription defect, showing that these ELOF1 mutants fully support transcription elongation (Fig. 3b,c).

To assay the function of ELOF1 mutants in TCR, we measured the ability of cells to recover RNA synthesis after UV irradiation, which is fully dependent on functional TCR[12]. Expression of ELOF1$^{WT}$ or ELOF1$^{E55A–E79A}$ fully rescued transcription restart after UV irradiation in ELOF1-deficient cells, suggesting that the two peripheral salt bridges do

not contribute to the ELOF1–CSA interaction. Conversely, expression of the ELOF1$^{\Delta dock}$ and ELOF1$^{5A}$ mutants failed to restore transcription recovery (Fig. 3d,e and Extended Data Fig. 3a,b). We next made use of the chemotherapeutic agent trabectedin, which is specifically toxic and causes DNA breakage only in TCR-proficient cells[34]. Functional assays with trabectedin confirmed that ELOF1$^{\Delta dock}$ does not support TCR (Extended Data Fig. 3c,d). Immunoprecipitation of endogenous Pol II to isolate intact TCR complexes[10] revealed that the UV-induced ubiquitylation of Pol II and the association of TFIIH was severely reduced in cells lacking ELOF1, as previously reported[12]. Re-expression of ELOF1$^{WT}$ fully restored both Pol II ubiquitylation and TFIIH binding, whereas ELOF1$^{\Delta dock}$ failed to rescue these cellular phenotypes (Fig. 3f). Although CSB binding was normal, we noticed a consistently weaker association of CSA with Pol II in the absence of ELOF1, which was dependent on

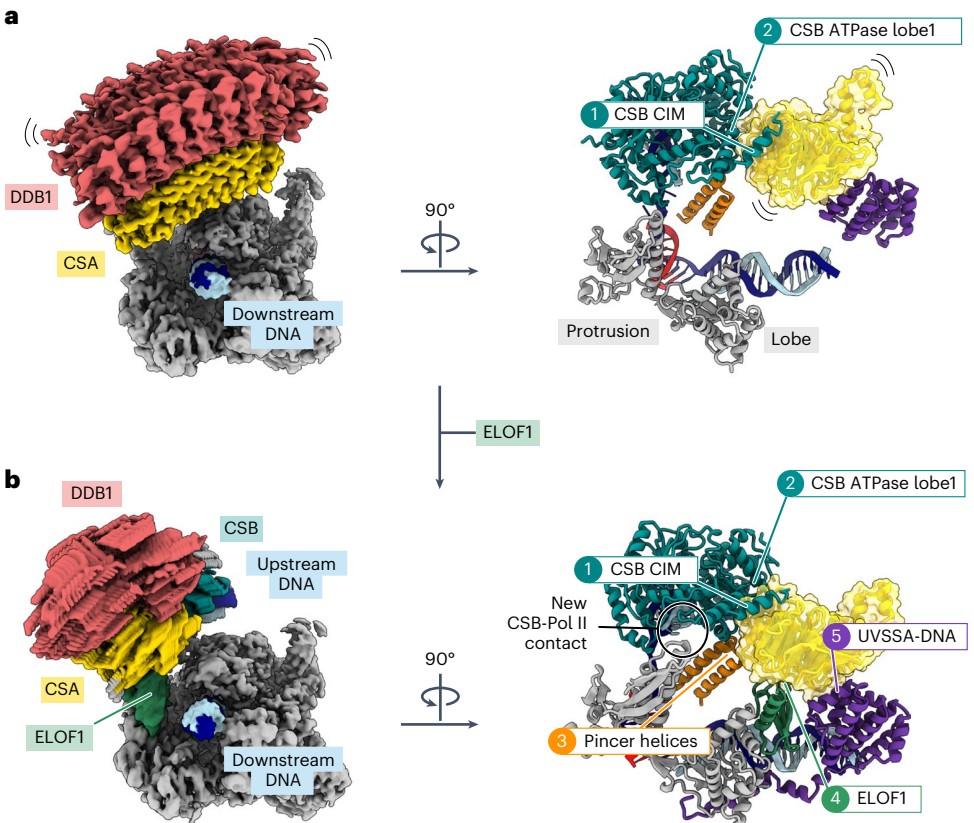

**Fig. 2 | ELOF1 anchors TCR factors to Pol II. a**, Raw maps resulting from the multi-body refinement of the Pol II–TCR complex (left) and ribbon model showing the contacts with CSA (right). **b**, Raw maps resulting from the multi-body refinement of the Pol II–TCR–ELOF1 complex (left) and ribbon model showing the contacts with CSA (right).

the N30A–H31A–E32A residues (Fig. 3f). This is consistent with a role for ELOF1 in facilitating the stable docking of CSA onto Pol II upon its recruitment by CSB[10]. Thus, although dispensable for its role as a Pol II elongation factor, our experiments reveal that the interaction between CSA and ELOF1 is essential for TCR.

**ELOF1 and UVSSA cooperatively drive Pol II ubiquitylation**

ELOF1 binds the VHS domain of UVSSA, thereby extending and stabilizing the CSA–UVSSA interface that was poorly defined in the absence of ELOF1 (ref. 14). The interface assembles around the CSA residue Y334 that inserts into a hydrophobic pocket formed by helices 6 and 8 of the VHS domain of UVSSA (Fig. 4a). Mutating the Y334 residue in CSA indeed causes a TCR-deficient phenotype in vivo (Fig. 4b,c and Extended Data Fig. 2d,e). Moreover, ELOF1-driven repositioning of the TCR complex allows additional UVSSA domains to stably bind Pol II and downstream DNA, as described in detail below. Repositioning of UVSSA also removes the VHS domain from the path of downstream DNA, as observed in the absence of ELOF1 (Supplementary Video 1)[14]. Overall, ELOF1 directly and cooperatively facilitates UVSSA integration into the Pol II–TCR complex[12,13].

To further investigate the interplay among CSA, ELOF1 and UVSSA, we assembled the Pol II–CSA–DDB1–CSB–ELOF1 complex in the absence of UVSSA and analyzed it by cryo-EM (Fig. 4d and Extended Data Fig. 4). Without UVSSA, we observed that around 50% of the particles contained CSA not bound to ELOF1. This is markedly different from the structure in the presence of UVSSA, where almost all particles that showed density for UVSSA exhibited stable docking of CSA to ELOF1, suggesting a critical role for UVSSA in facilitating CSA binding to ELOF1 (Fig. 4d). To investigate if the UVSSA-driven stabilization of the CSA–ELOF1 interface also affects Pol II ubiquitylation in vivo, we performed Pol II pull-down

experiments. Compared to WT cells, we observed that UVSSA-KO cells showed substantially reduced CSA recruitment and Pol II ubiquitylation after UV irradiation, which was rescued by re-expression of UVSSA (Fig. 4e). In vitro ubiquitylation assays with reconstituted Pol II–TCR–ELOF1 complex containing CRL4[CSA] validate that UVSSA promotes Pol II ubiquitylation in the presence of ELOF1 (Fig. 4f). Thus, ELOF1 and UVSSA jointly stabilize the docking and positioning of CRL4[CSA] on DNA damage-stalled Pol II, resulting in efficient Pol II ubiquitylation.

**CRL4[CSA] restructuring by neddylation guides Pol II ubiquitylation**

Integration of ELOF1 into the TCR assembly followed by UVSSA binding solidifies the position of CSA within the complex and provides a starting point to explore Pol II targeting by CRL4[CSA]. In a subpopulation of imaged Pol II–TCR–ELOF1 particles, we could visualize the CUL4A and RBX1 subunits of the CRL4[CSA] Ub ligase, because they were stabilized by the interaction between UVSSA and the winged-helix B (WHB) domain at the C-terminus of CUL4A (Fig. 5a and Extended Data Fig. 1). Modeling of an E2–Ub conjugate onto the RBX1 subunit[35] positions the activated Ub around 60 Å away from the ubiquitylation site on Pol II, which would unlikely result in productive ubiquitylation[36]. However, the activation and restructuring of CRL ligases often requires site-specific neddylation of the cullin subunit[19]. To study the impact of CRL4 neddylation, we immunoprecipitated Pol II-bound TCR complexes from cells treated with neddylation inhibitor. This treatment resulted in a complete loss of Pol II ubiquitylation after UV (Fig. 5b), as reported previously[11,17]. Although recruitment of CSB, CSA and UVSSA to lesion-stalled Pol II was unaffected, we observed the loss of mono-ubiquitylated UVSSA and strongly reduced association of both RBX1 and TFIIH with Pol II (Fig. 5b). To investigate the effect of neddylation on the conformation

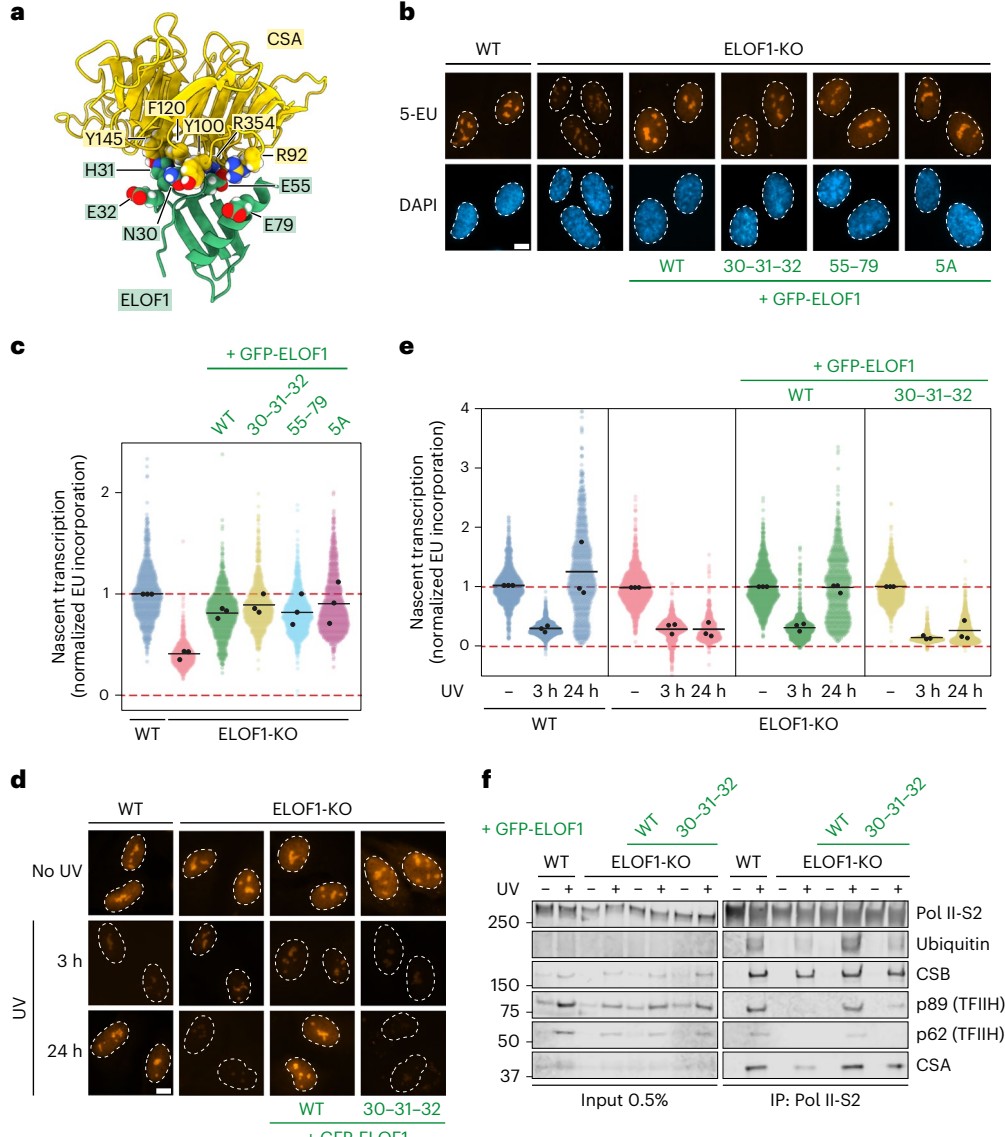

**Fig. 3 | CSA docks onto Pol II-bound ELOF1. a**, Zoom-in on the interfaces between ELOF1 and CSA. **b**, Nascent transcript levels in ELOF1[WT] and mutant cells measured by 5-EU incorporation. Representative microscopy images in the indicated RPE1-iCas9 cell lines. Scale bar, 10 μm. **c**, Quantification of the EU nuclear signal of images in **b**. The experiment was performed three times. Each colored circle represents one cell. Each black circle represents the mean of two technical replicates, with more than 80 cells collected per technical replicate. The black lines represent the mean of all three independent experiments. **d,e**, Measuring transcription recovery (RRS) by 5-EU labeling in the indicated RPE1-iCas9 cells after UV irradiation (3 h or 24 h; 12 J m[−2]). Representative images (**d**)

and quantification (**e**) of 5-EU levels normalized to mock treatment for each cell line. The experiment was performed three times. Each colored circle represents one cell. Each black circle represents the mean of two technical replicates, with more than 80 cells collected per technical replicate. The black lines represent the mean of all three independent experiments. Scale bar, 10 μm. **f**, Endogenous Pol II-S2 immunoprecipitation (IP) after UV irradiation (9 J m[−2], 1-h recovery) in RPE1-iCas9-WT or ELOF1-KO cells complemented with indicated GFP-tagged mutants of ELOF1. TCR complex assembly and Pol II ubiquitylation were analyzed with the indicated antibodies. The data shown represent at least three independent experiments.

of the CRL4[CSA] ligase and its activity toward Pol II, we quantitatively neddylated the ligase in vitro (C[N]RL4[CSA]) and analyzed the modified Pol II–TCR–ELOF1 complex by cryo-EM (Extended Data Fig. 5). Neddylation did not affect the overall structure of the complex, but it increased the flexibility of the RBX1 RING domain and the WHB domain, which could no longer be visualized (Fig. 5a). The modification of WHB likely disrupts its interaction with UVSSA and increases the conformational freedom of the WHB–RING–NEDD8 subcomplex, as observed for other neddylated CRLs not bound to a substrate[37].

To stabilize C[N]RL4[CSA] and better mimic the Ub transfer, we bound the ligase to an isopeptide-linked UBCH5B[C85K]–Ub conjugate (a stable proxy of the thioester-linked E2–Ub intermediate). Including the

conjugate caused issues during sample preparation with Pol II (Methods). Still, we observed dissociated C[N]RL4[CSA]–E2–Ub subcomplexes, which could be processed to an overall resolution of 3.5 Å (Fig. 5c and Extended Data Fig. 6). The complex is seemingly poised for auto-ubiquitylation of CSA, with Ub facing the side of CSA usually bound by UVSSA. More importantly, the resulting structure reveals how NEDD8 rearranges the CUL4A module that binds E2–Ub, as recently observed for neddylated CRL1 (ref. 37) (detailed in Extended Data Fig. 7 and Supplementary Video 3). To better understand how these changes affect the ligase function within the TCR assembly and Pol II targeting, we added UVSSA to C[N]RL4[CSA]–E2–Ub and analyzed the complex by cryo-EM (Extended Data Fig. 6). Although, at medium

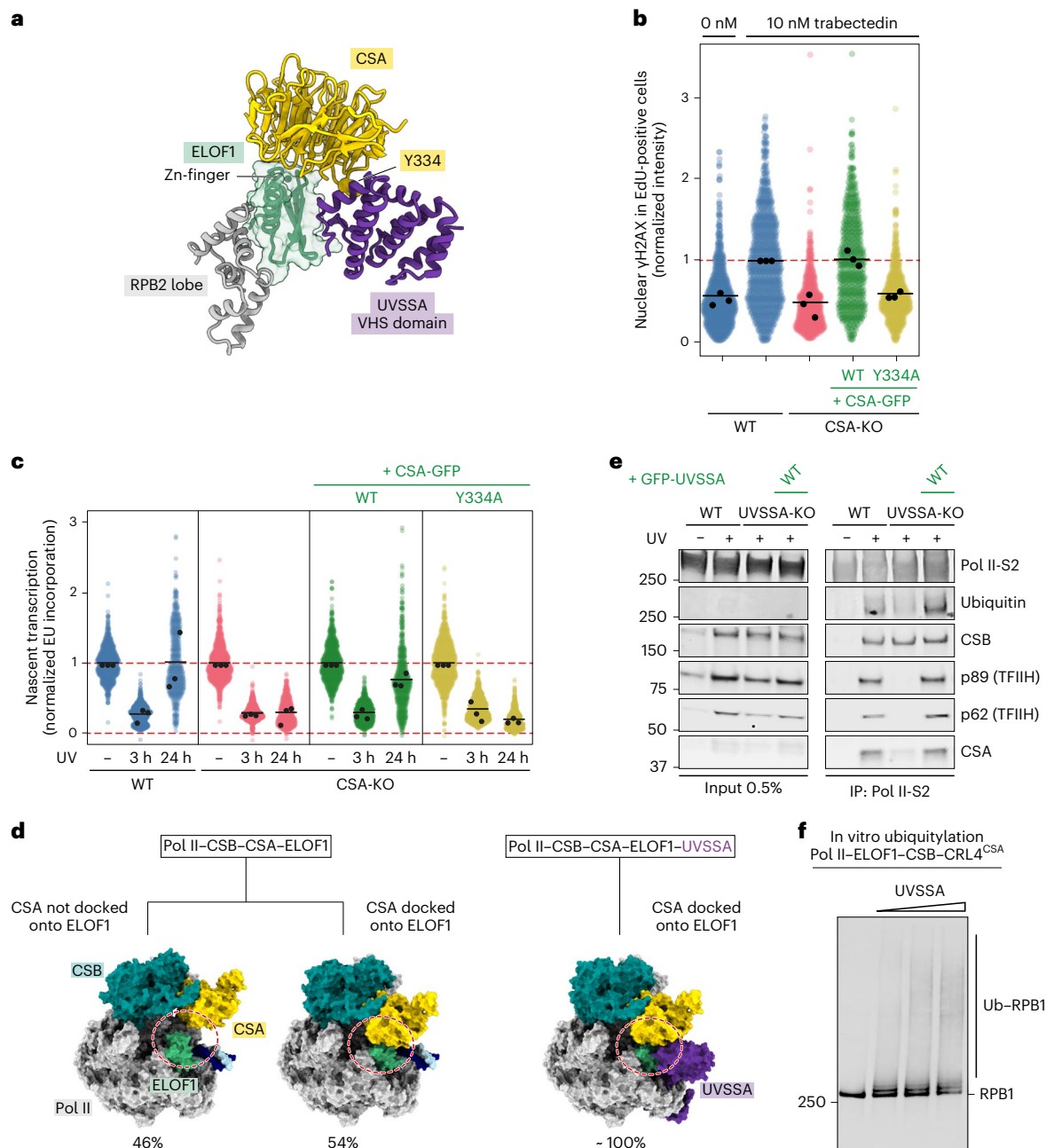

**Fig. 4 | UVSSA stabilizes the CSA–ELOF1 interface and stimulates Pol II ubiquitylation. a**, Zoom-in on the interfaces among UVSSA, ELOF1 and CSA. **b**, TCR-dependent γH2AX induction after trabectedin exposure in replicating cells labeled with 5-EdU in the indicated RPE1-iCas9 cells. The γH2AX levels were normalized to trabectedin-treated WT cells within each experiment. The experiment was performed three times. Each colored circle represents one cell. Each black circle represents the mean of two technical replicates, with more than 70 cells collected per technical replicate. The black lines represent the mean of all three independent experiments. **c**, Transcription recovery (RRS) by 5-EU labeling in the indicated RPE1-iCas9 cells after UV irradiation (24 h; 9 J m$^{-2}$). The 5-EU levels were normalized to mock treatment for each cell line. The experiment

was performed three times. Each black circle represents the mean of two technical replicates. The black lines represent the mean of all three independent experiments. **d**, Percentage of particles with CSA docked onto ELOF1 in the Pol II–CSB–CSA–ELOF1 dataset (left) and the Pol II–CSB–CSA–ELOF1-UVSSA dataset (right). **e**, Endogenous Pol II-S2 immunoprecipitation (IP) after UV irradiation (9 J m$^{-2}$, 1-h recovery) in RPE1-iCas9-WT or UVSSA-KO cells complemented with GFP-tagged UVSSA. TCR complex assembly and Pol II ubiquitylation were analyzed with the indicated antibodies. The data shown represent at least three independent experiments. **f**, In vitro ubiquitylation assay of Pol II in the presence of CSB, CRL4$^{CSA}$ and ELOF1 with increasing amounts of UVSSA. The experiment was repeated two times.

resolution, ranging from 4 Å to 8 Å, clear secondary structure features in densities for all polypeptides could be used to unambiguously dock the C$^N$RL4$^{CSA}$–E2–Ub and UVSSA structures and create a model that we superimposed onto Pol II–TCR–ELOF1 (Fig. 5a).

Comparing CRL4$^{CSA}$ and C$^N$RL4$^{CSA}$ reveals major conformational changes in the ligase induced by neddylation and binding to E2–Ub (Fig. 5

and Supplementary Video 3). In the absence of neddylation, CRL4$^{CSA}$ is stabilized by the interaction between CUL4A and DDB1 on one end and the interaction between WHB and UVSSA on the other end. Upon neddylation and loading with E2–Ub, the beta-propeller B of DDB1 turns 40° in a clockwise direction in relation to the rest of CSA–DDB1, which breaks the interface between CUL4A and DDB1 and allows the E2

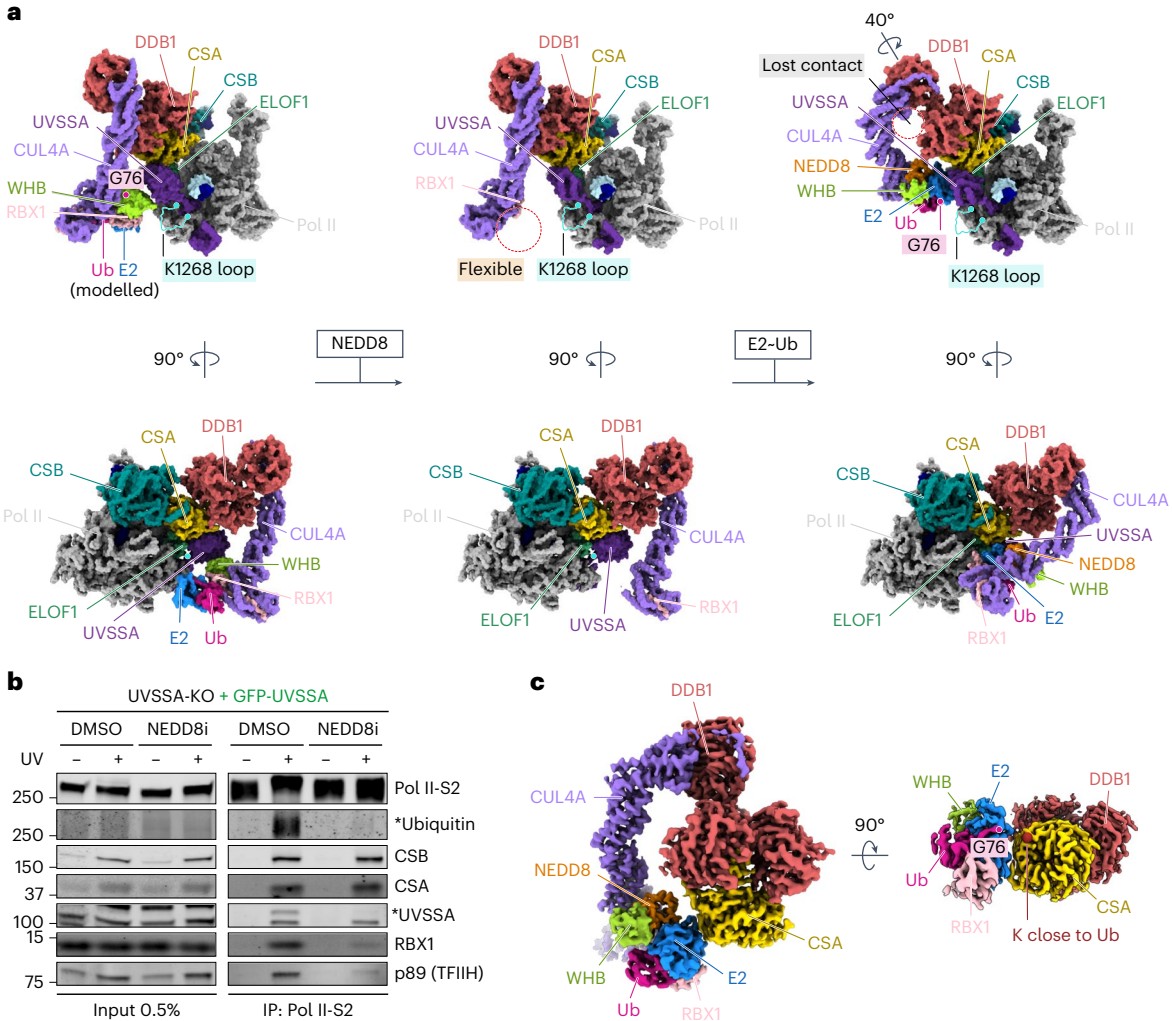

**Fig. 5 | Effect of CRL4^CSA neddylation on Pol II targeting. a**, Left, Pol II–TCR–ELOF1 structure containing non-neddylated CRL4^CSA with E2–Ub conjugate modeled based on a RNF4 RING E2–Ub structure (PDB: 4AP4)[35]; middle, Pol II–TCR–ELOF1 structure containing neddylated C^NRL4^CSA; and right, C^NRL4^CSA–E2–Ub–UVSSA structure superimposed onto Pol II–TCR–ELOF1 structure. The unstructured loop carrying the ubiquitylation residue is highlighted in blue. Position of G76 of Ub is marked for easier comparison. The structures below show a 90° counterclockwise rotation around the vertical

axis of the upper structures. **b**, Endogenous Pol II-S2 immunoprecipitation (IP) after UV irradiation (20 J m⁻², 1-h recovery) in U2OS UVSSA-KO cells complemented with GFP-UVSSA. TCR complex assembly and Pol II ubiquitylation were analyzed with the indicated antibodies. Pol II ubiquitylation was detected in a separate IP experiment in the presence of deubiquitylase inhibitor N-ethylmaleimide (NEM). The asterisks refer to the ubiquitylated form of the indicated proteins. The data shown represent at least three independent experiments. **c**, Cryo-EM density of the C^NRL4^CSA–E2–Ub complex poised for CSA auto-ubiquitylation.

to engage helix 7 of the VHS domain of UVSSA that also moves slightly toward the ligase. As a result, the position of the activated Ub shifts approximately 30 Å closer to the unstructured Pol II loop carrying K1268, which halves the distance compared to the non-neddylated ligase (Fig. 5a). The donor Ub is still positioned slightly away from Pol II likely to prevent off-target modification of surface-bound residues. However, K1268 resides in an unstructured loop and might be able to enter the ubiquitylation zone of C^NRL4^CSA, which would be an intriguing strategy to achieve specificity in this system. Overall, the positioning of CRL4^CSA on the Pol II surface by ELOF1 and UVSSA, in conjunction with ligase restructuring by neddylation, guides Pol II ubiquitylation after DNA damage-induced transcription arrest.

### ELOF1-stabilized UVSSA reveals new structural elements

The repositioning and compaction of TCR factors around ELOF1 reveal additional structural elements on Pol II. We observed a helical density that docks on the Pol II jaw and contacts downstream DNA, where it kinks and continues for more than 30 Å until it reaches the Pol II funnel

(Fig. 1b). Guided by crosslinking mass spectrometry data and bulky residues, we could assign this density to the very C-terminus of UVSSA (Extended Data Fig. 8a–c). Thus, the C-terminus folds back, contacts the N-terminal VHS domain, binds the downstream DNA and enters Pol II. Two basic residues of the C-terminus, K679 and R683, insert into the minor groove of the downstream DNA, thereby fully closing the DNA entry tunnel and locking the position of the downstream DNA (Fig. 6a,b). We observed another density bound to the Pol II jaw that could be assigned to a Zn-finger in UVSSA as predicted by AlphaFold[38] and confirmed by bulky side chain densities and crosslinking data (Fig. 1b and Extended Data Figs. 1 and 8b,d,e). The protein density extends from the Zn-finger along the Pol II surface toward the pore, where it disappears due to increased flexibility.

### The C-terminus of UVSSA is essential for Pol II ubiquitylation and TCR

To investigate the functional relevance of the C-terminus of UVSSA, we stably expressed GFP-tagged UVSSA lacking amino acids 667–699

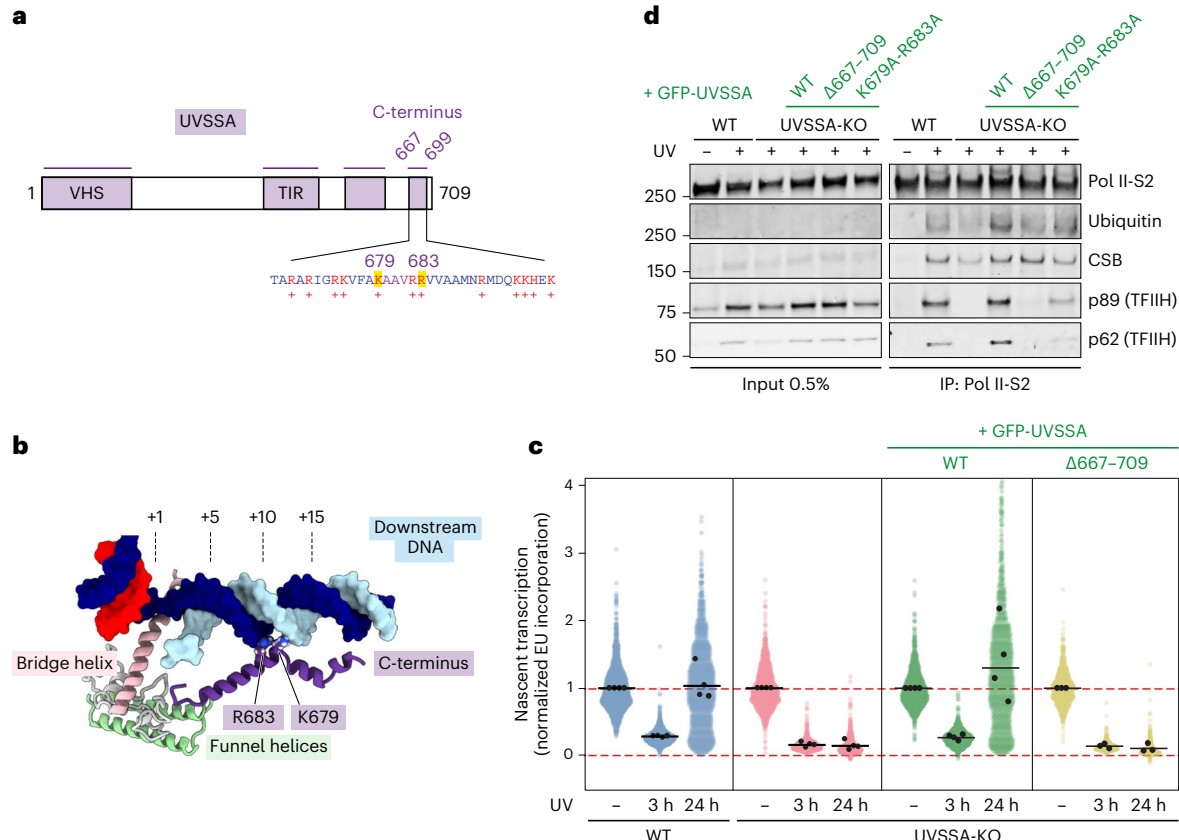

**Fig. 6 | The C-terminus of UVSSA stimulates Pol II ubiquitylation and TFIIH recruitment. a**, Domain composition for UVSSA featuring novel elements. **b**, Zoom-in on the UVSSA C-terminus binding to downstream DNA. **c**, Quantification of transcription recovery (RRS) by 5-EU labeling after UV irradiation (3 h or 24 h; 12 J m⁻²) in RPE1-iCas9 UVSSA-KO cells complemented with the indicated GFP-tagged WT or mutant UVSSA. 5-EU levels are normalized to mock treatment for each cell line. The experiment was performed three times. Each colored circle represents one cell. Each black circle represents the mean of two technical replicates, with more than 80 cells collected per technical replicate. The black lines represent the mean of all three independent experiments. **d**, Endogenous Pol II-S2 immunoprecipitation (IP) after UV irradiation (9 J m⁻², 1-h recovery) in RPE1-iCas9-WT or UVSSA-KO cells complemented with GFP-tagged WT or mutant UVSSA. TCR complex assembly and Pol II ubiquitylation were analyzed with the indicated antibodies. The data shown represent at least three independent experiments.

(hereafter, ΔC) or a mutant lacking the two positively charged residues, K679 and R683, that insert into the minor groove of the downstream DNA (Fig. 6a,b and Extended Data Fig. 9). Transcription recovery assays after UV irradiation revealed that expression of UVSSA^WT or UVSSA^K679A/R683A fully restored the transcription recovery defect in UVSSA-KO cells. In contrast, the UVSSA^ΔC mutant was indistinguishable from full UVSSA-KO cells (Fig. 6c and Extended Data Fig. 9c,d). Trabectedin assays confirmed that the UVSSA^ΔC mutant shows a strong TCR-deficient phenotype (Extended Data Fig. 9e,f). These experiments show that the C-terminus of UVSSA is essential for its function. Twelve out of the 33 residues in this region are positively charged, suggesting that other residues contribute to or can compensate for the loss of K679 and R683. Consistent with these phenotypes, immunoprecipitation of endogenous Pol II to isolate intact TCR complexes[10] revealed that UVSSA^K679A/R683A still supported TFIIH recruitment, albeit at lower levels than UVSSA^WT, but that the UVSSA^ΔC mutant did not support Pol II ubiquitylation or TFIIH recruitment to DNA damage-stalled Pol II (Fig. 6d and Extended Data Fig. 10a). In contrast to UVSSA^WT, the UVSSA^ΔC mutant showed strongly reduced interaction with CSB or Pol II after UV irradiation (Extended Data Fig. 10a). Thus, the C-terminus of UVSSA is essential for the stable incorporation of UVSSA in the Pol II-bound TCR complex.

**The newly identified UVSSA Zn-finger mimics TFIIS binding to Pol II**

ELOF1 changed the position of UVSSA on Pol II, which enabled stable binding and characterization of a previously unidentified Zn-finger

in UVSSA (Figs. 1b and 7a,b). The Zn-finger docks on the Pol II jaw: both knuckles of the finger insert their residues into a shallow groove formed by helix 1, consecutive beta-strand and helix 3 of the jaw domain, whereas the extension of the C-terminal knuckle contacts a small helix 2 on its way toward the Pol II pore. Interestingly, the positioning of these UVSSA elements almost perfectly overlaps with the binding site of transcription factor IIS (TFIIS) on Pol II (Fig. 7a)[39]. The domain II of TFIIS forms a three-helix bundle that inserts into the same groove on the Pol II jaw. Thus, UVSSA and TFIIS utilize very different structural motifs to bind the same site on Pol II. Moreover, the TFIIS domain II is followed by a linker and domain III that adopts a Zn-ribbon fold with extended beta-hairpin and inserts into the pore of Pol II (ref. 39). The UVSSA AlphaFold model reveals that the Zn-finger of UVSSA is also followed by a linker and an extended beta-hairpin (Fig. 7a). Indeed, our crosslinking data show that three lysine residues in the beta-hairpin of UVSSA extensively crosslink to Pol II residues lining the pore (Extended Data Fig. 8e). These observations suggest that UVSSA may also insert the beta-hairpin into the Pol II pore, thereby fully mimicking the binding of TFIIS to Pol II.

**UVSSA Zn-finger is essential for TFIIH recruitment**

To test the importance of the Zn-finger of UVSSA, we stably expressed GFP-tagged versions of UVSSA^WT, UVSSA^ΔZnF2 (C567A-C577A) or UVSSA^ΔZnF4 (C567A-C577A-C585A-H588A) in UVSSA-deficient RPE1 cells (Fig. 7b and Extended Data Fig. 9). Expression of UVSSA^WT fully rescued transcription restart after UV irradiation in UVSSA-deficient

cells, whereas UVSSA$^{\Delta ZnF2}$ failed to restore transcription recovery (Fig. 7c,d). Trabectedin assays confirmed that UVSSA$^{\Delta ZnF2}$ does not support TCR (Extended Data Fig. 9e, f). Both UVSSA$^{WT}$ and UVSSA$^{\Delta ZnF}$ mutants rescued CSA binding and Pol II ubiquitylation, whereas only the WT protein, but not the ΔZnF mutants, supported the association of TFIIH (Fig. 7e and Extended Data Fig. 10b). Pull-down of the UVSSA$^{\Delta ZnF}$ mutants showed a normal interaction with CSB or Pol II after UV irradiation, suggesting that the Zn-finger is dispensable for integration of UVSSA in the TCR complex (Extended Data Fig. 10c). These findings show an uncoupling between Pol II ubiquitylation and TFIIH recruitment at the level of UVSSA and reveal that different structural features of UVSSA mediate distinct cellular phenotypes. The N-terminal region forms a tight interface with ELOF1–CSA and stimulates efficient CSA docking and Pol II ubiquitylation, and the Zn-finger is critical for recruiting TFIIH.

## UVSSA inactivates arrested Pol II by blocking TFIIS
The encounter of DNA damage by Pol II in the template strand results in Pol II arrest and backtracking[1]. The function of TFIIS is to induce the cleavage of backtracked RNA by Pol II and reactivate transcription elongation, thereby acting as a potent anti-backtracking factor[40,41]. To investigate if UVSSA interferes with TFIIS anti-backtracking activity, we designed a biochemical assay that monitors TFIIS-induced cleavage of RNA in pre-assembled backtracked elongation complexes (Fig. 7f). When we add an increasing concentration of TFIIS to the pre-formed Pol II–TCR–ELOF1 complex lacking UVSSA, we observe efficient RNA cleavage (Fig. 7g). However, when we add UVSSA into the reaction, the RNA cleavage is strongly reduced, suggesting that UVSSA inhibits the function of TFIIS, most likely by blocking the access of TFIIS to Pol II. Thus, UVSSA prevents reactivation of backtracked Pol II by TFIIS, which may be important during the removal of Pol II from the site of DNA damage, especially as extensive backtracking of Pol II is likely involved[42].

## Discussion
Our current work reveals that the docking of CSA onto Pol II–ELOF1 restructures the TCR complex and positions the CRL4$^{CSA}$ ligase, providing a structural basis of how ELOF1 drives Pol II ubiquitylation. Together, our work reveals an elegant mechanism to restrict the mobility of an E3 ligase within a higher-order assembly to regulate specific ligase targeting and activation, thereby preventing off-target ubiquitylation.

### UVSSA and ELOF1 jointly stabilize CSA binding for Pol II ubiquitylation
The CSB-dependent recruitment of CSA is followed by the integration of UVSSA in the TCR complex, which is mediated by the interaction between CSA and the N-terminal VHS domain in UVSSA[10,14,21]. Our current work reveals that ELOF1 also facilitates UVSSA recruitment to the growing TCR assembly by interacting with UVSSA, tightening the UVSSA–CSA interface and repositioning UVSSA in a way that allows additional UVSSA elements to grip on the Pol II surface and downstream DNA. The C-terminus of UVSSA also contributes to this stabilization by folding back to contact the N-terminal VHS domain and binding downstream DNA. This ELOF1-driven repositioning delivers UVSSA close to the Pol II loop that carries the ubiquitylated residue K1268

and rationalizes why UVSSA recruitment to arrested Pol II is impaired in ELOF1-deficient cells[12,13].

### Functions and outcomes of Pol II ubiquitylation
The UV-induced Pol II ubiquitylation involves both proteolytic K48 and non-proteolytic K63-Ub linkages on the same site, which are both dependent on CSA[11]. Although examples of K63-Ub conjugation by CRL4 ligases have been reported[43,44], this effect could also be indirect. Here we show that loss of CSA, ELOF1, and UVSSA leads to a strong reduction in Pol II ubiquitylation using antibodies that recognize all types of conjugated Ub. Knockout of these TCR genes or loss of the K1268 ubiquitylation site leads to strongly reduced TFIIH binding to lesion-stalled Pol II (refs. 11,12), which is phenocopied by treatment with neddylation inhibitor (Fig. 5b). We, therefore, envision that a key function of Pol II ubiquitylation is to stimulate the assembly of the TCR complex, presumably through non-proteolytic Ub chains. Because CSB and potentially UVSSA contain Ub-binding domains, Pol II ubiquitylation could increase the affinity of TCR factors for Pol II (refs. 7,45). In addition, Pol II is degraded in response to UV irradiation at later timepoints. Pol II degradation is largely absent in cells deficient in CSA and CSB[7,46,47], whereas UVSSA-deficient cells show normal degradation[7] (confirmed in Extended Data Fig. 10d,e). These findings suggest that Pol II is decorated with a variety of different Ub linkages during TCR with different functional outcomes. It will be essential to dissect and manipulate the different Ub linkages on Pol II to understand their dynamics and functional roles[48].

### UVSSA inactivates lesion-arrested Pol II
The arrest of Pol II at various obstacles, including small base damages, is often resolved by TFIIS-dependent RNA cleavage triggering Pol II reactivation[3,33]. However, TFIIS is unable to facilitate bypass of bulky lesions that are processed by TCR[33]. Here we show that the repositioning and compaction of TCR factors around ELOF1 facilitates the binding of a Zn-finger in UVSSA to the Pol II jaw in a manner that overlaps with the binding site of TFIIS on Pol II (ref. 39). Crosslinking data further suggest that UVSSA inserts a beta-hairpin into the Pol II pore, similar to the beta-hairpin that TFIIS uses to induce cleavage of backtracked RNA in Pol II and reactivate transcription[40,41]. Consistent with a direct competition model, UVSSA inhibits the function of TFIIS in vitro in a concentration-dependent manner. Thus, UVSSA acts as a TFIIS mimic that prevents reactivation of backtracked Pol II at DNA lesions by blocking the access of TFIIS to Pol II. This is reminiscent of how pausing factor NELF inactivates Pol II (ref. 49). Thus, the two separate factors likely evolved to prevent Pol II activity in different biological contexts while using similar strategies. Such Pol II inactivation may be required for non-interrupted and possibly TFIIH-driven Pol II backtracking or Pol II dissociation to fully expose the DNA lesion to the downstream repair machinery.

### Pol II ubiquitylation and TFIIH recruitment: converging functions of UVSSA
The recruitment of the TFIIH complex to lesion-stalled Pol II depends on at least two distinct signals: (1) the ubiquitylation of Pol II at K1268 (ref. 11) and (2) protein–protein interactions with UVSSA[10]. RPB1-K1268R

**Fig. 7 | The Zn-finger of UVSSA inactivates Pol II and is essential for TCR.** **a**, Comparison between binding of TFIIS (PDB: 1PQV)[39] and UVSSA to Pol II. The UVSSA hairpin was not visible in our structure and was modeled based on chemical crosslinking and structure prediction (Methods). **b**, Domain composition for UVSSA featuring novel elements. **c,d**, Measuring transcription recovery (RRS) by 5-ethynyl-uridine (5-EU) labeling after UV irradiation (3 h or 24 h; 12 J m$^{-2}$) in RPE1-iCas9 UVSSA-KO cells complemented with the indicated GFP-tagged WT or Zn-finger mutant UVSSA. Representative images (**c**) and quantification (**d**) of 5-EU levels normalized to mock treatment for each cell line. The experiment was performed three times. Each colored circle represents one cell. Each black circle represents the mean of two technical replicates, with more

than 80 cells collected per technical replicate. The black lines represent the mean of all three independent experiments. Scale bar, 10 μm. **e**, Endogenous Pol II-S2 immunoprecipitation (IP) after UV irradiation (9 J m$^{-2}$, 1-h recovery) to detect TCR complex assembly in RPE1-iCas9-WT or UVSSA-KO cells complemented with GFP-tagged WT or Zn-finger mutant UVSSA. The data shown represent at least three independent experiments. **f**, Schematic representation of the assay used to monitor the activity of TFIIS. **g**, UVSSA prevents TFIIS-mediated activation of backtracked elongation complexes and counteracts the cleavage of nascent RNA. The average of three experiments is displayed, with the error bars representing the standard deviation. **h**, Model for transcription-coupled repair based on intermediates that we structurally define here. C-ter, C-terminus; ZnF, zinc-finger.

knock-in cells that are deficient in Pol II ubiquitylation show strongly reduced TFIIH recruitment[11]. These cells show normal recruitment of CSB, CRL4[CSA] and UVSSA to stalled Pol II (ref. [11]). Likewise, neddylation inhibitor MLN4924 also strongly reduces both Pol II ubiquitylation[11,17] and TFIIH recruitment to DNA damage-stalled Pol II, without affecting

recruitment of CSB, CRL4[CSA] and UVSSA to lesion-stalled Pol II. Elucidating how Pol II ubiquitylation contributes to TFIIH recruitment is an important future goal. Second, the recruitment of TFIIH to lesion-stalled Pol II is fully dependent on UVSSA[10,11,20]. The TFIIH-interacting region (TIR), located between amino acids 400 and 500, is essential for the interaction

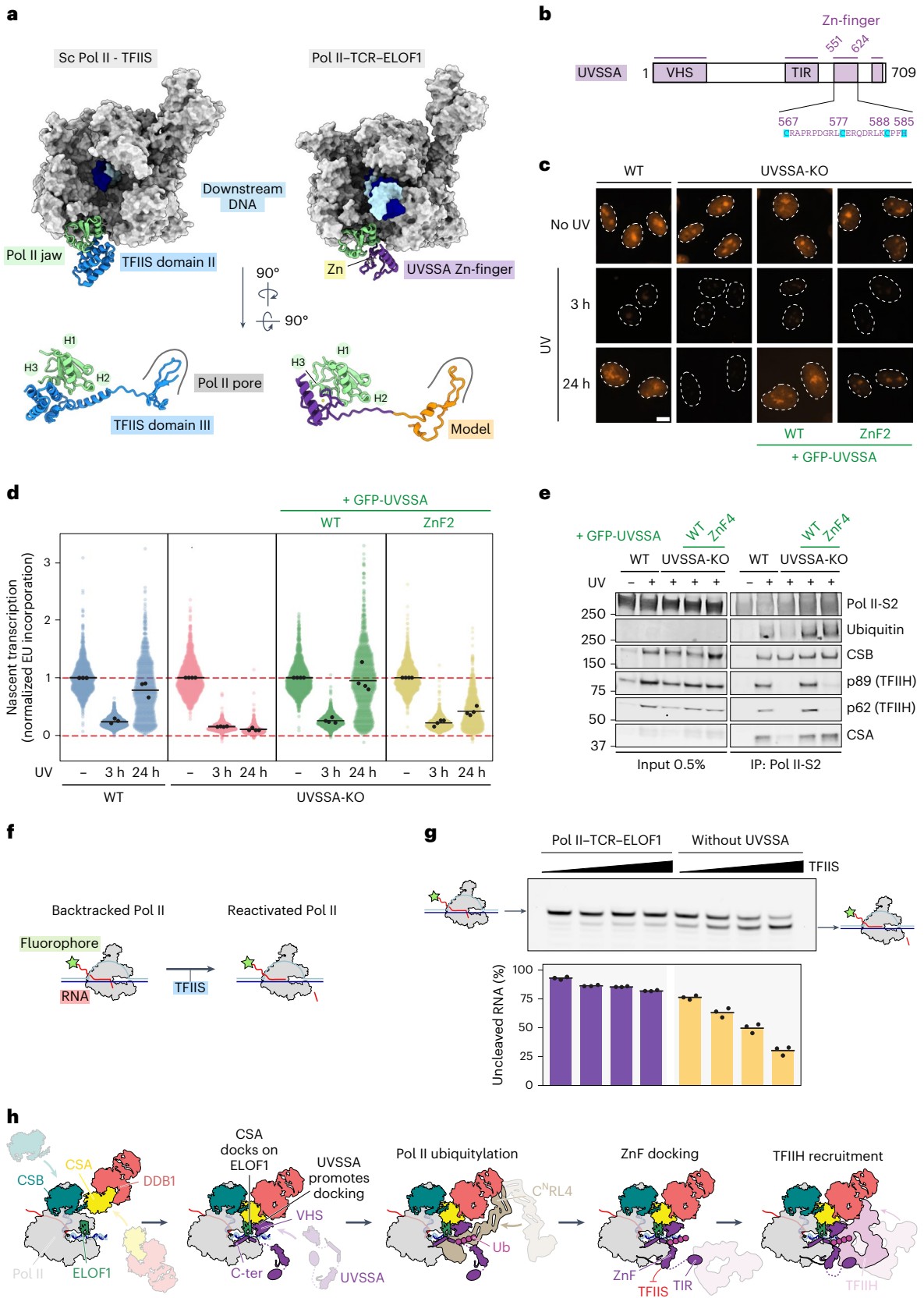

of UVSSA with TFIIH and its integration in the Pol II–ELOF1-bound TCR complex[10,12,21]. The Zn-finger in UVSSA (567–588) that binds the Pol II jaw is adjacent to the TIR motif (400–500). Mutating the Zn-finger in UVSSA did not affect Pol II ubiquitylation but specifically impaired TFIIH recruitment. This is reminiscent of a UVSSA mutant lacking its ubiquitylation site (K414) located in the TIR, which showed normal Pol II ubiquitylation but strongly reduced TFIIH interaction with lesion-stalled Pol II (ref. 11). An intriguing possibility is that the ELOF1-dependent repositioning of UVSSA enables its folding on the surface of Pol II, leading to (1) Zn-finger binding to the Pol II jaw, (2) optimal placement of the TIR for efficient UVSSA ubiquitylation by CRL4$^{CSA}$ and (3) TFIIH recruitment.

## A molecular model for TCR

Together with published work, our findings converge in the following molecular model (Fig. 7h). Pol II transcribes DNA with the help of several elongation factors that aid transcription and mask the binding site for the TCR machinery[14]. These elongation factors include DSIF, which binds the upstream DNA and TFIIS that is able to reactivate backtracked elongation complexes. Upon Pol II stalling on a lesion site, CSB competes out the elongation factor DSIF for Pol II binding and recruits the CRL4$^{CSA}$ ligase. CSA can sample a large conformational space on the Pol II surface, including unstable docking onto ELOF1. Once docked, the CSA–ELOF1 complex provides an ideal binding site for the VHS domain of UVSSA. Repositioning of the TCR complex triggered by the docking of CSA onto ELOF1 allows other UVSSA elements to wrap around the Pol II elongation complex, spanning the Pol II surface from the active site to the pore. Such UVSSA positioning occludes TFIIS from the elongation complex and represses transcription reactivation. The stabilized CSA–ELOF1–UVSSA ternary complex optimally positions the CRL4$^{CSA}$ ligase for Ub transfer onto Pol II upon ligase activation and restructuring by neddylation. Collectively, our findings reveal the molecular basis for how ELOF1 guides TCR from transcription toward DNA repair by restructuring the TCR complex to promote UVSSA integration, Pol II ubiquitylation, Pol II inactivation and TFIIH recruitment.

## Online content

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

## Methods

### Cloning and protein expression

Core TCR factors—CSB, UVSSA, CSA–DDB1 and CRL4[CSA]—were cloned, expressed and purified using Sf9, Sf21 and Hi5 insect cells (Thermo Fisher Scientific, 12659017; Expression Systems, 94-003F; and Expression Systems, 94-002F, respectively) as described[14]. The DNA coding sequence for human ELOF1 was gene optimized for expression in *Escherichia coli* and purchased from Integrated DNA Technologies (IDT). ELOF1 sequence was cloned into the 1-B vector (Addgene, plasmid 29653) by ligation-independent cloning (LIC)[50]. The DNA coding sequence for human APPBP1 and UBA3 were gene optimized for *E. coli* expression and ordered from IDT. UBA3 was cloned into the 14-A vector (Addgene, plasmid 48307) and APPBP1 into the 14-B vector (Addgene, plasmid 48308), followed by cloning both sequences on a single plasmid by LIC[50]. A modified, pSMT3 (LifeSensors)-based vector encoding UBCH5B with an N-terminal His6-SMT3-tag[51] was used as a template to generate the UBCH5B[C85K] mutant by ligation-free, site-directed mutagenesis (see Supplementary Table 1 for primers).

Human ELOF1 was expressed in *E. coli* BL21 DE3 cells in LB media supplemented with 1% glucose (w/v) for 4 h at 37 °C after induction with 0.5 mM IPTG. Cells were harvested by centrifugation for 30 min and resuspended in buffer containing 20 mM HEPES (pH 7.5), 300 mM NaCl, 1 mM dithiothreitol (DTT), 10% glycerol (v/v), 30 mM imidazole, 0.284 µg ml⁻¹ leupeptin, 1.37 µg ml⁻¹ pepstatin A, 0.17 mg ml⁻¹ PMSF and 0.33 mg ml⁻¹ benzamidine. The resuspended cells were stored at −80 °C until purification. APPBP1–UBA3 was expressed in *E. coli* BL21 DE3 cells for 12 h at 18 °C after induction with 1 mM IPTG. Cells were harvested by centrifugation for 30 min and resuspended in buffer containing 20 mM HEPES (pH 7.5), 400 mM NaCl, 1 mM DTT, 10% glycerol (v/v), 30 mM imidazole, 0.284 µg ml⁻¹ leupeptin, 1.37 µg ml⁻¹ pepstatin A, 0.17 mg ml⁻¹ PMSF and 0.33 mg ml⁻¹ benzamidine. The resuspended cells were stored at −80 °C until purification. UBCH5B[C85K] was expressed in *E. coli* BL21 DE3 cells at 18 °C for 16 h after induction with 0.5 mM IPTG. Cell pellets were resuspended in 1× PBS supplemented with 2 mM 2-mercaptoethanol.

### Protein purification

TCR factors and pig RNA Pol II were purified as described[14,52]. To purify ELOF1, the frozen cell pellet from 6 L of bacterial culture was disrupted by sonication. The lysate was cleared by centrifugation for 30 min at 49,192g, followed by filtration through a 0.8-µm syringe filter. The cleared lysate was applied to a 5-ml HisTrap column (Cytiva), which was equilibrated in lysis buffer containing 20 mM HEPES (pH 7.5), 300 mM NaCl, 1 mM DTT, 10% glycerol (v/v) and 30 mM imidazole. The column was washed with 50 ml of buffer containing 20 mM HEPES (pH 7.5), 1 M NaCl, 1 mM DTT, 10% glycerol (v/v) and 30 mM imidazole, followed by washing with 50 ml of lysis buffer. The bound protein was eluted with a linear gradient from 0% to 100% buffer B (lysis buffer supplemented with 500 mM imidazole). The peak fractions containing 6×His-ELOF1 were pooled, supplemented with TEV protease and dialyzed against the lysis buffer overnight. The dialyzed sample was applied to the 5-ml HisTrap column to remove the TEV protease and undigested 6×His-ELOF1. The unbound fractions containing ELOF1 were combined, concentrated using a 3,000 molecular weight cutoff (MWCO) Amicon Ultra Centrifugal Filter (Merck Millipore) and applied to a Superdex 75 10/300 GL size exclusion column (GE Healthcare), which was pre-equilibrated in lysis buffer without the imidazole. The peak fractions were combined, concentrated, aliquoted, flash frozen in liquid nitrogen and stored at −80 °C. APPBP1–UBA3 was purified as ELOF1 with the following differences: (1) the lysis buffer contained 400 mM NaCl; (2) the washing buffer contained 800 mM NaCl; (3) a 50,000 MWCO centrifugal filter was used; and (4) the last size exclusion step was performed using a Superdex 200 Increase 10/300 GL column (GE Healthcare). To purify UBCH5B[C85K], the clarified cell lysate was incubated with Protino Ni-NTA agarose (Macherey-Nagel) at 4 °C for 1 h while rotating, followed by three washes with PBS supplemented with 2 mM 2-mercaptoethanol. The protein was cleaved from the resin by incubation with ULP1 protease (purified in-house) at 4 °C for 16 h while rotating. Eluted protein was further purified via a Superdex 75 16/600 SEC column (Cytiva) equilibrated in PBS supplemented with 2 mM DTT. The E1 (UBA1) and Ub, as used for the preparation of the UBCH5B–Ub conjugate, were expressed and purified as described previously[51].

### Mass spectrometric identification of crosslinking sites

The complexes for chemical crosslinking were formed in the same way as for cryo-EM analysis of the Pol II–TCR–ELOF1 complex (see below) and purified by size exclusion chromatography in the final buffer containing 100 mM NaCl, 5% glycerol, 20 mM HEPES (pH 7.5), 1 mM MgCl₂ and 1 mM DTT. Peak fractions containing the complex of interest were pulled, supplemented with BS3 (1 mM final) and incubated at 30 °C for 30 min. The crosslinking reaction was quenched with ammonium bicarbonate (50 mM final).

Crosslink analysis was performed as previously described[14]. The crosslinked proteins were reduced with 5 mM DTT for 30 min at 37 °C, followed by alkylation with 20 mM iodoacetamide for 30 min at 25 °C. Unreacted iodoacetamide was quenched with additional 5 mM DTT and incubated for 10 min at 25 °C. Proteins were digested overnight at 37 °C in denaturing conditions (1 M urea) with trypsin (Promega) in a 1:20 (w/w) trypsin-to-protein ratio. The digested sample was acidified with formic acid (FA) to 0.1% (v/v), and acetonitrile (ACN) was added to 5% (v/v) final concentration. Peptides were purified with C18 Micro SpinColumns (Harvard Apparatus) by washing away salts and contaminants with 5% (v/v) ACN and 0.1% (v/v) FA, eluting bound peptides with 80% (v/v) ACN and 0.1% (v/v) FA and drying under vacuum. The sample was resuspended in 30 µl of 30% (v/v) ACN and 0.1% (v/v) trifluoroacetic acid (TFA) and subjected to size exclusion chromatography with a Superdex Peptide PC3.2/30 column (GE Healthcare) at a flow rate of 50 µl min⁻¹ of 30% (v/v) ACN and 0.1% (v/v) TFA. Peptides eluting in the range 1.1–2 ml were collected as 100-µl fractions, dried under vacuum and resuspended in 20 µl of 2% (v/v) ACN and 0.05% (v/v) TFA. Mass spectra were acquired on an Orbitrap Exploris 480 mass spectrometer (Thermo Fisher Scientific) coupled with a Dionex UltiMate 3000 UHPLC system (Thermo Fisher Scientific) and an in-house packed C18 column (ReproSil-Pur 120 C18-AQ, 1.9 µm pore size, 75 µm inner diameter, 30 cm length, Dr. Maisch). The samples were submitted as three 5-µl injection replicates, separated on a 74-min gradient: mobile phase A, 0.1% (v/v) FA; mobile phase B, 80% (v/v) ACN and 0.08% (v/v) FA; flow rate, 300 nl min⁻¹. For the first three fractions, the gradient was formed from 12% to 46% mobile phase B, and precursors with charge states 3–8 were selected for fragmentation. For the rest of the fractions, a gradient from 8% to 42% and charge states 2–8 were used. For MS1 acquisition, the following settings were employed: automatic gain control target, 300%; resolution, 120,000; mass range, 380–1,600 *m/z*; and maximum injection time set to 'Auto'. MS2 spectra were acquired with varying normalized collision energy for the different injection replicates (28%/30%/28–32%) and the following settings: isolation window, 1.6 *m/z*; resolution, 30,000; injection time, 128 ms; automatic gain control target, 100%; and dynamic exclusion, 15 s. Identification of crosslink peptides from raw files was achieved with pLink (version 1.23) (pFind group[53]) and the following parameters: missed cleavage sites, 3; fixed modification, carbamidomethylation of cysteines; variable modification, oxidation of methionines; peptide tolerance, 6 ppm; fragment tolerance, 20 ppm; peptide length, 6–60 amino acids; and spectral false discovery rate, separate/1%. The provided sequence database contained all proteins within the complex. Extended Data Fig. 8 was created with XiNet[54] and the XlinkAnalyzer (version 1.1) plugin in UCSF Chimera (1.13)[55].

### CRL4[CSA] neddylation

Next, 10 µM purified CRL4[CSA] was mixed with 100 nM APPBP1–UBA3, 100 nM UBE2M (R&D Systems) and 10 uM NEDD8 (R&D Systems) in a

buffer containing 100 mM NaCl, 50 mM Tris (pH 8), 2 mM ATP and 5 mM MgCl$_2$ and incubated at 37 °C for 30 min. The reaction was immediately loaded onto a Superdex 200 Increase 10/300 GL column (Cytiva) pre-equilibrated in 300 mM NaCl, 10% glycerol, 20 mM HEPES (pH 7.5) and 1 mM DTT to separate the neddylation machinery from the ligase and perform a buffer exchange step. The peak fractions containing C$^N$RL4$^{CSA}$ were pooled, concentrated and flash frozen. The efficiency of neddylation was monitored by SDS-PAGE, and quantitative shift of the band corresponding to CUL4A indicated that almost all ligase molecules received the modification.

## Synthesis of UBCH5B$^{C85K}$ isopeptide-linked Ub conjugate

Isopeptide-linked UBCH5B–Ub conjugate was isolated as previously described[56]. In brief, 50 µM UBCH5B was incubated with 100 µM Ub, 2 µM E1 and 5 mM ATP in 50 mM Tris (pH 9), 5 mM MgCl$_2$, 10 mM creatine kinase and 0.6 U ml$^{-1}$ creatine phosphokinase at 37 °C for 16 h. The resulting K85-linked UBCH5B–Ub conjugate was purified by size exclusion chromatography using a Superdex 75 26/600 column (Cytiva) equilibrated in PBS including 1 mM TCEP.

## Cryo-EM sample preparation

**Pol II–TCR–ELOF1.** Scaffolds used for the preparation of elongation complexes were as follows: GTA TTC GCT CTG CTC CTT CTC CCA TCC TCT CGA TGG CTA TGA GAT CAA CTA G (template strand); CTA GTT GAT CTC ATA TTT CAT TCC TAC TCA GGA GAA GGA GCA GAG CGA ATA C (non-template strand); and rArUrC rGrArG rArGrG rA (RNA). The template strand and the RNA were annealed in water by heating the solution to 90 °C, followed by cooling to 4 °C at the speed of 1 °C per minute. The elongation complexes were formed by mixing Pol II with the template strand:RNA hybrid in 1:1 ratio, followed by incubation at room temperature for 10 min and the addition of the 1.3× excess of the non-template strand. The elongation complexes were mixed with ELOF1, CSB, CRL4$^{CSA}$ and UVSSA in A 1:3:2:2 ratio in the final buffer containing 100 mM NaCl, 5% glycerol, 20 mM HEPES (pH 7.5), 1 mM MgCl$_2$, 1 mM DTT and 0.3 mM ADP:BeF$_3$. The solution was incubated on ice for 20 min and run over a Superose 6 Increase 3.2/300 column equilibrated in the final complex formation buffer. The peak fractions were pulled, crosslinked with 0.1% glutaraldehyde for 10 min on ice and quenched with lysine (50 mM final) and aspartate (20 mM final). The quenched solution was dialyzed in a Slide-A-Lyzer MINI Dialysis Device of 20 MWCO (Thermo Fischer Scientific) for 7 h against the complex formation buffer without glycerol. After the dialysis, the protein solution was immediately used for the preparation of cryo-EM grids. Then, 4 µl of the sample was applied onto freshly glow-discharged R2/1 carbon grids (Quantifoil), followed by grid blotting for 4 s and plunging into liquid ethane using a Vitrobot Mark IV (FEI) operating at 4 °C and 100% humidity.

**Pol II–TCR–ELOF1–C$^N$RL4$^{CSA}$.** Pol II–TCR–ELOF1–C$^N$RL4$^{CSA}$ was prepared and purified in the same way as the Pol II–TCR–ELOF1 complex, only neddylated E3 ligase was used instead of non-modified ligase.

**C$^N$RL4$^{CSA}$–UVSSA-E2–Ub.** C$^N$RL4$^{CSA}$–UVSSA–E2–Ub was prepared by mixing C$^N$RL4$^{CSA}$ with 2× excess of UVSSA and 10× excess of the E2–Ub conjugate in the complex formation buffer containing 150 mM NaCl, 20 mM HEPES (pH 7.5), 5% glycerol, 1 mM MgCl$_2$ and 1 mM DTT. The solution was incubated at room temperature for 30 min, and the complex was purified by a Superdex 200 Increase 3.2/300 column equilibrated in the complex formation buffer without glycerol. Fractions containing the complex of interest were supplemented with 0.004% n-octyl glucoside (w/v) and immediately used to prepare cryo-EM grids. Then, 4 µl of the sample was applied onto freshly glow-discharged R2/1 carbon grids (Quantifoil), followed by grid blotting for 4 s and plunging into liquid ethane using a Vitrobot Mark IV (FEI) operating at 4 °C and 100% humidity.

**Pol II–CSB–CSA–DDB1–ELOF1.** The Pol II–CSB–CSA–DDB1–ELOF1 complex was assembled and purified in the same way as the Pol II–TCR–ELOF1 complex, but, instead of CRL4$^{CSA}$, we added the CSA–DDB1 complex and omitted the addition of UVSSA.

**C$^N$RL4$^{CSA}$–E2–Ub.** We tried to solve the structure of the Pol II–TCR–ELOF1–C$^N$RL4$^{CSA}$–E2–Ub complex; however, we could only see the E2–Ub conjugate bound to the E3 ligase if we did not crosslink the sample and if we included a small amount of n-octyl glucoside before sample freezing. Thus, we prepared the Pol II–TCR–ELOF1–C$^N$RL4$^{CSA}$–E2–Ub complex in the same way as the Pol II–TCR–ELOF1–C$^N$RL4$^{CSA}$ complex but with the addition of 10× excess of E2–Ub conjugate over Pol II and without sample crosslinking. After the dialysis step, we also supplemented the protein solution with 0.004% n-octyl glucoside. Under these conditions, the Pol II–TCR complex was falling apart during cryo-EM grid preparation, and Pol II exhibited very strong preferred orientation distribution. However, we could solve the structure of the C$^N$RL4$^{CSA}$–E2–Ub subcomplex from dissociated or not-bound ligase particles.

## Data acquisition, image processing and model building

All final models were validated using MolProbity (4.5.1) (Table 1)[57].

## Pol II–TCR–ELOF1

Images of the sample were taken by an FEI Titan Krios transmission electron microscope using a K3 summit direct electron detector (Gatan) and a GIF quantum energy filter (Gatan) operating with a slit width of 20 eV. Acquisition of images was automated with Serial EM (3.8 beta 8)[58]. Data were collected at a magnification of ×81,000 (1.05 Å per pixel) with a dose of 1.05 e/Å$^2$ per frame over 40 frames. In total, 8,356 micrographs were collected and processed on-the-fly using Warp (version 1.0.7)[59], which included estimating the contrast transfer function (CTF), motion correction and particle picking. The first few hundred thousand particles were used for two-dimensional (2D) classification in cryoSPARC (2.14.2)[60], and 135,014 particles from the selected 2D classes were used to make an ab initio model. The ab initio model was used as an input model for heterogeneous refinement of 2,807,926 particles autopicked in Warp (version 1.0.7), which separated the Pol II-containing complexes from junk particles. A further round of heterogeneous refinement was used to select particles that contain stably bound TCR–ELOF1 complex. The particles from selected 3D classes were subjected to CTF refinement and particle polishing in RELION 3.0 (ref. [61]). Further rounds of focused classification and refinements were used to obtain the highest quality reconstruction of individual parts of the complex, as indicated in Extended Data Fig. 1. The final composite map was created from Pol II and TCR-focused refined maps and denoised in Warp (version 1.0.7)[59].

Models for Pol II, CSA, CSA, DDB1 and VHS domain of UVSSA (Protein Data Bank (PDB): 7OO3)[14] were docked into the density, real-space refined in PHENIX (1.18)[62] and manually adjusted in Coot 0.9 (ref. [63]). The downstream DNA was 5 bp longer, so we extended the DNA in Coot 0.9 guided by the density. ELOF1 structure was initially predicted using AlphaFold[38], docked into the density, real-space refined in PHENIX (1.18) and adjusted in Coot 0.9. The C-terminus of UVSSA was identified based on the crosslinking data, and the register was determined by the kink between the last two helices that was also predicted by AlphaFold and by the bulky side chain densities. The element was built manually in Coot 0.9. The last seven amino acids of the C-terminus could be traced in the focused refined density, but, due to flexibility and worse quality of the density, the side chains of those amino acids were truncated. The UVSSA Zn-finger was predicted by AlphaFold and docked into the density. The accuracy of model positioning could be verified by bulky side chain densities. The linker that extends from the Zn-finger element to the pore was built manually by following the density. For Extended Data Fig. 8e, the beta-hairpin of UVSSA was modeled using AlphaFold and manually overlaid onto the domain III of TFIIS in UCSF Chimera (1.13)[64].

## Pol II–TCR–ELOF1–C$^N$RL4$^{CSA}$

Image acquisition and pre-processing were done as for the Pol II–TCR–ELOF1 complex. We collected 18,804 micrographs in total and autopicked 5,367,852 particles in Warp (version 1.0.7)[59]. The first 100,000 particles during data collection were used for ab initio reconstruction of four volumes in cryoSPARC (2.14.2)[60]. The four volumes were used as input classes during heterogeneous classification of the entire binned dataset in cryoSPARC (2.14.2). Particles included in classes corresponding to the Pol II–TCR complex were re-extracted in RELION 3.0 without binning and subjected to CTF refinement and polishing[61]. Further rounds of focused classification are depicted in Extended Data Fig. 5. Reconstruction of the C$^N$RL4$^{CSA}$ lacked a visible density for the RBX1 RING domain, CUL4A WHB domain and NEDD8, suggesting that these parts of the ligase became very flexible.

## Pol II–CSB–CSA–DDB1–ELOF1

Image acquisition and pre-processing were done in the same way as for the Pol II–TCR–ELOF1 complex. In total, 8,980 micrographs were acquired, and 3,458,952 particles were autopicked in Warp (version 1.0.7)[59]. Initial 40,919 particles were used for an ab initio reconstruction in cryoSPARC (2.14.2)[60], and the reconstruction was used as the input model for heterogeneous refinement. The first two rounds of heterogeneous refinement were used to remove junk particles or Pol II classes that did not show extra density for TCR factors. The third classification round nicely separated the Pol II particles with CSA properly docked on ELOF1 and Pol II particles with CSA not docked onto ELOF1. We further processed the docked particles to demonstrate final data quality and, to check if CSA–ELOF1 docking interface is unchanged, compared to the complex in presence of UVSSA. Unbinned particles corresponding to the best class were re-extracted in RELION 3.0 and subjected to CTF refinement and particle polishing[61]. The next rounds of 3D classifications, focused refinements and focused classifications are described in Extended Data Fig. 4. The final composite map was created from Pol II and TCR-focused refined maps and denoised in Warp (version 1.0.7)[59]. Models for Pol II, DNA, CSB, CSA and DDB1 were taken from the Pol II–TCR–ELOF1 structure and rigid body fitted into the final map in UCSF Chimera (1.13)[64]. Fitted models were combined in Coot 0.9 (ref. [63]), and real space was refined against the final map in PHENIX (1.18)[62].

## C$^N$RL4$^{CSA}$–E2–Ub

Data collection for this sample was performed as described for the Pol II–TCR–ELOF1 complex, except the data were collected on a stage tilted 40°. In total, 3,546 images were collected, and 1,288,908 particles were autopicked in Warp (version 1.0.7)[59]. Ab initio reconstruction on the first 228,094 particles in cryoSPARC (2.14.2)[60] produced three maps used as an input for heterogeneous refinement of all picked particles. After further rounds of classification, the final map was refined using non-uniform refinement. A histogram and a directional Fourier shell correlation (FSC) plot were created in cryoSPARC (2.14.2) using ThreeDFSC. The final refined map was used for model building. First, we docked the CSA and DDB1 models from the Pol II–TCR–ELOF1 structure into the density using UCSF Chimera (1.13). BPB and DDB1 were fitted separately and connected to the rest of the protein in Coot 0.9 (ref. [63]). The CUL4A subunit structure was predicted with AlphaFold[38], and the domains of the protein were fitted into the density in UCSF Chimera (1.13) and adjusted and connected in Coot 0.9. The density of the helix connecting C/R and WHB domains was visible only at higher map thresholds, likely due to its partial unfolding and increased flexibility, and it was left out of the final model. The models for NEDD8, Ub and UBCH5B were also predicted with AlphaFold and docked into the density in UCSF Chimera (1.13). The final model was assembled and manually adjusted in Coot 0.9, followed by real-space refinement against the final map in PHENIX (1.18)[62].

## C$^N$RL4$^{CSA}$–UVSSA–E2–Ub

Images of the sample were taken on a 200-keV Glacios Cryo-Transmission Electron Microscope (Thermo Fisher Scientific) using a Falcon 3 direct electron detector. Data were collected at a magnification of ×120,000 (1.23 Å per pixel) with a dose of 1.34 e/Å$^2$ per frame over 30 frames. In total, 2,592 micrographs were collected and processed in Warp (version 1.0.7)[59]. Automated picking of 1,201,107 particles was performed in Warp (version 1.0.7). The first 37,095 particles obtained during the data collection were used for ab initio reconstruction in cryoSPARC (2.14.2), and the reconstruction was used as an input model for heterogeneous refinement of all particles. Subsequent rounds of heterogeneous refinement were used to isolate the class with a stably bound UVSSA VHS domain. The final class was refined with non-uniform refinement. A histogram and a directional FSC plot were created in cryoSPARC (2.14.2) using ThreeDFSC. The model was built by fitting the components of the C$^N$RL4$^{CSA}$–E2–Ub from the corresponding structure solved in the absence of UVSSA into the density in UCSF Chimera (1.13), followed by manual adjustment and model assembly in Coot 0.9. The VHS domain of UVSSA was taken from the Pol II–TCR–ELOF1 structure and rigid-body docked into the density with UCSF Chimera (1.13). The final model was real-space refined in PHENIX (1.18)[62].

## TFIIS-induced RNA cleavage assay

The assay was used to check if UVSSA interferes with TFIIS-induced RNA cleavage by Pol II. Elongation complexes were formed the same way as for cryo-EM, only 1.2× excess of Pol II over the template strand:RNA hybrid was used. The scaffold sequences were as follows: CGC TCT GCT CCT TCT CCC ATC CTC TCG ATG GCT ATG AGA TCA ACT AG (template stand); CTA GTT GAT CTC ATA TTT CAT TCC TAC TCA GGA GAA GGA GCA GAG CG (non-template stand); and /56-FAM/rUrArC rArArA rArUrC rGrArG rArGrG rArCrC (RNA). The elongation complexes (0.3 μM) were mixed with ELOF1, CSA–DDB1, UVSSA and CSB in a 1:2:1.2:1.2:1.2 ratio in the final buffer containing 100 mM NaCl, 20 mM HEPES (pH 7), 1 mM MgCl$_2$ and 1 mM DTT. In the reaction without UVVSA, UVSSA was omitted from the mix. The reactions were started by the addition of TFIIS (5 nM, 10 nM, 20 nM and 40 nM) and incubated for 10 s at room temperature. Reactions were quenched with 2× quenching buffer (7 M urea in TBE buffer, 20 mM EDTA and 10 μg ml$^{-1}$ proteinase K (Thermo Fisher Scientific)) and digested for 30 min at 37 °C. RNA products were separated by denaturing gel electrophoresis (20% Bis-Tris acrylamide 19:1 gel in 1× TBE buffer) and visualized by using a Typhoon FLA 9500 imager (GE Healthcare Life Sciences). Images were quantified with ImageJ version 1.48 (ref. [65]) and plotted with GraphPad Prism 8 (version 8.4.2).

## Pol II ubiquitylation assay

Pol II elongation complexes were formed in the same way as for the Pol II–TCR–ELOF1 structure. Elongation complexes (150 nM final) were mixed with ELOF1 (150 nM), C$^N$RL4$^{CSA}$ (100 nM), ATPase deficient CSB (100 nM) and UVSSA (no UVSSA, 150 nM, 300 nM and 600 nM). The reaction was supplemented with Ub (100 μM), UBCH5B (1.75 μM), E1 (100 nM) and ATP (2 mM) in final reaction buffer containing 50 mM Tris-Cl (pH 7.9), 4 mM MgCl$_2$ and 50 mM NaCl. Reactions were incubated at 37 °C for 30 min and quenched by the addition of EDTA (10 mM final). The proteins were separated on a 3–8% Tris-acetate gel (Invitrogen) and transferred onto a PVDF membrane using a Trans-Blot Transfer System (Bio-Rad). The membrane was incubated with 5% (w/v) milk in PBS supplemented with 0.1% Tween 20 (PBST) for 1 h at room temperature. After washing with PBST, the membrane was incubated with fluorescently labeled 8WG16 antibodies (1:1,000) in PBST and 2% milk (w/v) for 1 h at room temperature. The membrane was washed three times with PBST, and the antibody was visualized by scanning the membrane with the Typhoon FLA 9500.

## Cell lines

All cell lines are listed in Supplementary Table 2. RPE1-iCas9, U2OS (FRT) and HEK293T cells were cultured at 37 °C in an atmosphere of 5% CO$_2$ in DMEM GlutaMAX (Thermo Fisher Scientific) supplemented with penicillin–streptomycin (Sigma-Aldrich) and 10% FBS (Bodinco BV or Thermo Fischer Scientific (Gibco)). Sf9 insect cells (Thermo Fisher Scientific, 12659017) were cultured in Sf-9000TM III SFM medium (Thermo Fisher Scientific) at 27 °C. Hi5 and Sf21 cells (Expression Systems, 94-002F and 94-003F, respectively) were cultured in ESF921 medium (Expression Systems, 96-001-01) at 27 °C.

## Generation of KO cells

Parental RPE1-hTERT cells stably expressing inducible Cas9 (iCas9) that are also KO for *TP53* and the puromycin *N*-acetyltransferase *PAC1* gene were described previously (referred to as RPE1-iCas9)[12]. RPE1-iCas9 cells were transfected with Cas9-2A-GFP (pX458, Addgene 48138) containing a guide RNA from the TKOv3 library using Lipofectamine 2000 (Invitrogen). The used sgRNAs are listed in Supplementary Table 3, and plasmids are listed in Supplementary Table 4. Cells were FACS sorted on GFP and plated at low density, after which individual clones were isolated, expanded and verified by western blot analysis and/or Sanger sequencing using the primers listed in Supplementary Table 5. ELOF1-KO clone 2–16, UVSSA-KO clone 3–9 and CSA-KO clone 3–8 were used for further analysis.

## Plasmids

All plasmids are listed in Supplementary Table 4. A region spanning the PGK promoter was amplified by PCR and used to replace the cytomegalovirus (CMV) promoter in pGFP-C1-IRES-Puro[66]. The ELOF1$^{WT}$ cDNA was amplified by PCR (primers for cloning are listed in Supplementary Table 1) and inserted into pPGK-GFP-N1-IRES-Puro as described[12]. Three 270-bp fragments of ELOF1 were synthesized (Gene Universal) as *NheI–AgeI* fragments, including the N30A–H31A–E32A (referred to as Δdock), the E55A–E79A or all 5A substitutions. These fragments were inserted into pPGK-GFP-ELOF1$^{WT}$-IRES-Puro. The UVSSA$^{WT}$ cDNA was amplified by PCR. Mutants of UVSSA were generated by overlap extension PCR. To tag UVSSA on its N-terminus, WT and mutant UVSSA fragments were inserted into pPGK-GFP-C1-IRES-Puro backbone using *BglII–EcoRI* restriction sites. The CSA$^{WT}$ cDNA was amplified by PCR. The CSA$^{Y334A}$ mutant was generated by overlap extension PCR. GFP-tagged CSA$^{WT}$ and CSA$^{Y334A}$ were subcloned as *BglII–SalI* fragments from pPGK-CSA$^{WT/Y334A}$-GFP-IRES-Puro plasmids into lentiviral expression plasmid pLenti-PGK-GFP-Puro linearized by *BamHI–SalI*. Plasmids and oligos are listed in Supplementary Tables 1 and 4. All sequences were verified by Sanger sequencing.

## Generation of stable cell lines

RPE1-hTERT ELOF1-KO clone 2–16 was transfected with pPGK-ELOF1-GFP-IRES-Puro plasmids (WT, N30A–H31A–E32A (Δdock), E55A–E79A or 5A) using Lipofectamine 2000 (Invitrogen). RPE1-hTERT UVSSA-KO clone 3–9 was transfected with pPGK-GFP-UVSSA-IRES-Puro plasmids (WT, Δ667–699 (ΔC), K679A–R683A, C567A–C577A (ΔZnF2) and C567A–C577A–C585A–H588A (ΔZnF4)) using Lipofectamine 2000 (Invitrogen). Cells were selected on 1 µg ml$^{-1}$ puromycin for 2 weeks and subsequently sorted on GFP to ensure homogenous expression, which was confirmed by fluorescence microscopy and western blotting.

## Lentiviral transduction

For lentiviral particle production, GFP in the lentiviral vector pLenti-PGK-GFP-Puro was replaced with CSA$^{WT}$–GFP or CSA$^{Y334A}$–GFP. Then, HEK293T cells (American Type Culture Collection, CRL-3216) were transfected with vectors expressing CSA$^{WT/Y334A}$–GFP fusions, VSV-G, RRE and REV using PEI (Sigma-Aldrich). The virus-containing supernatant was collected after 24 h and filtered with a 0.44-µm filter. RPE1-hTERT CSA-KO clone 3–8 was transduced with lentiviral

particles in the presence of 4 µg ml$^{-1}$ polybrene (Sigma-Aldrich) and 10 mM HEPES (pH 7.6). After 24 h, cells were selected with 1 µg ml$^{-1}$ puromycin. Expression of the GFP-tagged CSA proteins was checked by fluorescence microscopy and western blotting.

## Detection of TCR mutants in genomic DNA by Sanger sequencing

Genomic DNA was isolated by resuspending cell pellets in whole cell lysate (WCE) buffer (50 mM KCL, 10 mM Tris (pH 8.0), 25 mM MgCl$_2$, 0.1 mg ml$^{-1}$ gelatin, 0.45% Tween 20 and 0.45% NP-40) containing 0.1 mg ml$^{-1}$ proteinase K (EO0491, Thermo Fisher Scientific) and incubating for 1 h at 56 °C, followed by a 10-min heat inactivation of proteinase K at 96 °C. Fragments of approximately 1 kilobase (kb) spanning the introduced mutations were PCR amplified, followed by Sanger sequencing using the primers listed in Supplementary Table 5.

## Recovery of RNA synthesis

Cells were irradiated with UV-C light (9 J m$^{-2}$ or 12 J m$^{-2}$), allowed to recover for the indicated periods and pulse labeled with 400 µM 5-ethynyl-uridine (5-EU; Jena Bioscience) for 1 h, followed by a 15-min medium chase with DMEM without supplements. Cells were fixed with 3.7% formaldehyde in PBS for 15 min, permeabilized with 0.5% Triton X-100 in PBS for 10 min at room temperature and blocked in 1.5% BSA (Thermo Fisher Scientific) in PBS. Nascent RNA was visualized by click-it chemistry, labeling the cells for 1 h with a mix of 60 µM Atto azide-Alexa 594 (Atto Tec), 4 mM copper sulfate (Sigma-Aldrich), 10 mM ascorbic acid (Sigma-Aldrich) and 0.1 µg ml$^{-1}$ DAPI in a 50 mM Tris buffer (pH 8). Cells were washed extensively with PBS and mounted in Polymount (Brunschwig).

## Incision assay (γH2AX after trabectedin)

Cells were treated with 10 nM trabectedin (MedChemExpress) for 4 h. During the last 15 min, 20 µM 5-ethynyl-2′-deoxyuridine (5-EdU; Jena Bioscience) was added. Cells were then washed with 300 mM sucrose (Merck) in PBS on ice, pre-extracted with 300 mM sucrose and 0.25% Triton X-100 in PBS for 2 min on ice and fixed with 3.7% formaldehyde in PBS for 15 min at room temperature. Cells were then permeabilized with 0.5% Triton X-100 in PBS for 10 min at room temperature and blocked in 3% BSA (Thermo Fisher Scientific) in PBS. Dividing cells were visualized by click-it chemistry, labeling the cells for 30 min with a mix of 60 µM Atto azide-Alexa 594 or Atto azide-Alexa 647 (Atto Tec), 4 mM copper sulfate (Sigma-Aldrich) and 10 mM ascorbic acid (Sigma-Aldrich) in a 50 mM Tris buffer. After washing with PBS, cells were blocked with 100 mM glycine (Sigma-Aldrich) in PBS for 10 min at room temperature and subsequently with 0.5% BSA and 0.05% Tween 20 in PBS for 10 min at room temperature. To visualize γH2AX, cells were incubated with a primary antibody for phospho-Histone H2A.X Ser139 (JBW301, Merck) for 2 h at room temperature and then with a secondary antibody, anti-mouse Alexa 555 (A-21424, Thermo Fisher Scientific) or anti-mouse Alexa 647 (A-21235, Thermo Fisher Scientific), and DAPI for 1 h at room temperature and mounted in Polymount (Brunschwig).

## Microscopic analysis of fixed cells

Images of fixed samples were acquired on a Zeiss Axio Imager M2 or a D2 wide-field fluorescence microscope equipped with ×63 PLAN APO (1.4 NA) oil immersion objectives (Zeiss) and an HXP 120 metal-halide lamp used for excitation. Fluorescent probes were detected using the following filters for DAPI (excitation filter: 350/50 nm, dichroic mirror: 400 nm, emission filter: 460/50 nm), Alexa 555/594 (excitation filter: 545/25 nm, dichroic mirror: 565 nm, emission filter: 605/70 nm) or Alexa 647 (excitation filter: 640/30 nm, dichroic mirror: 660 nm, emission filter: 690/50 nm). Images were recorded using ZEN 2012 (Blue edition, version 1.1.0.0) and analyzed in ImageJ (version 1.47 and version 1.48). Graphs were plotted and analyzed using GraphPad Prism 8 (version 8.4.2), Microsoft Excel 365, PlotsOfData webtool[67] and Adobe Illustrator 2021.

## Immunoprecipitation of endogenous Pol II and GFP-tagged proteins

Cells were mock treated or irradiated with UV-C light (9 J m$^{-2}$) and harvested 1 h after UV. For endogenous Pol II-S2 immunoprecipitation, chromatin-enriched fractions were prepared by lysing the cells for 20 min on a rotating wheel at 4 °C in 1 ml of EBC-1 buffer (50 mM Tris (pH 7.5), 150 mM NaCl, 2 mM MgCl$_2$, 0.5% NP-40 and protease inhibitor cocktail (Roche)), followed by centrifugation and removal of the supernatant. The chromatin-enriched cell pellets were resuspended in 1 ml of ECB-1 buffer supplemented with 500 U ml$^{-1}$ Benzonase Nuclease (Novagen) and 2 µg of Pol II-S2 (ab5095, Abcam) for 1 h at 4 °C. Then, the salt concentration was increased to 300 mM NaCl, and the samples were incubated for another 30 min on a rotating wheel at 4 °C. Samples were then centrifuged for 10 min at 14,000 r.p.m. at 4 °C. Next, 50 µl of the supernatants was saved as input fraction, and the rest was transferred to fresh tubes. The protein complexes were immunoprecipitated by incubation with 20 µl of pre-washed Protein A agarose beads (Millipore) for 90 min at 4 °C. After incubation, the beads were washed six times with ECB-2(300) buffer (50 mM Tris (pH 7.5), 300 mM NaCl, 1 mM EDTA, 0.5% NP-40 and protease inhibitor cocktail (Roche)).

For immunoprecipitation of GFP-tagged proteins, chromatin-enriched fractions were prepared as described above. The chromatin pellets were resuspended in 1 ml of ECB-1 buffer supplemented with 500 U ml$^{-1}$ Benzonase Nuclease (Novagen) for 1 h at 4 °C. Then, the salt concentration was increased to 450 mM NaCl, and the samples were incubated for another 30 min on a rotating wheel at 4 °C. Samples were then centrifuged for 10 min at 14,000 r.p.m. at 4 °C. Next, 50 µl of the supernatants was saved as input fraction, and the rest was transferred to fresh tubes. The protein complexes were immunoprecipitated by incubation with 20 µl of pre-washed GFP-Trap (ChromoTek) for 90 min at 4 °C. After incubation, the beads were washed four times with ECB-2(450) buffer (50 mM Tris (pH 7.5), 450 mM NaCl, 1 mM EDTA, 0.5% NP-40 and protease inhibitor cocktail (Roche)) and twice with ECB-2(300) buffer. For subsequent analysis by western blotting, the samples were boiled for 10 min in Laemmli SDS sample buffer.

## Western blotting

Proteins were separated on Criterion XT Tris-Acetate 3–8% Protein Gels (Bio-Rad, 3450131) in Tris/Tricine/SDS running buffer (Bio-Rad, 1610744) or on Criterion Xt Bis-Tris 4–12% gels in MOPS running buffer and then blotted onto PVDF membranes (IPFL00010, EMD Millipore) in Tris/Glycine blotting buffer with 20% methanol. Membranes were blocked with blocking buffer (Rockland, MB-070-003) or 5% fat-free milk in PBS with 0.1% Tween 20 for 1 h at room temperature. Membranes were then probed with indicated antibodies in 5% fat-free milk in PBS with 0.1% Tween 20 or Rockland blocking buffer (antibodies are listed in Supplementary Table 6). Proteins were stained with fluorochrome-conjugated secondary antibodies. Western blot images were acquired using an Odyssey CLx with Image Studio Lite software (version 5.2).

## Reporting summary

Further information on research design is available in the Nature Portfolio Reporting Summary linked to this article.

## Data availability

The electron density reconstructions and structure coordinates were deposited to the Electron Microscopy Database (EMDB) and to the Protein Data Bank (PDB) under the following accession codes: EMDB EMD-15825 and PDB 8B3D for the Pol II–TCR–ELOF1 structure; EMDB EMD-15829 and PDB 8B3I for the C$^N$RL4$^{CSA}$–E2–Ub structure; and EMBD EMD-15827 and PDB 8B3G for the C$^N$RL4$^{CSA}$–E2–Ub–UVSSA structure. The crosslinking mass spectrometry data have been deposited to the ProteomeXchange Consortium via PRIDE with the dataset identifier PXD042388. Source data are provided with this paper.

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

## Acknowledgements

We thank current and former members of the Cramer and Luijsterburg laboratories. S.L. acknowledges support by the Deutsche Forschungsgemeinschaft (SFB1565; grant 469281184; P17). P.C. was supported by the Deutsche Forschungsgemeinschaft (SFB860 and SPP1935), the European Research Council Advanced Investigator Grant TRANSREGULON (grant agreement no. 693023) and the Volkswagen Foundation. M.S.L. was supported by the Netherlands Scientific Organization (ENW grant OCENW.M20.056 and Vici grant VI.C.212.005) and the European Research Council Consolidator Grant STOP-FIX-GO (grant agreement no. 101043815). The funders had no role in study design, data collection and analysis, decision to publish or preparation of the manuscript.

## Author contributions

G.K. carried out all in vitro and cryo-EM experiments. G.Y. and D.v.d.H. generated and validated all cell lines expressing GFP-tagged TCR proteins. G.Y. performed all co-immunoprecipitation experiments in RPE1 cells. A.P.W., D.v.d.H. and G.Y. performed recovery of RNA synthesis experiments. P.J.v.d.M. and G.Y. performed trabectedin assays. Y.v.d.W. performed co-immunoprecipitation experiments in U2OS cells. I.F. prepared ELOF1. A.C. carried out crosslinking mass spectrometry. H.U. supervised mass spectrometry. T.J.F. prepared the E2–Ub conjugate, under the supervision of S.L. P.C. designed and supervised research. G.K., G.Y., P.C. and M.S.L. interpreted the data and wrote the manuscript, with input from all authors.

## Funding

## Competing interests

The authors declare no competing interests.

## Additional information

**Extended data** is available for this paper at https://doi.org/10.1038/s41594-023-01207-0.

**Correspondence and requests for materials** should be addressed to Patrick Cramer or Martijn S. Luijsterburg.

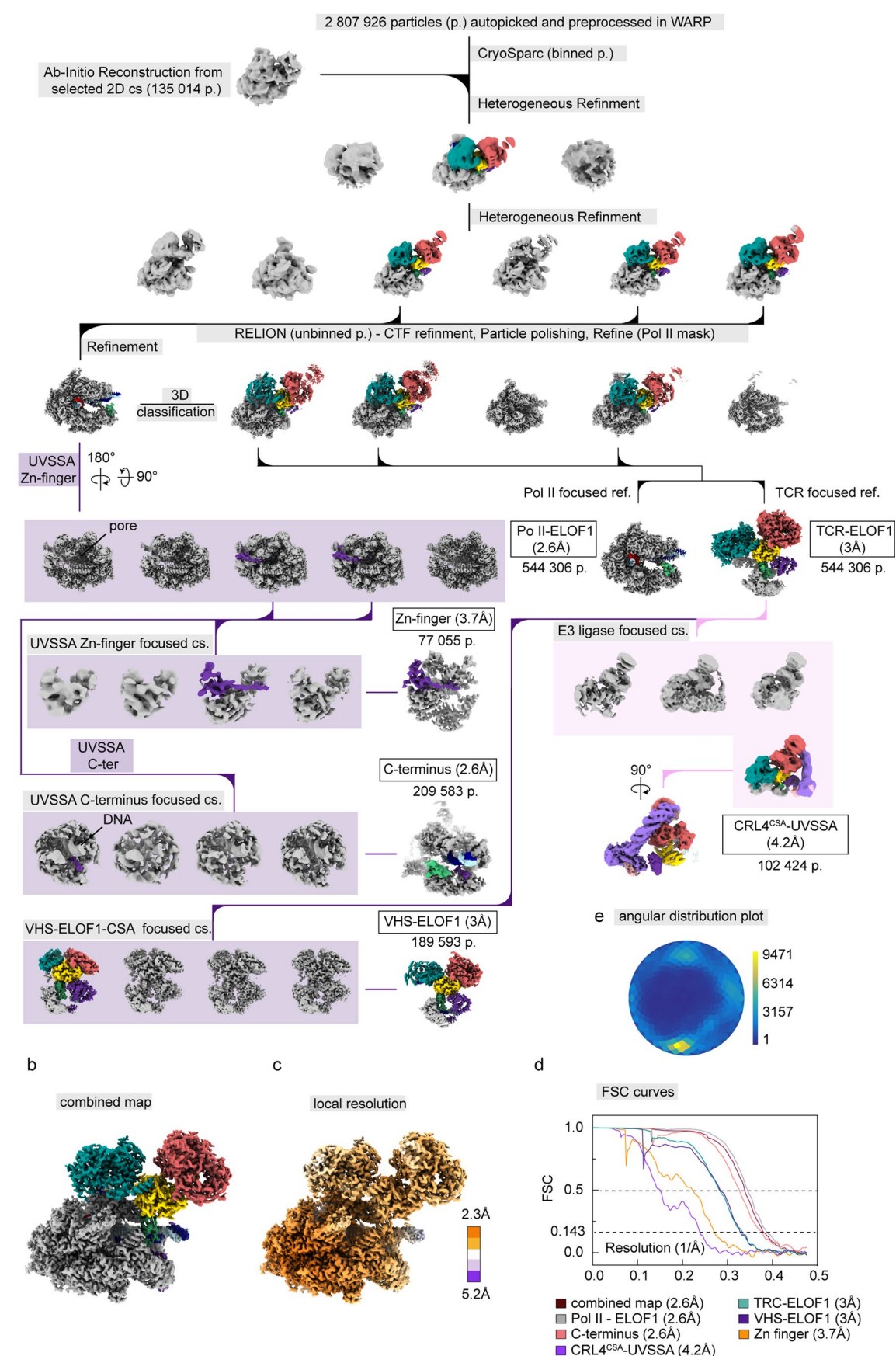

**Extended Data Fig. 1 | Cryo-EM analysis of the Pol II-TCR-ELOF1 complex.**
**a**. Processing tree. The number of particles included in refined maps is indicated below the map, together with the map resolution. Classes used for further processing are color-coded according to complex subunits. **b**. Final composite map. **c**. Local resolution estimate for the composite map. **d**. Fourier shell correlation plot for all focused refined maps and the composite map. **e**. Angular distribution plot for the high-resolution Pol II class used as a starting point for focused classifications.

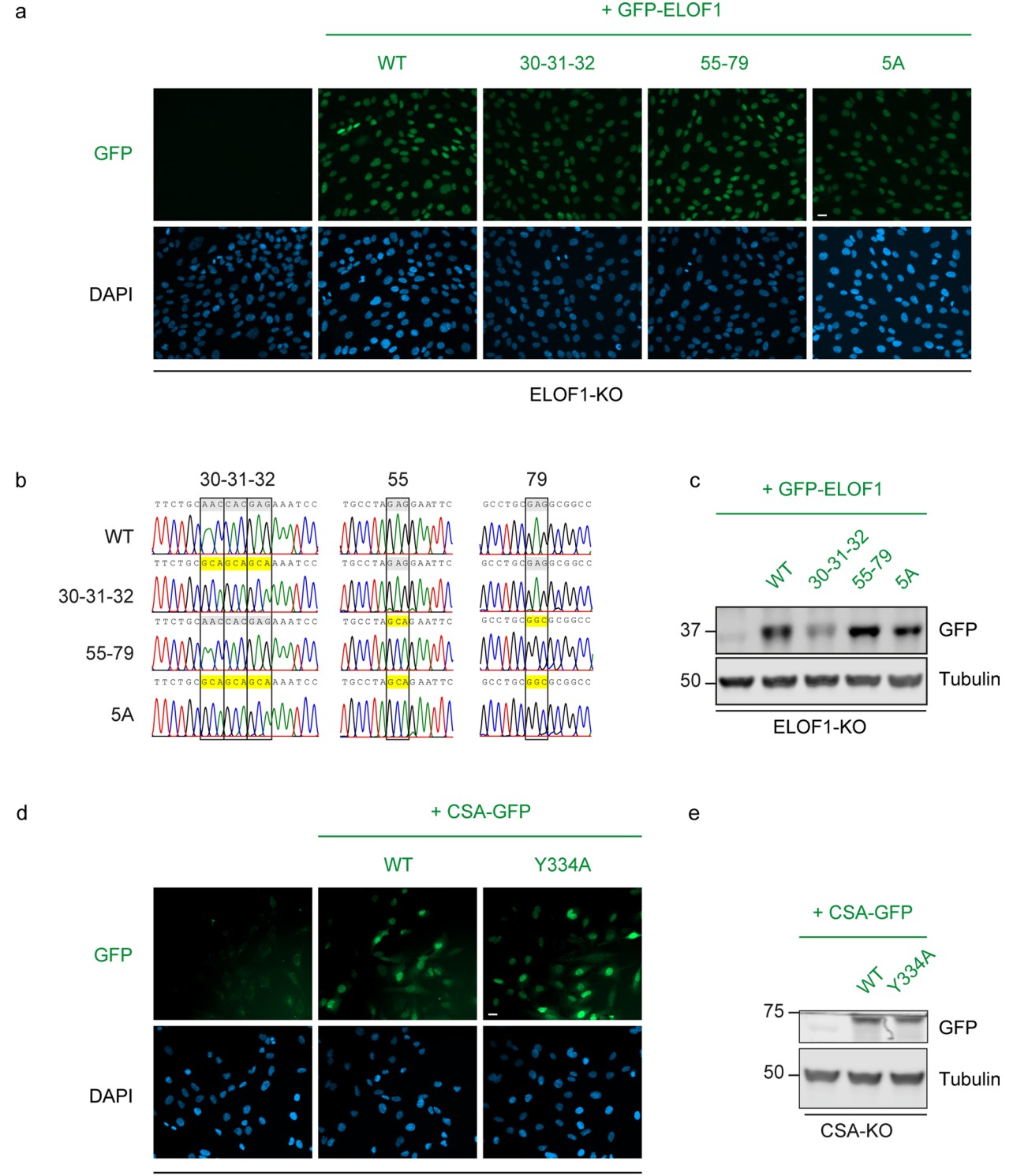

**Extended Data Fig. 2 | Sequencing and expression of ELOF1 and CSA mutants.**
**a**. Expression of the indicated GFP-ELOF1 proteins (or empty control) in RPE1-iCas9 ELOF1-KO cells. The data shown represent at least three independent experiments. Scale bar, 15 μm. **b**. Sanger sequencing of genomically integrated cDNA to confirm the presence of mutations. **c**. Western blot analysis of GFP-ELOF1 proteins (or empty control) in RPE1-iCas9 ELOF1-KO cells. The data shown represent at least three independent experiments. **d**. Expression of the indicated CSA-GFP proteins (or empty control) in RPE1-iCas9 CSA-KO cells. The data shown represent at least three independent experiments. Scale bar, 15 μm. **e**. Western blot analysis of CSA-GFP proteins (or empty control) in RPE1-iCas9 CSA-KO cells. The data shown represent at least three independent experiments.

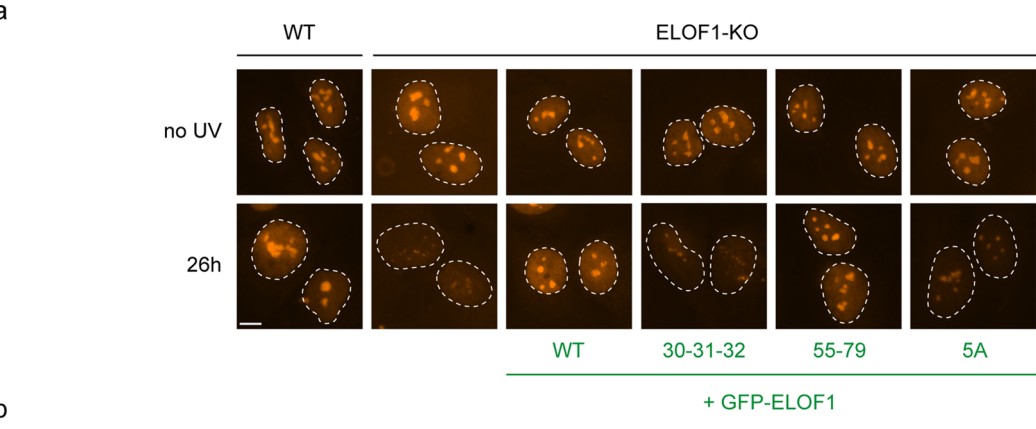

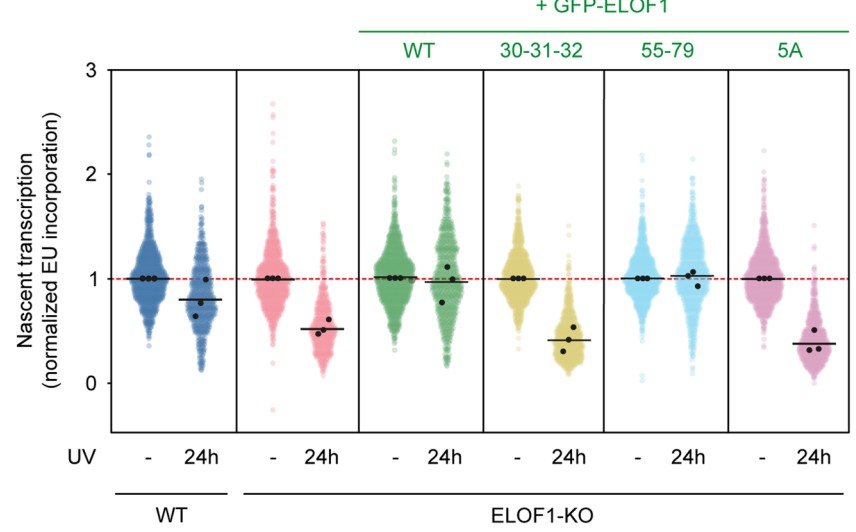

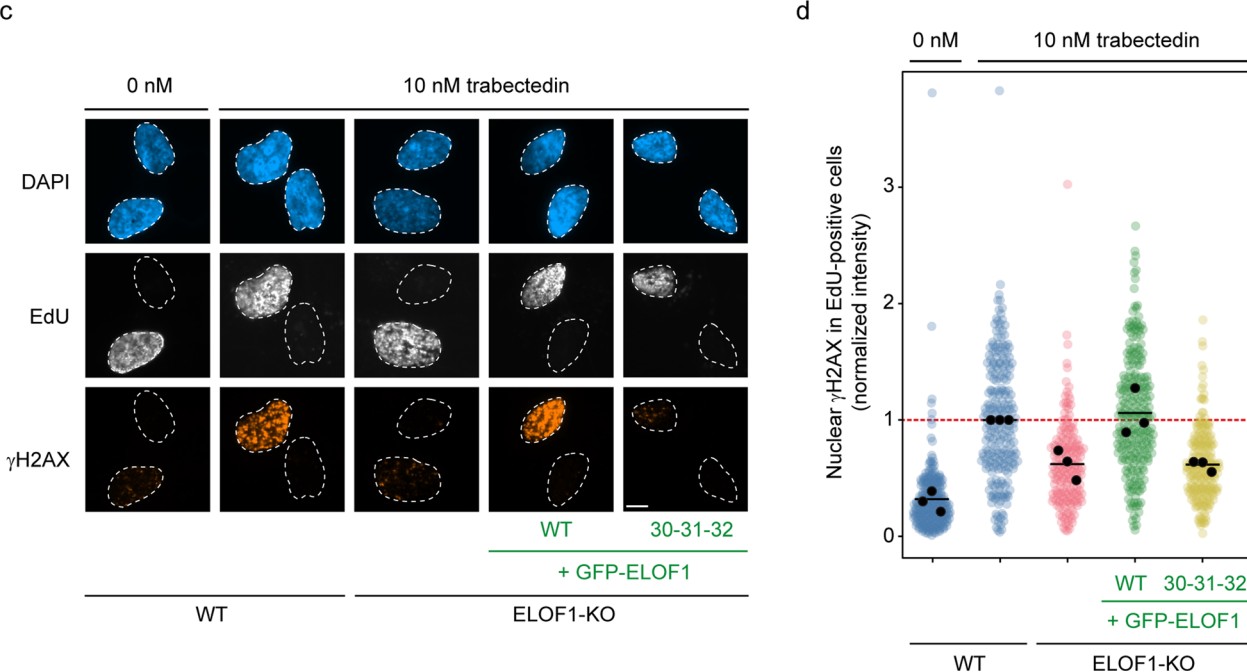

**Extended Data Fig. 3 | See next page for caption.**

**Extended Data Fig. 3 | Transcription recovery supported by ELOF1 mutants.**
**a, b**. Measuring transcription recovery (RRS) by 5-ethynyl-uridine labelling in the indicated RPE1-iCas9 cells following UV irradiation (24 h; 12 J/m$^2$). Representative images (**a**) and quantification (**b**) of 5-EU levels normalized to mock treatment for each cell line. The experiment was performed three times. Each black circle represents the mean of two technical replicates, with >80 cells collected per technical replicate. The black lines represent the mean of all three independent experiments. Scale bar, 10 μm. **c, d**. Measuring TCR-dependent γH2AX induction following trabectedin exposure in replicating cells labelled with 5-ethynyl-deoxyuridine (EdU) in the indicated RPE1-iCas9 cells. Representative images (**c**) and quantification (**d**) of γH2AX levels normalized to trabectedin-treated WT cells within each experiment. The experiment was performed three times. Each coloured circle represents 1 cell. Each black circle represents the mean of two technical replicates, with >70 cells collected per technical replicate. The black lines represent the median of all three independent experiments. Scale bar, 10 μm.

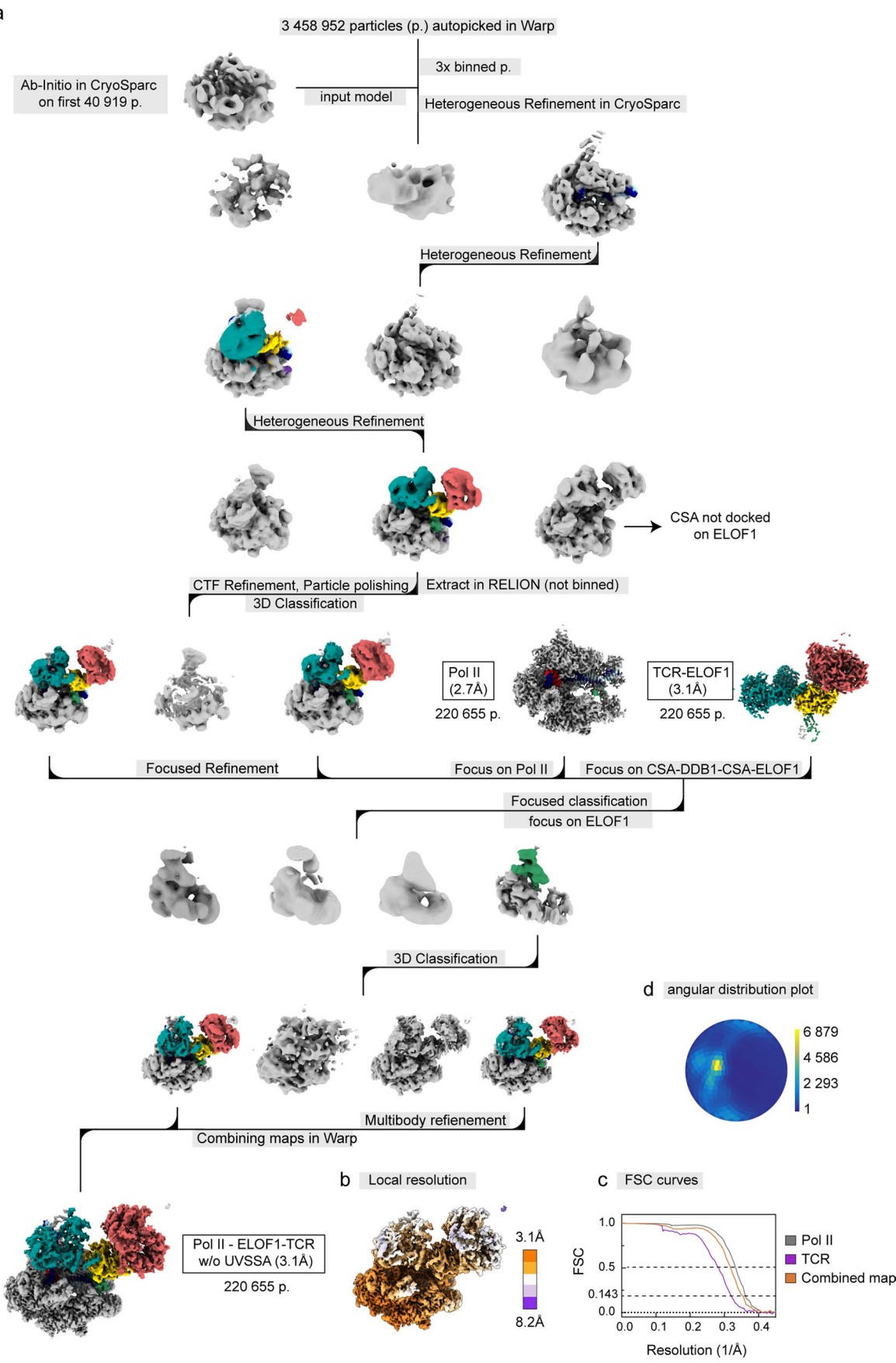

**Extended Data Fig. 4 | See next page for caption.**

**Extended Data Fig. 4 | Cryo-EM analysis of the Pol II-CSA-DDB1-CSB-ELOF1 complex. a**. Processing tree. The number of particles included in refined maps is indicated below the map, together with the map resolution. Classes used for further processing are color-coded according to complex subunits. Classification revealed particles with CSA docked onto ELOF1 and particles with CSA not docked onto ELOF1. Particles with docked CSA were further processed to demonstrate data quality. **b**. Local resolution estimate for the composite map. **c**. Fourier shell correlation plot for all focused refined maps and the composite map. **d**. Angular distribution plot for the high-resolution Pol II class used as a starting point for focused classifications.

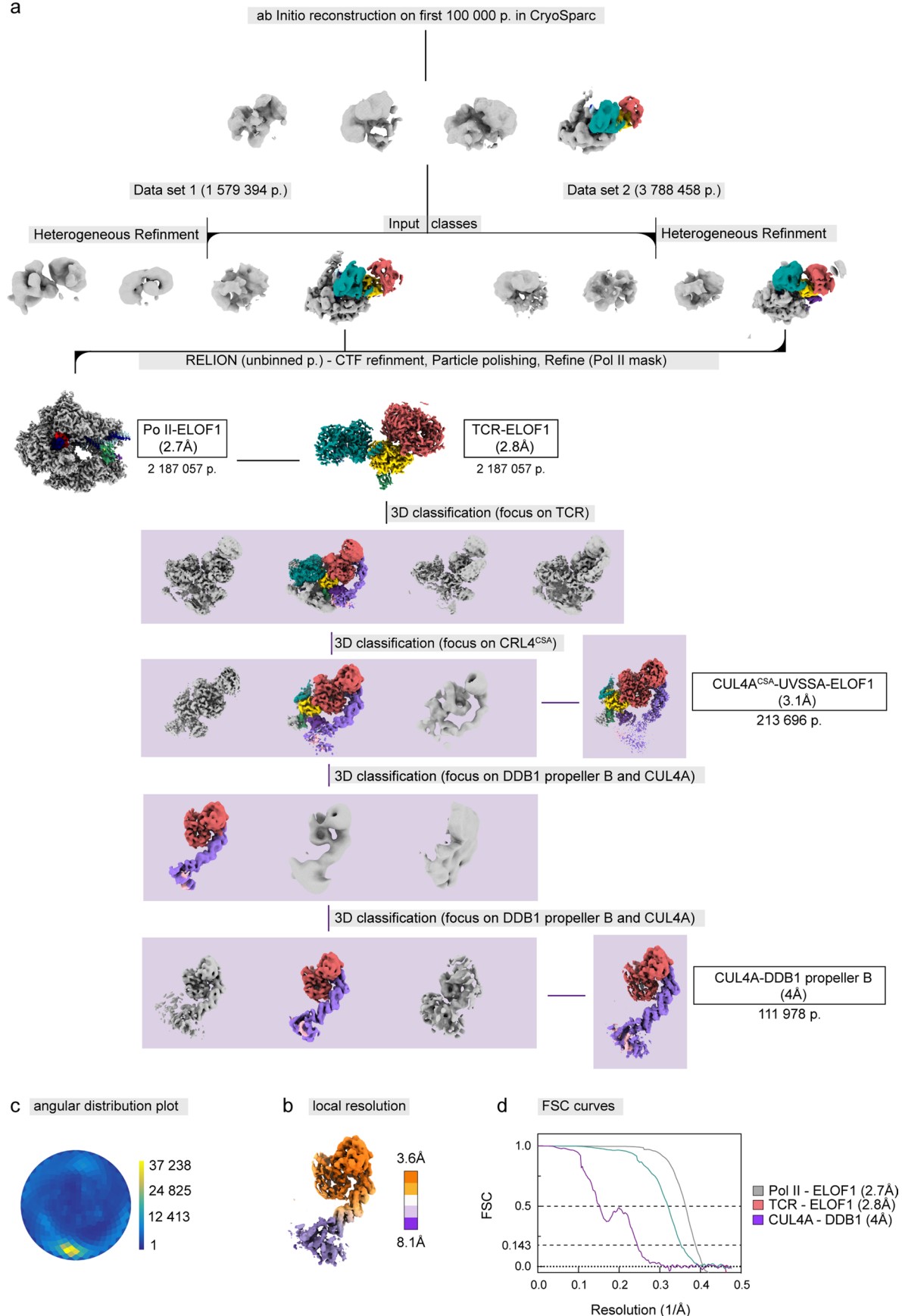

**Extended Data Fig. 5 | Cryo-EM analysis of the Pol II-TCR-ELOF1-NEDD8 complex. a.** Processing tree. The number of particles included in refined maps is indicated below the map, together with the map resolution. Classes used for further processing are color-coded according to complex subunits. **b.** Local resolution estimate for the DDB1 BPB-CUL4A-RBX1 map. **c.** Angular distribution plot for the high-resolution Pol II class used as a starting point for focused classifications. **d.** Fourier shell correlation plot for all focused refined maps.

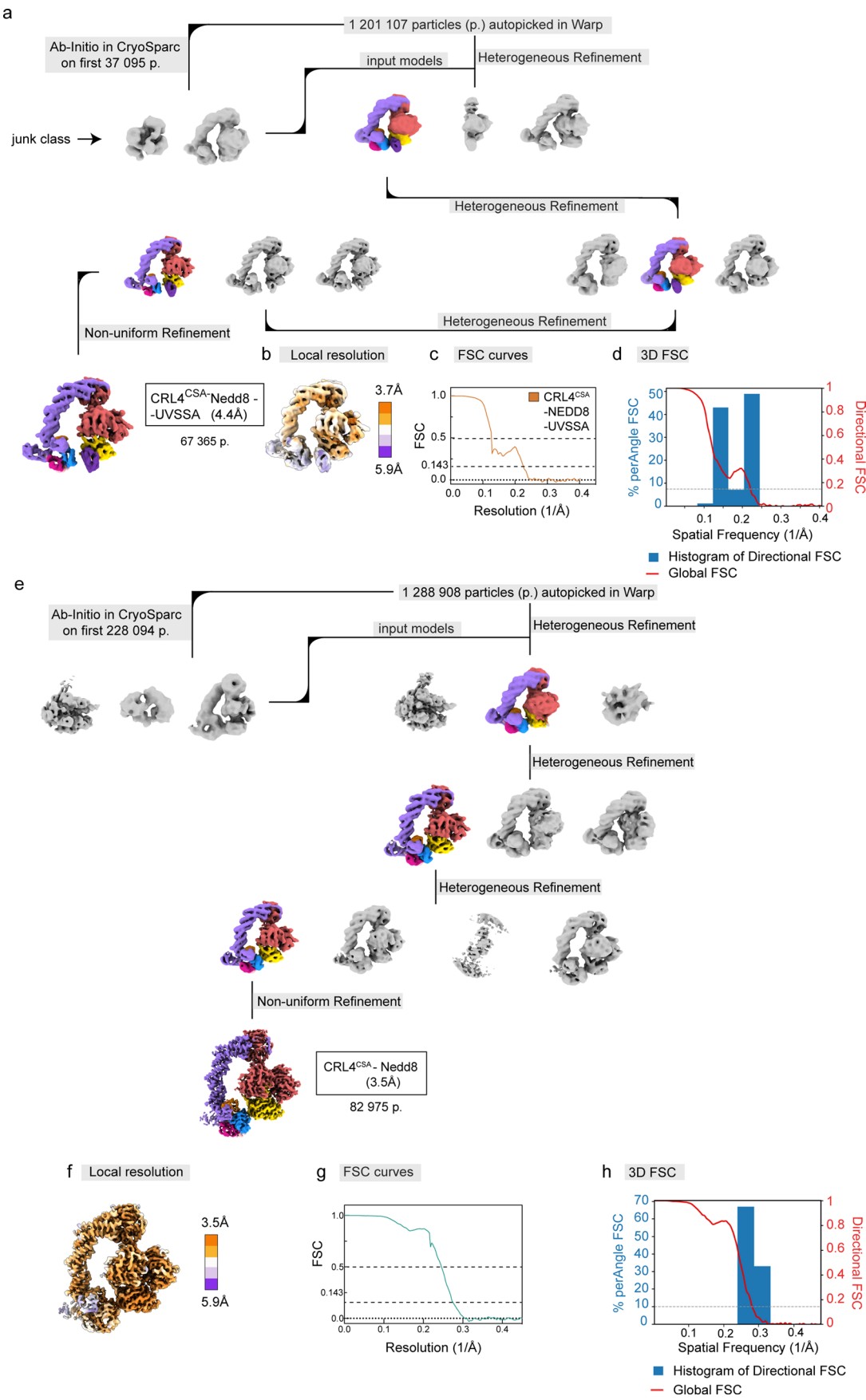

**Extended Data Fig. 6 | See next page for caption.**

**Extended Data Fig. 6 | Cryo-EM analysis of C$^N$RL4$^{CSA}$-E2-Ub-UVSSA and C$^N$RL4$^{CSA}$-E2-Ub complexes. a**. Processing tree for the C$^N$RL4$^{CSA}$-E2-Ub-UVSSA complex. The number of particles included in refined maps is indicated below the map, together with the map resolution. Classes used for further processing are color-coded according to complex subunits. **b**. Local resolution estimate for the C$^N$RL4$^{CSA}$-E2-Ub-UVSSA map. **c**. Fourier shell correlation (FSC) plot for the C$^N$RL4$^{CSA}$-E2-Ub-UVSSA map. **d**. 3D FSC plot for the C$^N$RL4$^{CSA}$-E2-Ub-UVSSA map. **e**. Processing tree for the C$^N$RL4$^{CSA}$-E2-Ub complex. The number of particles included in refined maps is indicated below the map, together with the map resolution. Classes used for further processing are color-coded according to complex subunits. **f**. Local resolution estimate for the C$^N$RL4$^{CSA}$-E2-Ub map. **g**. Fourier shell correlation plot for refined C$^N$RL4$^{CSA}$-E2-Ub map. **h**. 3D FSC plot for the C$^N$RL4$^{CSA}$-E2-Ub map.

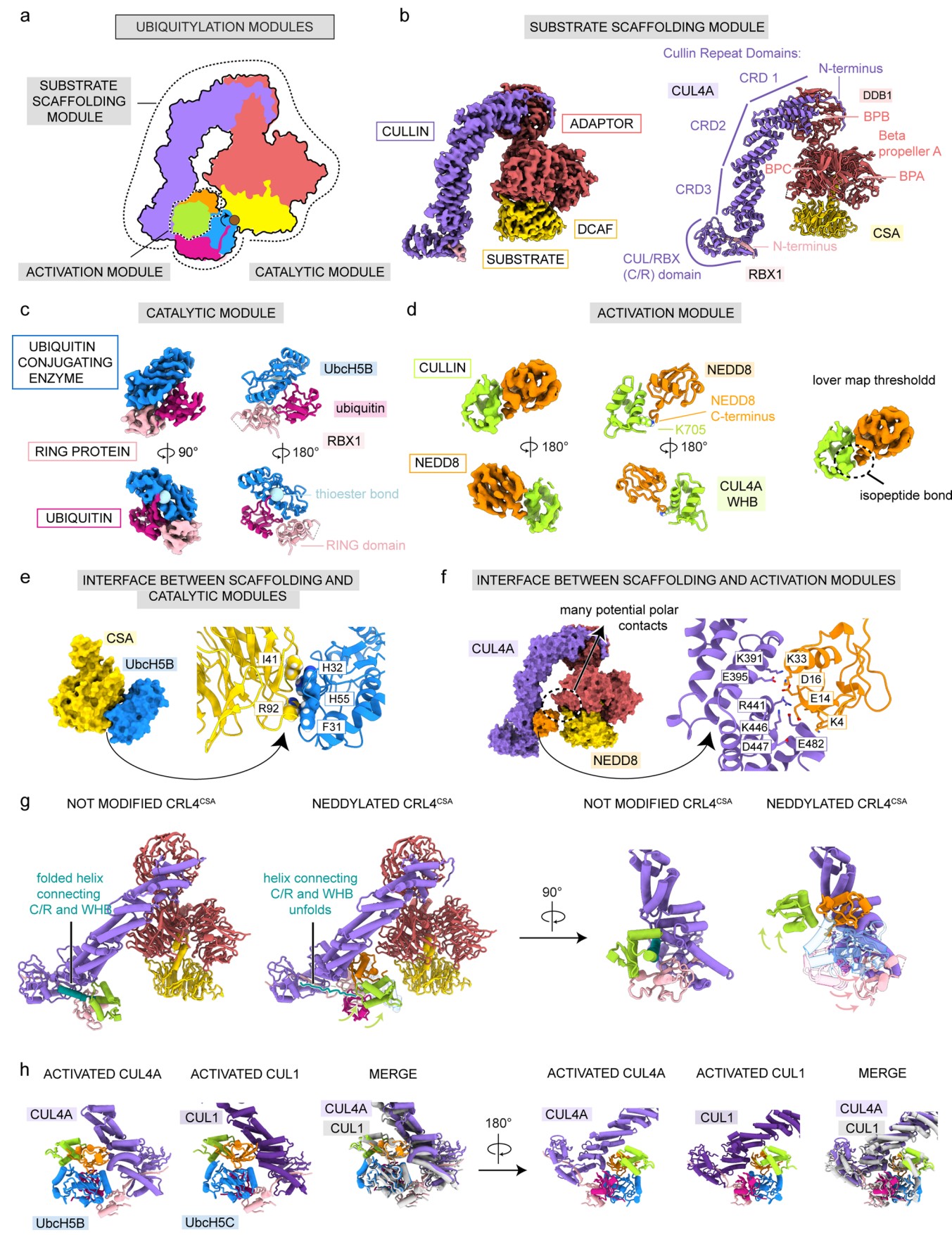

**Extended Data Fig. 7 | See next page for caption.**

**Extended Data Fig. 7 | Conformational changes in CRL4$^{CSA}$ upon neddylation and comparison to cullin 1. a**. Schematic representation of ubiquitylation modules in C$^N$RL4$^{CSA}$-E2-Ub according to the classification in[37]. **b**. Cryo-EM maps and a ribbon model of the substrate scaffolding module. **c**. Cryo-EM maps and a ribbon model of the catalytic module. **d**. Cryo-EM maps and a ribbon model of the activation module. **e**. Zoom-in on the interface between the substrate scaffolding and the catalytic module. Prominent R92 is inserted in the hydrophobic interface between CSA and UBCH5B. **f**. Zoom-in on the interface between the substrate scaffolding and the activating module. The interface is dominated by many complementary charged interactions. **g**. Conformational changes induced by neddylation. The helix connecting C/R and WHB domain unfolds to allow repositioning of NEDD8-bound WHB (left). Neddylation also induces dramatic repositioning of RBX1 RING (right). **h**. Conformational changes upon neddylation of CUL4A are almost identical to the changes in CUL1, suggesting a conserved mechanism of CRL activation by neddylation. The structure solved in this work was compared to the structure of C$^N$RL1$^{ß-TRCP}$ (PDB: 6TTU)[37].

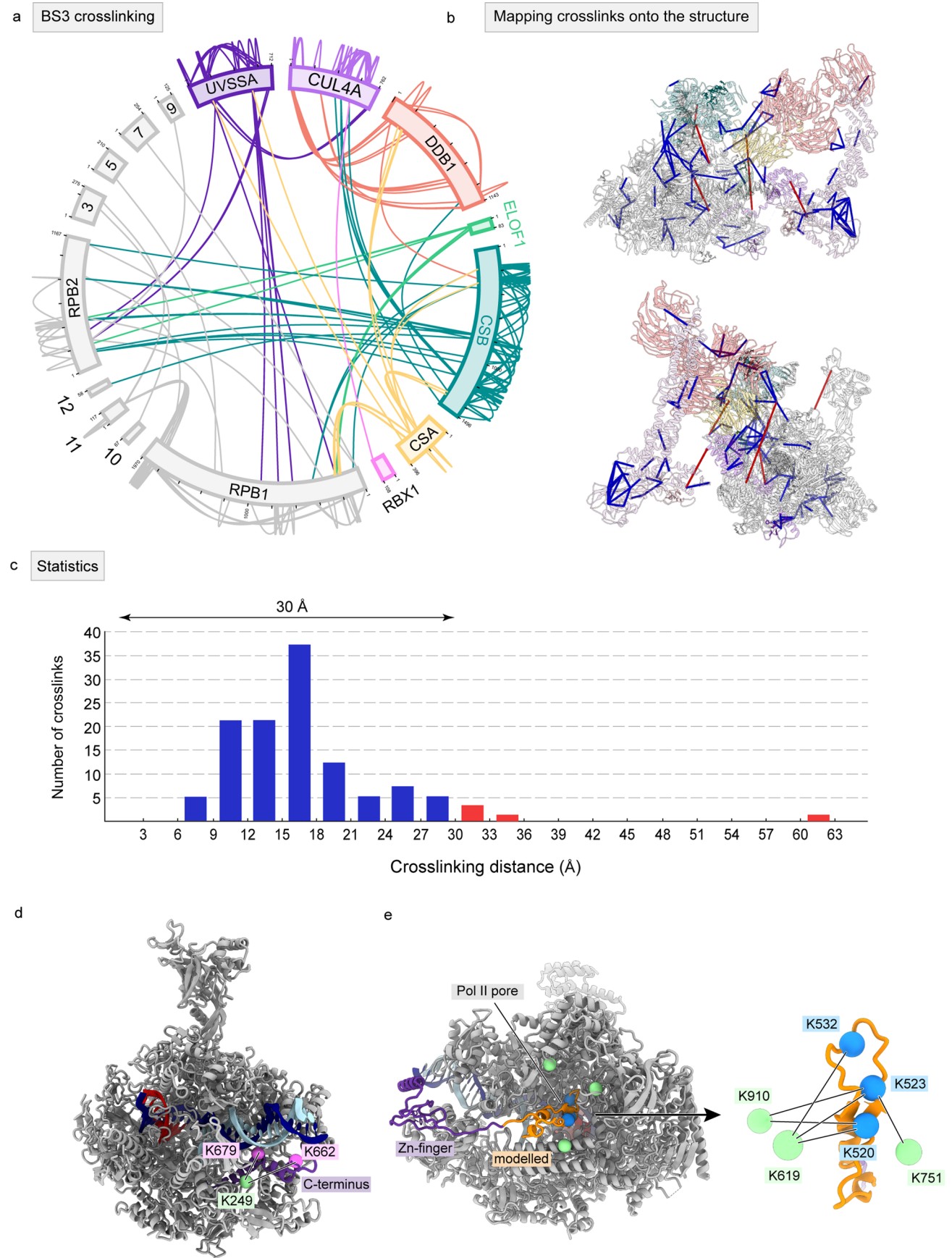

**Extended Data Fig. 8 | See next page for caption.**

**Extended Data Fig. 8 | Crosslinking mass-spectrometry analysis of the Pol II-TCR-ELOF1 complex. a**. Crosslinking network between the subunits of the Pol II-TCR-ELOF1 complex. Crosslinks with a score above 3 that were detected at least twice are shown. **b**. Crosslinks from **a** mapped onto the Pol II-TCR-ELOF1 structure. Crosslinks within the permitted distance of 30 Å are shown in blue and crosslinks violating the distance are shown in red. Over 90% of crosslinks fall within the permitted distance. Crosslinks outside the permitted range likely emerge due to complex flexibility or technical errors. **c**. Histogram shows the number of crosslinks detected at a particular distance. **d**. Crosslinks used to identify the C-terminus of UVSSA. **e**. Crosslinks used to position the UVSSA hairpin in the Pol II pore.

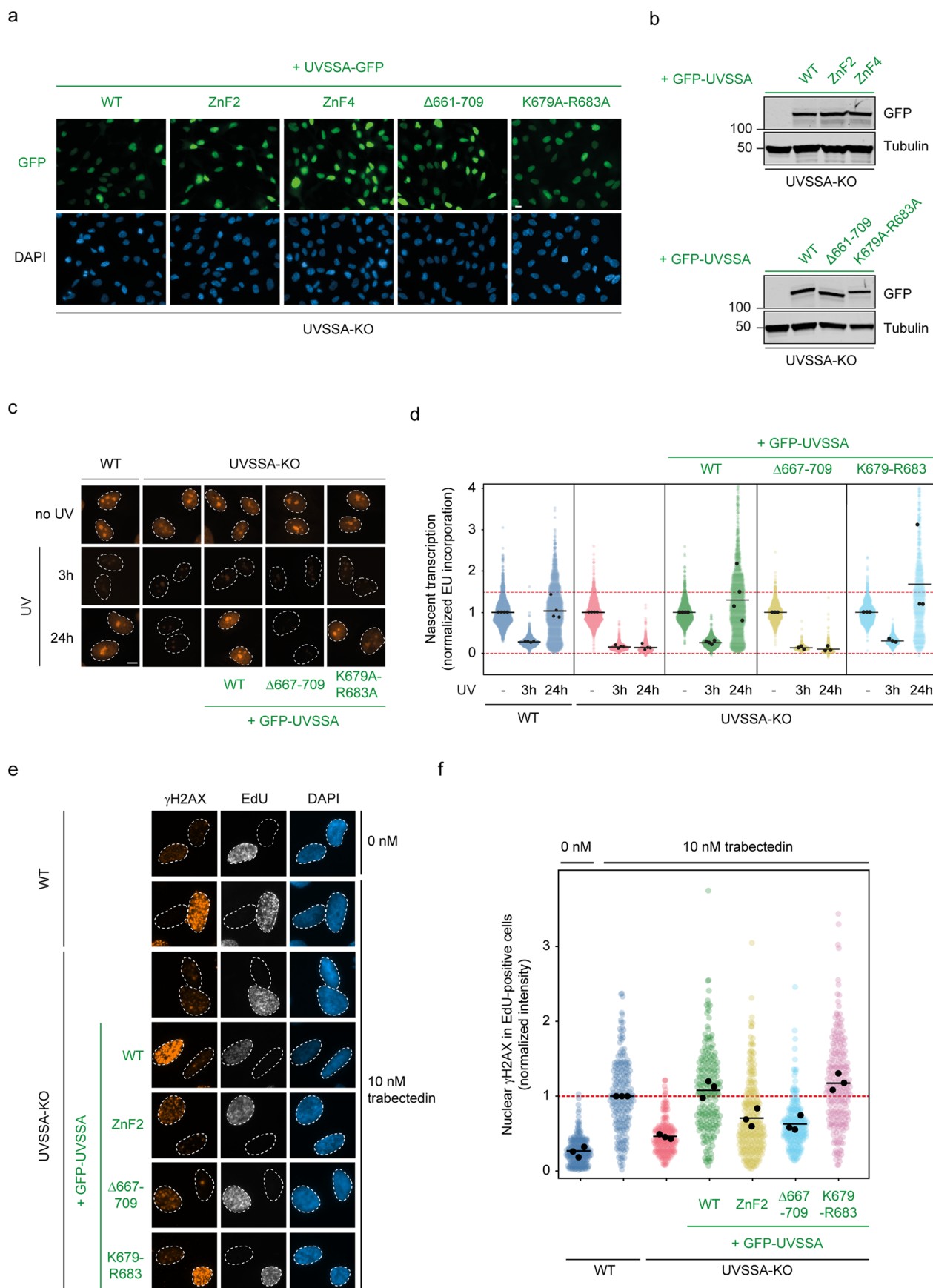

**Extended Data Fig. 9 | See next page for caption.**

**Extended Data Fig. 9 | Expression and TCR activity of UVSSA mutants.**
**a**. Expression of the indicated GFP-UVSSA proteins in RPE1-iCas9 UVSSA-KO cells. The data shown represent at least three independent experiments. Scale bar, 15 μm. **b**. Western blot analysis of GFP-UVSSA proteins (or empty control) in RPE1 UVSSA-KO cells. The data shown represent at least three independent experiments. **c, d**. Measuring transcription recovery (RRS) by 5-ethynyl-uridine labelling in the indicated RPE1-iCas9 cells following UV irradiation (3 h or 24 h; 12 J/m$^2$). Representative images (**c**) and quantification (**d**) of 5-EU levels normalized to mock treatment for each cell line. The experiment was performed three times. Each coloured circle represents 1 cell. Each black circle represents the mean of

two technical replicates, with >80 cells collected per technical replicate. The black lines represent the mean of all three independent experiments. **e, f**. Measuring TCR-dependent γH2AX induction following trabectedin exposure in replicating cells labelled with 5-ethynyl-deoxyuridine (EdU) in the indicated RPE1-iCas9 cells. Representative images (**e**) and quantification (**f**) of γH2AX levels normalized to trabectedin-treated WT cells within each experiment. The experiment was performed three times. Each black circle represents the mean of two technical replicates, with >70 cells collected per technical replicate. The black lines represent the mean of all three independent experiments. Scale bar, 10 μm.

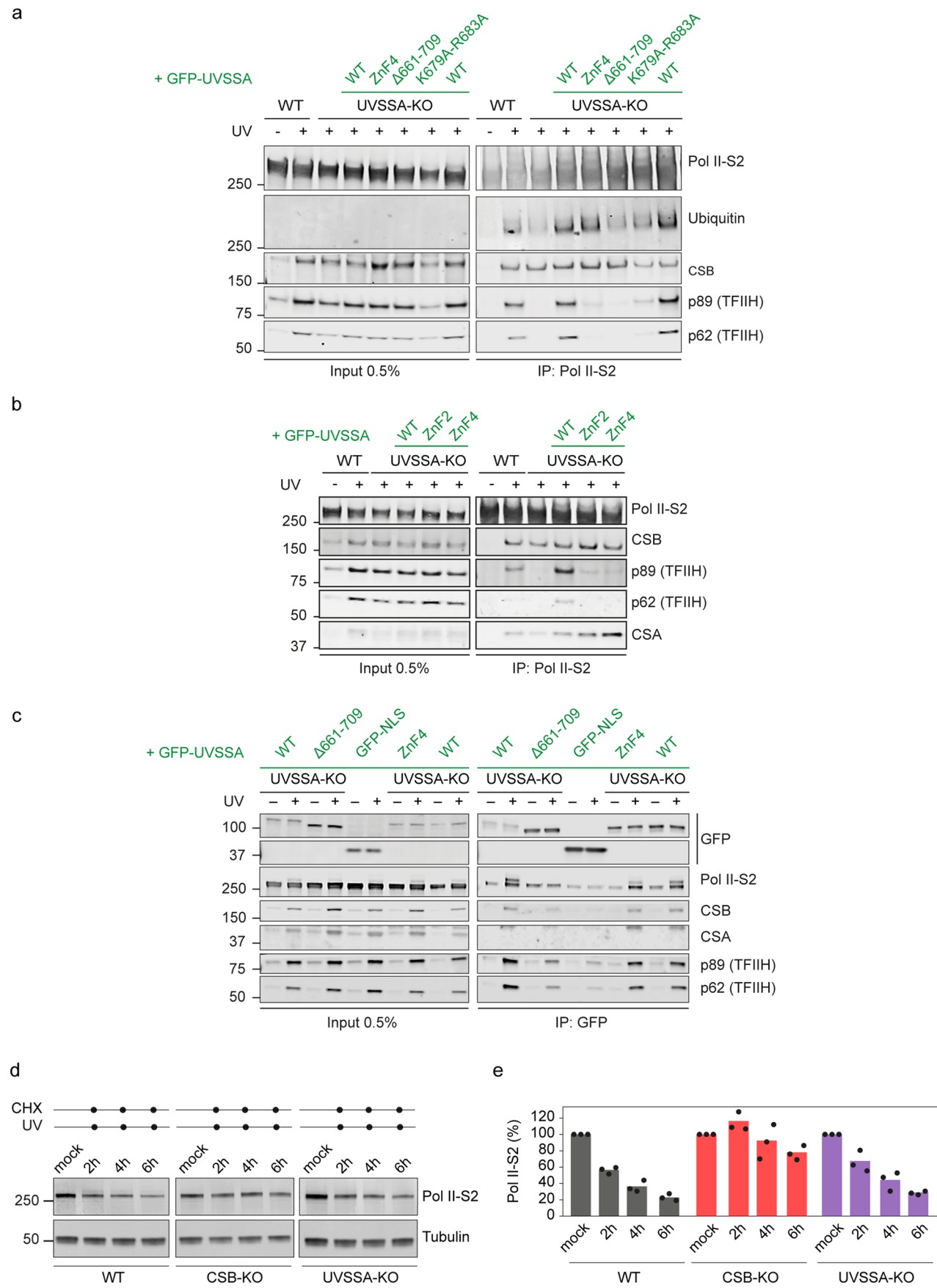

**Extended Data Fig. 10 | See next page for caption.**

**Extended Data Fig. 10 | Immunoprecipitation on Pol II or UVSSA and Pol II degradation in TCR-KO cells after UV. a**. Endogenous Pol II-S2 immunoprecipitation (IP) following UV irradiation (9 J/m$^2$, 1 h recovery) to detect TCR complex assembly in RPE1-iCas9 WT or UVSSA-KO cells complemented with GFP-tagged UVSSA proteins. The data shown represent at least three independent experiments. **b**. Endogenous Pol II-S2 immunoprecipitation (IP) following UV irradiation (9 J/m$^2$, 1 h recovery) to detect TCR complex assembly in RPE1-iCas9 WT or UVSSA-KO cells complemented with GFP-tagged UVSSA proteins. The data shown represent at least three independent experiments.

**c**. GFP-UVSSA or GFP-NLS immunoprecipitation (IP) following UV irradiation (9 J/m$^2$, 1 h recovery) to detect TCR complex assembly in RPE1-iCas9 WT or UVSSA-KO cells complemented with GFP-tagged UVSSA proteins. The data shown represent at least three independent experiments. **d**. Western blot analysis of the indicated RPE1-iCas9 cells after UV irradiation (30 J/m$^2$, 2, 4, 6 h recovery). Cells were treated with 100 µM cycloheximide (CHX) for 1h before UV irradiation. **e**. Quantification of Pol II levels from three independent experiments. Bars represent the average of three independent experiments. Each black circle represents the mean of an independent experiment.

# nature research

# Reporting Summary

Nature Research wishes to improve the reproducibility of the work that we publish. This form provides structure for consistency and transparency in reporting. For further information on Nature Research policies, see our Editorial Policies and the Editorial Policy Checklist.

## Statistics

For all statistical analyses, confirm that the following items are present in the figure legend, table legend, main text, or Methods section.

| n/a | Confirmed | |
|---|---|---|
| ☐ | ☒ | The exact sample size (*n*) for each experimental group/condition, given as a discrete number and unit of measurement |
| ☐ | ☒ | A statement on whether measurements were taken from distinct samples or whether the same sample was measured repeatedly |
| ☒ | ☐ | The statistical test(s) used AND whether they are one- or two-sided<br>*Only common tests should be described solely by name; describe more complex techniques in the Methods section.* |
| ☒ | ☐ | A description of all covariates tested |
| ☒ | ☐ | A description of any assumptions or corrections, such as tests of normality and adjustment for multiple comparisons |
| ☐ | ☒ | A full description of the statistical parameters including central tendency (e.g. means) or other basic estimates (e.g. regression coefficient) AND variation (e.g. standard deviation) or associated estimates of uncertainty (e.g. confidence intervals) |
| ☒ | ☐ | For null hypothesis testing, the test statistic (e.g. *F*, *t*, *r*) with confidence intervals, effect sizes, degrees of freedom and *P* value noted<br>*Give P values as exact values whenever suitable.* |
| ☒ | ☐ | For Bayesian analysis, information on the choice of priors and Markov chain Monte Carlo settings |
| ☒ | ☐ | For hierarchical and complex designs, identification of the appropriate level for tests and full reporting of outcomes |
| ☒ | ☐ | Estimates of effect sizes (e.g. Cohen's *d*, Pearson's *r*), indicating how they were calculated |

*Our web collection on statistics for biologists contains articles on many of the points above.*

## Software and code

Policy information about availability of computer code

| Data collection | Serial EM 3.8 beta 8; pLink (v. 1.23). Microscopy images were acquired using a Zeiss AxioImager M2 or D2 widefield fluorescence microscope and ZEN 2012 software (blue edition, version 1.1.0.0). Western blot images were acquired using a Odyssey CLx with Image studio lite software (v5.2). |
|---|---|
| Data analysis | RELION 3.0, UCSF Chimera 1.13, Coot 0.9, Warp v1.0.7, PHENIX 1.18, cryoSPARC 2.14.2, Prism v8.4.2, Molprobity 4.5.1, XlinkAnalyzer version 1.1. Microscopy images were analyzed in Image J (1.47v-1.48v). Graphs were plotted and analyzed using Graphpad Prism 8 (v8.4.2), Microsoft Excel 365, PlotsOfData webtool, and Adobe Illustrator 2021. |

For manuscripts utilizing custom algorithms or software that are central to the research but not yet described in published literature, software must be made available to editors and reviewers. We strongly encourage code deposition in a community repository (e.g. GitHub). See the Nature Research guidelines for submitting code & software for further information.

## Data

Policy information about availability of data

All manuscripts must include a data availability statement. This statement should provide the following information, where applicable:
- Accession codes, unique identifiers, or web links for publicly available datasets
- A list of figures that have associated raw data
- A description of any restrictions on data availability

The electron density reconstructions and structure coordinates were deposited to the Electron Microscopy Database (EMDB) and to the PDB under the following accession codes: EMD-15825 and PDB 8B3D for the Pol II-TCR-ELOF1 structure, EMD-15829 and PDB 8B3I for the CNRL4CSA-E2-Ub structure and EMD-15827 and PDB 8B3G for the CNRL4CSA-E2-Ub-UVSSA structure. The crosslinking mass spectrometry data have been deposited to the ProteomeXchange Consortium via PRIDE with the dataset identifier PXD042388.

Following structures were used for model building or figure making: Pol II-CSB-CSA-DDB1-UVSSA structure (PDB code 7OO3), NEDD8-CUL1-RBX1 N98R-SKP1-monomeric b-TRCP1dD-IkBa-UB~UBE2D2 (PDB code 6TTU) and RNA polymerase II-TFIIS complex (1PQV).

# Field-specific reporting

Please select the one below that is the best fit for your research. If you are not sure, read the appropriate sections before making your selection.

☒ Life sciences    ☐ Behavioural & social sciences    ☐ Ecological, evolutionary & environmental sciences

For a reference copy of the document with all sections, see nature.com/documents/nr-reporting-summary-flat.pdf

# Life sciences study design

All studies must disclose on these points even when the disclosure is negative.

| | |
|---|---|
| Sample size | No statistical methods were used to predetermine sample size. Sample sizes were chosen for the different experimental approaches based on the technical difficulty and throughput of the individual assays, the chosen sample sizes are consistent with previous publications. All biochemical and cell culture experiments were replicated two or more times. Structural data was collected on five independently prepared samples. |
| Data exclusions | No data were excluded from the analyses. |
| Replication | All attempts at replication were successful, at least two repetitions for biochemical assays were performed. Cryo-EM single particle analysis inherently relies on averaging over a large number of independent observations. For cell culture approaches, the number of replicate experiments are indicated in the figure legends of the manuscript. At least two replicates were performed for each individual approach. Effects of knock-out of proteins of interest were confirmed by rescue experiments in at least three independent experiments. |
| Randomization | There was no allocation of test subjects for any experiments, thus randomization was not applicable to our study |
| Blinding | Data analyses were performed by unbiased software programs/algorithms blinding was therefore not applicable to our study |

# Reporting for specific materials, systems and methods

We require information from authors about some types of materials, experimental systems and methods used in many studies. Here, indicate whether each material, system or method listed is relevant to your study. If you are not sure if a list item applies to your research, read the appropriate section before selecting a response.

## Materials & experimental systems

| n/a | Involved in the study |
|---|---|
| ☐ | ☒ Antibodies |
| ☐ | ☒ Eukaryotic cell lines |
| ☒ | ☐ Palaeontology and archaeology |
| ☒ | ☐ Animals and other organisms |
| ☒ | ☐ Human research participants |
| ☒ | ☐ Clinical data |
| ☒ | ☐ Dual use research of concern |

## Methods

| n/a | Involved in the study |
|---|---|
| ☒ | ☐ ChIP-seq |
| ☒ | ☐ Flow cytometry |
| ☒ | ☐ MRI-based neuroimaging |

# Antibodies

| | |
|---|---|
| Antibodies used | CSA/ERCC8 Mouse Santa Cruz, #sc-376981 (D2) WB: 1:500 aML#025<br>CSA/ERCC8 Rabbit Abcam, #137033 (EPR9237) WB: 1:500 aML#028<br>CSB/ERCC6 Rabbit Santa Cruz, #sc-25370 (H-300) WB: 1:300 aML#003<br>CSB/ERCC6 Rabbit Bethyl Laboratories, #A301-345A WB: 1:600 aML#187<br>GFP Mouse Roche, #11814460001 (7.1 and 13.1) WB: 1:1000 aML#011<br>GFP Rabbit Abcam, #ab290 WB: 1:1000 aML#044<br>Mouse Alexa 555 Goat Thermo fisher Scientific, A-21424 IF: 1:1000 aML#015<br>Mouse Alexa 647 Goat Thermo fisher Scientific, A-21235 IF: 1:1000 aML#017<br>Mouse IgG (H+L) CF770 Goat Biotium, VWR #20077 WB: 1:10000 aML#009<br>p62/GTF2H1 Mouse Santa Cruz, #sc-48431 (G10) WB: 1:500 aML#099<br>p89/XPB/ERCC3 Mouse Millipore, #MABE1123 WB: 1:2000 aML#101<br>phospho-H2A.X Ser139 Mouse Merck, #05-636 (JBW301) IF: 1:1000 aML#161<br>Pol II-S2 Rabbit Abcam, #ab5095 WB: 1:1000 aML#024<br>Rabbit IgG (H+L) CF680 Goat Biotium, VWR #20067 WB: 1:10000 aML#010<br>RBX1 Rabbit Cell Signaling, 11922S WB: 1:6000 aML#155<br>RPB1 (fluorescently labelled 8WG16) Mouse Cramer lab in-house purified WB: 1:1000 |

Ubiquitin (FK2) Mouse ENZO Life Sciences, BML-PW8810-0500 WB: 1:1000 aML#102
Ubiquitin (P4D1) Mouse Cell Signaling, , mAb#3936 WB: 1:1000 aML#192
αTubulin  Mouse Sigma, #T6199 (DM1A) WB: 1:1000 aML#008

| Validation | The following antibodies were validated in knockout cells and Co-IP experiments:

CSA/ERCC8, Mouse, Santa Cruz #sc-376981 (D2), WB: 1:500, aML#025
CSA/ERCC8, Rabbit, Abcam #137033 (EPR9237), WB: 1:500, aML#028
CSB/ERCC6, Rabbit, Santa Cruz #sc-25370 (H-300) WB: 1:300 aML#003
CSB/ERCC6, Rabbit, Bethyl Laboratories #A301-345A, WB: 1:600 ,aML#187

The following antibodies were validated in Co-IP experiments:

GFP, Mouse, Roche #11814460001 (7.1 and 13.1), WB: 1:1000, aML#011
p62/GTF2H1, Mouse, Santa Cruz #sc-48431 (G10), WB: 1:500, aML#099
p89/XPB/ERCC, 3Mouse, Millipore #MABE1123, WB: 1:2000, aML#101
RNAPII-S2, Rabbit, Abcam #ab5095, WB: 1:1000, aML#024

This antibody was validated by western blot of cells expressing GFP tagged proteins:
GFP, Rabbit, Abcam #ab290, WB: 1:1000, aML#044

This antibody was validated in fluorescence microscopy experiments:
phospho-H2A.X Ser139, Mouse, Merck #05-636 (JBW301), IF: 1:1000, aML#161

This antibody is a commonly used loading control:
Tubulin, Mouse, Sigma #T6199 (DM1A), WB: 1:1000, aML#008

This antibody is a commonly used ubiquitin antibody:
Ubiquitin (P4D1), Mouse, Cell Signaling, WB: 1:1000, aML#192 |

# Eukaryotic cell lines

Policy information about cell lines

| Cell line source(s) | Sf9 insect cells (ThermoFisher, 12659017) were cultured in Sf-9000TM III SFM medium (Thermo Fisher Scientific)
Hi5 (Expression systems, 94-002F) were cultured in ESF921 medium (Expression Systems, 96-001-01)
Sf21 cells (Expression systems, 94-003F) were cultured in ESF921 medium (Expression Systems, 96-001-01)
HEK293T (ATCC CRL-3216)
RPE1-iCas9 (van der Weegen, et al. 2021)
RPE1-iCas9 CSB-KO (1-15) (van der Weegen, et al. 2021)
RPE1-iCas9 ELOF1-KO (2-16) (van der Weegen, et al. 2021)
RPE1-iCas9 ELOF1-KO (2-16) + GFP-ELOF1-WT, This study
RPE1-iCas9 ELOF1-KO (2-16) + GFP- ELOF1-N30A-H31A-E32A (Δdock), This study
RPE1-iCas9 ELOF1-KO (2-16) + GFP- ELOF1-E55A-E79A, This study
RPE1-iCas9 ELOF1-KO (2-16) + GFP-ELOF1-N30A-H31A-E32A-E55A-E79A, This study
RPE1-iCas9 UVSSA-KO (3-9) (van der Weegen, et al. 2021)
RPE1-iCas9 UVSSA-KO (3-9) + GFP-UVSSA-WT, This study
RPE1-iCas9 UVSSA-KO (3-9) + GFP-UVSSA-Δ667-699 (ΔC), This study
RPE1-iCas9 UVSSA-KO (3-9) + GFP-UVSSA-K679A-R683A,This study
RPE1-iCas9 UVSSA-KO (3-9) + GFP-UVSSA-C567A-C577A (ΔZnF2), This study
RPE1-iCas9 UVSSA-KO (3-9) + GFP-UVSSA-C567A-C577A-C585A-H588A (ΔZnF4), This study
RPE1-iCas9 CSA-KO (3-8) (van der Weegen, et al. 2021)
RPE1-iCas9 CSA-KO (3-8) + CSA-WT-GFP, This study
RPE1-iCas9 CSA-KO (3-8) + CSA-Y334A-GFP, This study
U2OS (FRT) UVSSA-KO (1-8) + GFP-UVSSAWT-3 (van der Weegen, et al. 2020)
U2OS (FRT): Gift from Daniel Durocher (Toronto, Ontario,) |
| Authentication | None of the cell lines were authenticated. |
| Mycoplasma contamination | Hi5, Sf9, and Sf21 cell lines were not tested for mycoplasma contamination. All RPE1-iCas9, U2OS (FRT) and HEK293T cell lines were regularly tested for mycoplasma contamination and were negative. |
| Commonly misidentified lines (See ICLAC register) | No commonly misidentified cell lines were used. |

