## [Peer Review File · Nature Structural & Molecular Biology]

Peer Review Information

Manuscript Title: Structural basis for RNA pol II ubiquitylation and inactivation in transcription-coupled repair

Corresponding author name(s): Patrick Cramer, Martijn Luijsterburg

Reviewer Comments & Decisions:

Decision Letter, initial version:

Message: 2nd Aug 2023

Dear Dr. Luijsterburg,

Thank you again for submitting your manuscript "Structural basis for RNA pol II ubiquitylation and inactivation in transcription-coupled repair". We now have comments (below) from the 3 reviewers who evaluated your paper. In light of these reports, we remain rather interested in your study and would like to see your response to the comments of the referees, in the form of a revised manuscript.

You will see that all referees appreciate the novelty of the findings, their importance to the TCR field, and the robustness of the structural data and the accompanying cellular data. However, the authors offer several suggestions and raise some concerns that need to be addressed in a revised manuscript.

More specifically, you will see that both reviewers #2 and #3 request various clarifications, expanded introductory comments and discussion of the data. In addition, reviewer #2 points out areas that can/should be technically improved (points 12, 18, 19) and poses certain mechanistic questions, whose answer would further boost the value of this work (points 5,6, 16, 20). Along similar lines, both reviewer #1 and #3 mechanistically ponder how interaction with the TFIIH is regulated; additional experimental data or adequately discussing potential downstream events would be appreciated. Finally, reviewer #3 (point 3) would like you to better contextualise the various findings given the absence of a DNA lesion in the in vitro analyses. As always, we ask you to address the issues raised by the experts in their entirety in a point-by-point response.

We expect to see your revised manuscript within 2-3 months. If you cannot send it within this time, please contact us to discuss an extension; we would still consider your revision, provided that no similar work has been accepted for publication at NSMB or published elsewhere.

Reporting Summary:

Data availability: this journal strongly supports public availability of data. All data used in accepted papers should be available via a public data repository, or alternatively, as Supplementary Information. If data can only be shared on request, please explain why in your Data Availability Statement, and also in the correspondence with your editor. Please note that for some data types, deposition in a public repository is mandatory - more information on our data deposition policies and available repositories can be found below: <https://www.nature.com/nature-research/editorial-policies/reporting-standards#availability-of-data>

[Redacted]

Sincerely,

Dimitris Typas
Associate Editor
Nature Structural & Molecular Biology
ORCID: 0000-0002-8737-1319

Referee expertise:

Referee #1: cryo-EM and transcription-PolII complexes

Referee #2: cryo-EM and ubiquitin ligases and/or TCR-PolII complexes

Referee #3: TCR/NER (structural biology, molecular biology, and biochemistry)

Reviewers' Comments:

Reviewer #1:

Remarks to the Author:

The manuscript entitled "Structural basis for RNA pol II ubiquitylation and inactivation in transcription-coupled repair" by Kokic et al., report biochemical reconstitution and cryo-EM structure of a ELOF1 containing transcription-coupled repair complex. The findings are exciting as the major outcome of this manuscript is the understanding at the molecular level of RNA polymerase II ubiquitylation and inactivation during transcription coupled repair.

The manuscript is well written, concise and the figures are very explanatory. The experiments in cellulo are very informative and logically informed by the cryo-EM structure. The cryo-EM data collection and processing are state-of-the art.

It does not happen frequently to have to review a manuscript so well conceived and rationale in its reporting. I applaud the authors for the efforts made into this study. Of course it will be interesting to see how TFIIF is recruited onto this complex and I presume this will be the next effort by the authors.

Reviewer #2:

Remarks to the Author:

Luijsterburg and colleagues structurally and functionally establish the repair factor ELOF1 as a key adaptor that connects and positions CUL4(CSA) and UVSSA on Pol II. UVSSA further prevents reactivation of the UV-lesion-stalled polymerase by competing with TFIIS in an unexpected manner. The work is of high quality and a landmark in the field transcription-coupled repair (TCR). I particularly enjoyed the integration of both structural and functional data into a coherent molecular understanding of TCR. I highly recommend publication in NSMB and mostly have minor suggestions to improve the paper.

Comments:

1. The authors should provide a more comprehensive introduction and mention what the previous structural work (Kokic, 2021) had already shown regarding the TCR complex. In particular, it should be emphasized that only CSB was contacting Pol II directly in the previously available structures.

2. Can the authors expand a bit more on the roles of ELOF1, what is known in humans and what for the yeast orthologue Elf1? Can ELOF1 be conceived as a standard elongation factor within the canonical EC as suggested in yeast (Ehara et al., Science 2017)? And if not, why not?

3. What is missing in the discussion is the description of the state of the art/current model. Should the canonical EC complex (Pol II, PAF, DSIF, TFIIS), and its displacement by CSB over an 'arrest sequence' (Kokic, 2021) still be considered for the final model? The mechanism provided here must be put into the context of this bigger picture!! In the current form the reader has to go back and forth to the previous study to understand the new model.
4. Can the authors mention more clearly which of the maps described along the manuscript are composite? Also, they need to elaborate more on the data processing leading to the final composite map. For example, it is not clear whether in Figure 5, Cul4 was included in the specimen or if the density from a different structure was then modelled on the complex. More graphical aids would be of help: labels and measurements of distances between Cul4 and the critical Pol II epitopes etc..
5. CSA appears to bind ELOF1 at or near the "substrate" binding site of the propeller. Does the linear ELOF1 epitope engaging CSA (or its surrounding sequence) have known sites for PTMs such as phosphorylation, methylation or acetylation? Could these be regulatory for repair?
6. The finding that CRL4(CSA)-E2~Ub in the absence of UVSSA is "seemingly poised for autoubiquitylation of CSA" is very interesting as a potential proofreading step for the assembly of the full complex? Is this experimentally also observed in in vitro ubiquitination reactions?
7. One thing, arguably a bit peripheral to this study but still biologically relevant is the control of the CUL4 neddylation state in response to damage; the current CSN/CRL models predicts that the ligase in complex with its substrate is not subject to CSN binding and hence remains neddylated and active; along these lines, the neddylated CRL4(CSA) complex alone would surely be a CSN substrate. If one were to structurally superimpose the CUL4-CSN complexes (using CRL4(DDB2)-CSN as a model) onto CRL4(CSA), which additional complex member(s) would clash with CSN in the neddylated states (would that be Pol II and/or UVSSA and/or ELOF1)? In other words, how is this ligase regulated and escape CSN/de-neddylation? And is the conformational change upon neddylation of CUL4(CSA) in the presence of Pol II, UVSSA, CSB, ELOF1 involved in the escape from CSN.
8. The neddylated structures and the ensuing conformational changes in the presence of the E2 are exciting. Yet for the statement "loading with E2~Ub, the beta-propeller B of DDB1 turns 40°" it was not quite clear to me where/what the hinge/rigid bodies are that move. Is it the B-domain of DDB1 or the C-terminus of CRL4?
9. Could this statement be better explained: "This stabilization of the downstream DNA might help explain the previously observed stimulatory effect of UVSSA on transcription in vitro"
10. Is it not still somewhat odd that "UVSSA-deficient cells show normal degradation" along with the model presented? Could the authors comment.
11. As a note of caution: a role of for CUL4 and CSA in K63 ubiquitination would be unexpected and unusual for CRL4s. There has to my knowledge not been very convincing

evidence for this.

12. As for the in vitro assays are concerned, another word of caution: there is quite good evidence that for CUL4 ligases - at least for CRBN - UBE2D3 is the priming E2, while UBE2G1 extends the chain (PMID: 30042095). Using these E2 may clean up some of the in vitro assay contradictions/surprises, with UBCH5 being very promiscuous and sometimes misleading.

13. I struggled a bit with a non-degradation model where Pol II ub. by CSA first leads to TFIID recruitment and then to degradation. I would ask the authors to consider re-writing this paragraph in the discussion, which is also a bit redundant with the results. There are many other processes that would explain ELOF1/UVSSA-dependent TFIID recruitment and be consistent with the data shown, even without invoking a non-degradative role of the ubiquitination signal. This is future work material for sure, but it is a bit too prominently discussed for my liking.

14. A number of zinc-finger TFs are transcriptional repressors, e.g. members of the KRAB/SCAN family of ZFsetc.. Could these have a similar binding mode than the ZFs of ELOF1; could the authors look at these interfaces and examine conservation? What I am getting at, could this be a more common mechanism for transcriptional repression.

15. The green/blue ELFOF1 and CSB colours in Fig.1 are very close, could this be changed?

16. The mutually exclusive interaction of UVSSA and TFIIS with Pol II is exciting. Besides the activity competition assay, would it be possible to demonstrate this via a binding competition assay? The experiment could be similar to that shown for CSB and DSIF in Kocic 2021 for example?

17. What is the molecular evidence demonstrating that Pol II ubiquitylation is required for TFIID recruitment? Is this happening via the rearrangement which incorporates UVSSA into the TCR complex following ubiquitination? How can this be reconciled with the finding that TFIID recruitment to TCR seems to be mainly dependent on the UVSSA TIR and flanking Zinc finger?

18. For the IP assays, it would be nice to include the gels for the transfected GFP-ELOF1 or GFP-UVSSA baits.

19. For the in vitro ubiquitination assays additional controls such as UVSSA devoid of its Zinc finger would be very helpful.

20. To corroborate the final model, it would be helpful to test the complex in a transcription assay as in Kocic et al 2021, by adding ELOF1 to see the potential additive effect on Pol II passage over an arrest sequence. It would be even better to see TFIIS displacement by UVSSA incorporated into TCR.

Typos:

21. Line 35: complementary instead of 'complimentary'

22. Line 62: the 'resolution' term seems off in that context. Maybe: the inactive state of Pol II blocked by various obstacles... is typically resolved ...

23. Line 399: Figure 7F should be 7H

Reviewer #3:

Remarks to the Author:

Kokic et al report a series of structural and mutagenic studies of ubiquitylation of RNA polymerase II (Pol II) in the early steps of transcription-couple repair (TCR). The authors have identified functionally important interfaces amongst ELOF1 of the transcription machinery, CSA of CRL4 ubiquitin ligase and DNA repair-specific factor UVSSA in the initial assembly of TCR. They provide detailed structures of the recruitment of CRL4CSA ubiquitin ligase and UVSSA via ELOF1 and conformational rearrangement and activation of CRL4CSA ubiquitin ligase by neddylation of the cullin subunit. Although ubiquitylation of Pol II has not been observed by cryoEM or in vitro studies and inactivation of Pol II should not take place in the absence of a DNA lesion as in the studies reported here, the new findings provide critical missing information toward our understanding of TCR and this manuscript merits publication in NSMB. This said, some explanations are needed to clarify following points.

Major concerns:

1. In the assembly of Pol-TCR-ELOF1 complex, CSB, which is the first co-factor of TCR arriving at a stalled Pol II and requires the presence of DNA lesion before bringing in CRL4CSA ubiquitin ligase and UVSSA, appears not to interact with the upstream DNA. The absence of CSB-DNA interactions may be the reason for many observations reported here, e.g. the mobility of TCR components. Is the upstream DNA too short? Could the authors please include a diagram of the DNA scaffold in Fig. 1 and explain the absence of CSB-DNA interaction?
2. In the absence of a Pol II blocking DNA lesion, is Pol II ubiquitylated at K1268? The in vitro ubiquitylation assay with increasing concentrations of UVSSA (Fig.4e) does not specify whether K1268 is ubiquitylated. Could the absence of a structure of Pol II being ubiquitylated at K1268 be caused by the absence of K1268 ubiquitylation per se or by structural heterogeneity due to the mobile reaction partners?
3. Inactivation of Pol II and DNA repair have to take place in the presence of a DNA lesion and stalled transcription. However, all in vitro analyses reported in this manuscript were carried out in the absence of a transcription-blocking DNA lesion. Since without a DNA lesion inactivation of Pol should not take place as it would be suicidal in normal cell growth, interpretation and discussion in this manuscript needs to take the absence of DNA lesion into account. In the same vein, the title of manuscript "Structural basis for RNA Pol II ubiquitylation and inactivation in transcription-coupled repair" does not accurately summarize the results reported.
4. As TFIIH interacts with Pol II without TCR co-factors in the pre-initiation complexes (PIC), how do TFIIH-Pol II interactions in TCR differ from PIC? In the current Pol-TCR-ELOF1 complex, can TFIIH bind to downstream DNA and Pol II? Functionally, in normal transcription it is TFIIH that departs, but in TCR it is Pol II that departs. What governs the coming and the going?

Other suggestions:

5. In line 87, "ELOF1 is the substrate for stable recruitment of TCR machinery", "substrate" appears an inappropriate description, and the authors may mean "key factor".
6. Please clearly list structures determined in this manuscript and which ones are used in comparison of with and without ELOF1. Physiologically, when is ELOF1 absent in the Pol II

transcription machinery?

7. Would the interactions between UVSSA-Pol II that are stabilized by ELOF1 prevent Pol II from dissociation for TCR to take place?

8. For TCR to take place, Pol II is already stalled by a DNA lesion. Why does Pol II need to be further inactivated by UVSSA? Wouldn't inactivation of Pol II by UVSSA in the absence any DNA lesion be toxic to a cell?

9. In line 349-351, it is stated that "loss of K1268 ubiquitylation site leads to strongly reduced TFIIH binding to lesion stalled Pol II". Please add a reference or experimental data here. Does this mean that without Ub-K1268, TFIIH is not recruited or doesn't bind stably to Pol II?

10. In Fig. 3b, is the 5A mutant ELOF1 (N30Q-H31A-E32A-E55A-E79A) less defective than 3A mutant (N30Q-H31A-E32A or dock)? Why?

11. In Fig. 3d-e, the 3A mutant ELOF1 appears to be more defective than ELOF1 null. Why?

12. In Fig. 3f, labels indicating which form of ELOF1 is added to ELOF1-KO cells are incomplete (3 experiments with 2 labels). Descriptions of the two different panels are also missing.

13. Fig. 4a, interaction of Y334 of CSA with the hydrophobic pocket of UVSSA is unclear as Y334 is behind UVSSA. Which residues in UVSSA form the hydrophobic pocket? Is Y334A mutant CSA defective in TCR?

14. The section and figures describing neddylation of CRL4CSA and ubiquitylation of Pol II may be moved to the end of Results after reporting how UVSSA interacts with ELOF1, Pol II and DNA.

Author Rebuttal to Initial comments

Point-to-point response NSMB-A47916

Reviewer #1:

The manuscript entitled "Structural basis for RNA pol II ubiquitylation and inactivation in transcription-coupled repair" by Kocik et al., report biochemical reconstitution and cryo-EM structure of a `ELOF1 containing transcription-couple repair complex. The finding are exciting as the major outcome of this manuscript is the understanding at the molecular level of RNA polymerase II ubiquitylation and inactivation during transcription coupled repair. The manuscript is well written, concise and the figures are very explanatory. The experiments in cellulo are very informative and logically informed by the cryo-EM structure. The cryo-EM data collection and processing are state-of-the art. It does not happen frequently to have to review a manuscript so well-conceived and rationale in its reporting. I applaud the authors for the efforts made into this study. Of course it will be interesting to see how TFIIH is recruited onto this complex and I presume this will be the next effort by the authors.

We thank the reviewer for his/her assessment of our study and support. Our next effort will indeed be directed towards understanding how TFIIH is recruited to the Pol II-TCR complex.

Reviewer #2:

Luijsterburg and colleagues structurally and functionally establish the repair factor ELOF1 as a key adaptor that connects and positions CUL4(CSA) and UVSSA on Pol II. UVSSA further prevents reactivation of the UV-lesion-stalled polymerase by competing with TFIIS in an unexpected manner. The work is of high quality and a landmark in the field transcription-coupled repair (TCR). I particularly enjoyed the integration of both structural and functional data into a coherent molecular understanding of TCR. I highly recommend publication in NSMB and mostly have minor suggestions to improve the paper.

We thank the reviewer for his/her assessment of our study and support.

Comments:

1. The authors should provide a more comprehensive introduction and mention what the previous structural work (Kokic, 2021) had already shown regarding the TCR complex. In particular, it should be emphasized that only CSB was contacting Pol II directly in the previously available structures.

We agree with the reviewer and we included a sentence stating that the TCR complex is bound to Pol II via a single interface between CSB and Pol II. Other details, such as how CSA binds CSB and how UVSSA binds CSA are also included in the introduction.

2. Can the authors expand a bit more on the roles of ELOF1, what is known in humans and what for the yeast orthologue Elf1? Can ELOF1 be conceived as a standard elongation factor within the canonical EC as suggested in yeast (Ehara et al., Science 2017)? And if not, why not?

Only two studies have examined ELOF1's role in transcription elongation in human cells (Geijer et al., 2021; van der Weegen et al., 2021). Both studies found that knockout of ELOF1 reduces the elongation rate of Pol II with ~20% (from 2.0 kb/min to 1.6 kb/min). In that sense ELOF1 is a bona fide transcription elongation factor. In our structure, human ELOF1 interacts with Pol II and completes the Pol II DNA entry tunnel similar to its yeast counterpart. However, while human ELOF1 is 83 amino acids, yeast Elf1 orthologues are longer (145 amino acids) due to an extended C-terminal acidic tail. This extended tail could cover and hold exposed H2A-H2B, similar to the Spt16 subunit of FACT (Ehara et al., 2019).

3. What is missing in the discussion is the description of the state of the art/current model. Should the canonical EC complex (Pol II, PAF, DSIF, TFIIS), and its displacement by CSB over an 'arrest sequence' (Kokic, 2021) still be considered for the final model? The mechanism provided here must be put into the context of this bigger picture!! In the current form the reader has to go back and forth to the previous study to understand the new model.

We thank the reviewer for this helpful comment. We now expanded the molecular model for TCR in the discussion to include previous insights into the mechanism of TCR.

4. Can the authors mention more clearly which of the maps described along the manuscript are composite? Also, they need to elaborate more on the data processing leading to the final composite map. For example, it is not clear whether in Figure 5, Cul4 was included in the specimen or if the density from a different structure was then modelled on the complex. More graphical aids would be of help: labels and measurements of distances between Cul4 and the critical Pol II epitopes etc..

We extended the legend of Figure 5 to explain how the models were made, which should nicely complement the explanation already present in the main text.

5. CSA appears to bind ELOF1 at or near the "substrate" binding site of the propeller. Does the linear ELOF1 epitope engaging CSA (or its surrounding sequence) have known sites for PTMs such as phosphorylation, methylation or acetylation? Could these be regulatory for repair?

ELOF1 may be ubiquitylated at K38 (NHEKSCDVKMDRARNT) according to a number of di-Gly proteomic screens. This site is close to the N30-H31-E32 region that interacts with CSA and may regulate repair. However, we envision that the interaction between these proteins only takes place when CSB initiates TCR and recruits CSA to ELOF1-bound Pol II. It would be interesting to map and characterize post-translational modification in ELOF1 in future studies.

6. The finding that CRL4(CSA)-E2~Ub in the absence of UVSSA is “seemingly poised for autoubiquitylation of CSA” is very interesting as a potential proofreading step for the assembly of the full complex? Is this experimentally also observed in *in vitro* ubiquitination reactions?

It is well documented in the literature that this E3 ligase undergoes an efficient auto-ubiquitylation of the CSA subunit *in vitro*, although the sites were not mapped (Fischer et al., 2011), and it is very likely that we trapped this process using the E2~Ub conjugate. We were also intrigued by a very stable conformation of the E3 ligase-E2~Ub in absence of other factors, especially since the C-terminus of activated ubiquitin seems to be poised for CSA auto-ubiquitylation and is brought in close proximity to the K335 residue in CSA. We previously detected ubiquitylation of CSA^{K335} during an *in vitro* ubiquitylation assay with CRL4^{CSA} (Kokic et al., 2021). We recently reported the rapid degradation of CSA following UV irradiation of cells lacking either ELOF1, UVSSA, or in cells treated with USP7 inhibitor (van der Weegen et al., 2021). It is thus tempting to speculate that UVSSA-dependent delivery of the USP7 ubiquitin protease (Schwertman et al., 2012) is required to strip CSA modification and allow stable integration of UVSSA into the TCR complex, as well as to prevent CSA degradation. Since ELOF1 facilitates binding of UVSSA to Pol II, this would explain why ELOF1-deficient cells rapidly degrade CSA following UV irradiation and why this effect can also be reproduced by removal of UVSSA or by inhibition of USP7 (van der Weegen et al., 2021). Overall, CSA self-destruction might be a timed mechanism to disassemble TCR complex when UVSSA recruitment and following DNA repair is not possible or needed. We plan to follow this up and characterize the CSA-K335A mutant in more detail *in vivo*.

7. One thing, arguably a bit peripheral to this study but still biologically relevant is the control of the CUL4 neddylation state in response to damage; the current CSN/CRL models predicts that the ligase in complex with its substrate is not subject to CSN binding and hence remains neddylated and active; along these lines, the neddylated CRL4(CSA) complex alone would surely be a CSN substrate. If one were to structurally superimpose the CUL4-CSN complexes (using CRL4(DDB2)-CSN as a model) onto CRL4(CSA), which additional complex member(s) would clash with CSN in the neddylated states (would that be Pol II and/or UVSSA and/or ELOF1)? In other words, how is this ligase regulated and escape CSN/de-neddylation? And is the conformational change upon neddylation of CUL4(CSA) in the presence of Pol II, UVSSA, CSB, ELOF1 involved in the escape from CSN.

As reviewer suggested we compared the CRL4^{CSA} structure bound to CSN or to Pol II-TCR-ELOF complex. A model for the CSN-CRL4 structure was made by fitting the components of the CSN crystal structure (PDB: 4WNS) (Cavadini et al., 2016) and CRL4 components from the structures solved here into the EM maps of CSN-C^NR4A (EMD 3314-3317) (Cavadini et al., 2016). CSA was modelled based on its interaction with DDB1. Comparison shows that CSN indeed clashes considerably with multiple components of the Pol II-TCR complex: CSN3 clashes with CSB, CSN5/6 clash with ELOF1 and the bulk of CSN clashes with Pol II. Due to the severity of clashes, it seems that binding of CRL4^{CSA} to CSN or Pol II-TCR is mutually exclusive independently of the neddylation status of the ligase. It thus seems like the ligase is protected from CSN activity while in the TCR assembly, and dissociation of the ligase, presumably during the course of the repair process, exposes the ligase to the activity of CSN.

8. The neddylated structures and the ensuing conformational changes in the presence of the E2 are exciting. Yet for the statement “loading with E2~Ub, the beta-propeller B of DDB1 turns 40°” it was not quite clear to me where/what the hinge/rigid bodies are that move. Is it the B-domain of DDB1 or the C-terminus of CRL4?

We now added - in relation to the rest of CSA-DDB1 – to avoid this confusion. We hope that the Video 3 we prepared provides a clearer visual aid for the described process.

9. Could this statement be better explained: “This stabilization of the downstream DNA might help explain the previously observed stimulatory effect of UVSSA on transcription in vitro”

We previously observed a slight stimulatory effect of adding only UVSSA on transcription in vitro, which may be explained by the ability of UVSSA to bind Pol II and downstream DNA at the same time and thus increase Pol II processivity. Since this statement is speculative and it seems to cause confusion, we decided to omit it from the manuscript.

10. Is it not still somewhat odd that “UVSSA-deficient cells show normal degradation” along with the model presented? Could the authors comment.

As outlined in the discussion, we suggest that the ubiquitylation we detect on Pol II does not necessarily lead to degradation. Our data shows that CSA, ELOF1, and UVSSA are all required for robust Pol II ubiquitylation. Yet UVSSA-deficient cells show normal Pol II degradation, while cells knockout of CSA, CSB and ELOF1 do not. We need new tools to manipulate different ubiquitin linkages on Pol II to better dissect their roles.

11. As a note of caution: a role of for CUL4 and CSA in K63 ubiquitination would be unexpected and unusual for CRL4s. There has to my knowledge not been very convincing evidence for this.

As we reported in (Nakazawa et al., 2020; see below), we can detect both K48- and K63-linked ubiquitin chains on Pol II after UV irradiation, which are both to a large extent dependent on CSA (and can be inhibited by Neddylation inhibitor, MLN4924).

Please note that auto-ubiquitylation of CRL4^{DDb2} was detected in (Bacher et al., 2021) with both K48- and K63-linked ubiquitin chains in cells in a manner that is stimulated by kinase MEK1. Moreover, CRL4^{AMBRA1} was shown to ubiquitylate Beclin1 with K63-linked ubiquitin chains *in vitro* (Xia et al., 2013). Direct K63-linked ubiquitylation by CRL4^{CSA} would explain our results below, but we cannot exclude that this effect is indirect.

12. As for the *in vitro* assays are concerned, another word of caution: there is quite good evidence that for CUL4 ligases - at least for CRBN - UBE2D3 is the priming E2, while UBE2G1 extends the chain (PMID: 30042095). Using these E2 may clean up some of the *in vitro* assay contradictions/surprises, with UBCH5 being very promiscuous and sometimes misleading.

The results from the Pol II *in vitro* ubiquitylation assay with increasing amounts of UVSSA shown in Fig 4f using UBCH5B/UBE2D2 fit well with our *in vivo* results in UVSSA-deficient cells shown in Fig 4e. Notably, UBE2G1 and UBE2D3 are prominent hits in our genome-wide knockout CRISPR screen on Illudin S (van der Weegen et al., 2021; see below) to identify new regulators of TCR. We intend to follow this up both *in vivo* and *in vitro*, but we believe this is outside the scope of the current work.

13. I struggled a bit with a non-degradation model where Pol II ub. by CSA first leads to TFIIF recruitment and then to degradation. I would ask the authors to consider re-writing this paragraph in the discussion, which is also a bit redundant with the results. There are many other processes that would explain ELOF1/UVSSA-dependent TFIIF recruitment and be consistent with the data shown, even without invoking a non-degradative role of the ubiquitination signal. This is future work material for sure, but it is a bit too prominently discussed for my liking.

We agree that ELOF1/UVSSA could contribute to TFIIF recruitment in other ways too. Please note that we reported previously in (Nakazawa et al., 2020; see left panel below) that RPB1-K1268R knock-in cells that are deficient in Pol II ubiquitylation show strongly reduced TFIIF recruitment. Moreover, we have included new experiments in the revised manuscript showing that neddylation inhibitor, MLN4924, also strongly reduces TFIIF recruitment to DNA damage-stalled Pol II (see right panel below). Given that ELOF1/UVSSA are both required for robust Pol II ubiquitylation as well as for TFIIF recruitment (Nakazawa et al., 2020; van der Weegen et al., 2021), it seems likely that these proteins could also stimulate TFIIF recruitment through Pol II ubiquitylation. This is indeed a focus for future work.

14. A number of zinc-finger TFs are transcriptional repressors, e.g. members of the KRAB/SCAN family of ZFs etc. Could these have a similar binding mode than the ZFs of ELOF1; could the authors look at these interfaces and examine conservation? What I am getting at, could this be a more common mechanism for transcriptional repression.

ELOF1's zinc-finger is responsible for binding CSA and recruiting the E3 ligase, but it does not provide specificity for targeting transcription. As shown in our *in vivo* data, disruption of ELOF1's zinc-finger does not interfere with ELOF1's function in transcription suggesting that ELOF1 binding to Pol II is not mediated by the zinc-finger. It is unlikely that ELOF1's zinc-finger reveals a common mechanism for transcriptional repression but is rather a function an elongation factor evolved to support TCR. In respect to the UVSSA's zinc-finger, we have inspected but did not notice a compelling similarity to suggested members of transcriptional repressors.

15. The green/blue ELOF1 and CSB colours in Fig.1 are very close, could this be changed?

We thank the reviewer for thoughtfully inspecting the figures and taking care of figure clarity. We would however prefer to keep the current coloring scheme as many colors are exhausted when more proteins come into play. Since CSB and ELOF1 do not directly interact, we think that the current coloring is acceptable.

16. The mutually exclusive interaction of UVSSA and TFIIIS with Pol II is exciting. Besides the activity competition assay, would it be possible to demonstrate this via a binding competition assay? The experiment could be similar to that shown for CSB and DSIF in Kocic 2021 for example?

We thank the reviewer for this suggestion and for being so thoughtful of our previous work on the system. Since TFIIIS does not have a very high affinity for elongating Pol II (much higher access is needed to form an EC-TFIIIS complex that is somewhat SEC stable in human system), we decided to not go for such binding studies. Considering that TFIIIS has such a beautiful enzymatic activity, we thought it would be more elegant to utilize it to demonstrate competition with UVSSA. In this way we could also immediately address the effect of UVSSA on TFIIIS anti-backtracking activity. Since the assay results were so clear and the structural overlap between UVSSA and TFIIIS so compelling, we hope that the reviewer agrees such additional experiment would not add great value to the current data.

17. What is the molecular evidence demonstrating that Pol II ubiquitylation is required for TFIIH recruitment? Is this happening via the rearrangement which incorporates UVSSA into the TCR complex following ubiquitination? How can this be reconciled with the finding that TFIIH recruitment to TCR seems to be mainly dependent on the UVSSA TIR and flanking Zinc finger?

The molecular evidence is that RPB1-K1268R knock-in cells that are deficient in Pol II ubiquitylation show strongly reduced TFIIH recruitment (Nakazawa et al., 2020; see point 13). These cells show normal recruitment of CSB and CRL4^{CSA} to stalled Pol II, very similar to ELOF1-deficient cells. Neddylation inhibitor, MLN4924, also strongly reduces both Pol II ubiquitylation (shown in Nakazawa et al., 2020; Tufegdžić Vidaković et al., 2020) as well as TFIIH recruitment to DNA damage-stalled Pol II (see point 13). We believe that TFIIH recruitment to lesion-stalled Pol II depends on at least two signals: (1) the ubiquitylation of Pol II at K1268, and (2) protein-protein interactions with UVSSA's TIR stimulated by the flanking ZnF. Loss of either signal is sufficient to prevent TFIIH recruitment. For example, RPB1-K1268R knock-in cells, or NEDD8 inhibitor-treated cells show normal recruitment of CSB, CSA and UVSSA, but fail to recruit TFIIH (loss of signal 1). Likewise, mutating the ZnF (or TIR) in UVSSA does not affect recruitment of CSB, CSA, UVSSA or Pol II ubiquitylation, but specifically impairs TFIIH recruitment to lesion-stalled Pol II (loss of signal 2; shown in Nakazawa et al., 2020; van der Weegen et al., 2020).

18. For the IP assays, it would be nice to include the gels for the transfected GFP-ELOF1 or GFP-UVSSA baits.

Both Pol II and TFIIH subunits are quite sticky and show non-specific binding to the agarose beads we used under standard IP conditions. We therefore optimized conditions enabling us to capture specific binding of TFIIH to Pol II only after UV irradiation, which requires extensive washing with high-salt (up to 450 mM NaCl). These conditions are unfortunately not compatible with detecting GFP-ELOF1 or GFP-UVSSA interactions with Pol II. We therefore decided to not show these part of the gels. The GFP signal is shown below with signal in the input but no signal in the IP (due to the required stringent washes required for these experiments).

19. For the *in vitro* ubiquitination assays additional controls such as UVSSA devoid of its Zinc finger would be very helpful.

We thank the reviewer for the suggestion. We would like to note that UVSSA lacking its zinc-finger fully supports Pol II ubiquitylation, but fails to recruit TFIID *in vivo* (Fig 7e). Thus, we would not expect a significant impact of this mutation in the ubiquitylation assay

20. To corroborate the final model, it would be helpful to test the complex in a transcription assay as in Kokic et al 2021, by adding ELOF1 to see the potential additive effect on Pol II passage over an arrest sequence. It would be even better to see TFIIS displacement by UVSSA incorporated into TCR.

We thank the reviewer for the suggestions and creative thinking about the additional biochemical assays we could employ. The assay the reviewer is referring to measures the stimulatory effect of CSB on transcription *in vitro*, which is a consequence of CSB binding to stalled Pol II. The first suggested assay would thus potentially probe the increased affinity of the TCR complex for Pol II vs Pol II-ELOF1, which may translate into more efficient passage over the arrest sequence. However, the interaction between TCR complex and Pol II-ELOF1 was directly visualized and we demonstrated that the recruitment of UVSSA and CSA is negatively impacted by the loss of ELOF1 *in vivo*. Furthermore, the occupancy of TCR factors on Pol II is dramatically better in presence of ELOF1, which is an even more direct observation of efficient TCR complex assembly in the presence of ELOF1. We thus hope the reviewer agrees that such a laborious assay that includes running sequencing gels would be redundant. Regarding the second

suggestion, to examine TFIIIS displacement by UVSSA incorporation co-transcriptionally, would be technically very challenging since both TFIIIS and the TCR complex facilitate arrest sequence bypass.

Typos:

21. Line 35: complementary instead of 'complimentary'

Corrected.

22. Line 62: the 'resolution' term seems off in that context. Maybe: the inactive state of Pol II blocked by various obstacles... is typically resolved ...

We reformulated this sentence to: "The pausing of Pol II triggered by various obstacles, including small base damages, is typically overcome by transcription factor IIS (TFIIIS)-dependent RNA cleavage and Pol II reactivation"

23. Line 399: Figure 7F should be 7H

Corrected.

Reviewer #3:

Kokic et al report a series of structural and mutagenic studies of ubiquitylation of RNA polymerase II (Pol II) in the early steps of transcription-couple repair (TCR). The authors have identified functionally important interfaces amongst ELOF1 of the transcription machinery, CSA of CRL4 ubiquitin ligase and DNA repair-specific factor UVSSA in the initial assembly of TCR. They provide detailed structures of the recruitment of CRL4CSA ubiquitin ligase and UVSSA via ELOF1 and conformational rearrangement and activation of CRL4CSA ubiquitin ligase by neddylation of the cullin subunit. Although ubiquitylation of Pol II has not been observed by cryoEM or in vitro studies and inactivation of Pol II should not take place in the absence of a DNA lesion as in the studies reported here, the new findings provide critical missing information toward our understanding of TCR and this manuscript merits publication in NSMB. This said, some explanations are needed to clarify following points.

We thank the reviewer for the assessment of our study and support.

Major concerns:

1. In the assembly of Pol-TCR-ELOF1 complex, CSB, which is the first co-factor of TCR arriving at a stalled Pol II and requires the presence of DNA lesion before bringing in CRL4^{CSA} ubiquitin ligase and UVSSA, appears not to interact with the upstream DNA. The absence of CSB-DNA interactions may be the reason for many observations reported here, e.g. the mobility of TCR components. Is the upstream DNA too short? Could the authors please include a diagram of the DNA scaffold in Fig. 1 and explain the absence of CSB-DNA interaction?

We thank the review for the comment and apologize if this part of our work was not sufficiently clear. The same upstream DNA sequence was used here and in our previous study (Kokic et al., 2021), and binding of CSB to upstream DNA is identical and not affected by the addition of ELOF1. To further clarify this, we added that CSB binding to Pol II and upstream DNA remains largely unchanged in the presence of ELOF1 in the result section.

2. In the absence of a Pol II blocking DNA lesion, is Pol II ubiquitylated at K1268? The in vitro ubiquitylation assay with increasing concentrations of UVSSA (Fig.4e) does not specify whether K1268 is ubiquitylated. Could the absence of a structure of Pol II being ubiquitylated at K1268 be caused by the absence of K1268 ubiquitylation per se or by structural heterogeneity due to the mobile reaction partners?

When we immunoprecipitated Pol II from UV-irradiated cells, we detect clear Pol II ubiquitylation, which is virtually absent if cells are not UV irradiated. Thus, *in vivo* Pol II does not seem to be ubiquitylated at K1268 (at detectable

levels) in the absence of a Pol II-blocking DNA lesions. Please note that we use a Pol II elongation complex containing a large DNA bubble (Kokic et al., 2021), similar to the strategy initially used by the Dong Wang's laboratory to capture the Pol II – Rad26 interaction (the yeast homologue of CSB; see Xu et al., 2017). This *in vitro* substrate enabling the capture of a repair intermediate with many TCR factors bound. We have previously identified 11 ubiquitylation sites on RPB1, including residue K1268 as the highest-scoring site, under our experimental conditions by mass spectrometry (Kokic et al., 2021). Most importantly, our *in vivo* experiments confirm that UVSSA strongly stimulates Pol II ubiquitylation by using UVSSA knockout cells. We also show that re-expression of UVSSA in these cells fully rescues this phenotype (Fig 4c).

3. Inactivation of Pol II and DNA repair have to take place in the presence of a DNA lesion and stalled transcription. However, all *in vitro* analyses reported in this manuscript were carried out in the absence of a transcription-blocking DNA lesion. Since without a DNA lesion inactivation of Pol II should not take place as it would be suicidal in normal cell growth, interpretation and discussion in this manuscript needs to take the absence of DNA lesion into account. In the same vein, the title of manuscript “Structural basis for RNA Pol II ubiquitylation and inactivation in transcription-coupled repair” does not accurately summarize the results reported.

We would like to note that during the initial stages of TCR that we investigate in this work it is the stalled Pol II and not the presence of the lesion that is detected by TCR factors. Indeed, it was shown previously that Pol II ubiquitylation is also triggered by alpha-amanitin, a toxin that blocks Pol II elongation without triggering a DNA lesion (Anindya et al., 2007; Lee et al., 2002). The high-resolution crystal structure of Pol II arrested on a UV-induced DNA lesion shows that arrest on a DNA lesion does not induce any conformational change in Pol II (Brueckner et al., 2007). Furthermore, the DNA lesion is not accessible to repair proteins at this stage of the repair pathway. Thus, the physical presence of a DNA lesion is not necessary for the *in vitro* studies performed here, otherwise we could not be able to form stable Pol II-TCR complexes that only form following DNA damage induction in living cells (van der Weegen et al., 2020). We would also like to note that all major insights from the *in vitro* studies obtained here were confirmed *in vivo* under conditions that specifically activate TCR.

4. As TFIIF interacts with Pol II without TCR co-factors in the pre-initiation complexes (PIC), how do TFIIF-Pol II interactions in TCR differ from PIC? In the current Pol-TCR-ELOF1 complex, can TFIIF bind to downstream DNA and Pol II? Functionally, in normal transcription it is TFIIF that departs, but in TCR it is Pol II that departs. What governs the coming and the going?

This is currently unclear since we lack structural information on where TFIIF binds during TCR. This is an important future goal that we are currently working towards. What we know is that TFIIF recruitment during TCR is fully dependent on the sequential recruitment of CSB, CSA and ultimately UVSSA, which directly interacts with TFIIF through its TIR domain (van der Weegen et al., 2020). This region is unstructured and not resolved in our work. Another clear difference is that the XPD helicase is activated during TCR in a manner that is stimulated by XPA and XPG (Kokic et al., 2019), while XPD is not involved during transcription initiation and not engaged with DNA (He et al., 2016). In fact, dissociation of the kinase module of TFIIF is required and has been suggested to convert TFIIF from a transcription factor into a DNA repair factor (Coin et al., 2008). The combination of TFIIF activity and / or Pol II ubiquitylation likely facilitates Pol II removal during TCR, while TFIIF remains bound to mediate subsequent repair complex assembly. In contrast, TFIIF during transcription initiation is recruited to promoters by TFIIE. Upon promoter escape, both TFIIE and TFIIF dissociate following the recruitment of transcription elongation factors, including DSIF (Compe et al., 2019).

Other suggestions:

5. In line 87, “ELOF1 is the substrate for stable recruitment of TCR machinery”, “substrate” appears an inappropriate description, and the authors may mean “key factor”.

We changed this to key factor as suggested.

6. Please clearly list structures determined in this manuscript and which ones are used in comparison of with and without ELOF1. Physiologically, when is ELOF1 absent in the Pol II transcription machinery?

We expanded the Figure legends to make it clear when structures without ELOF1 from our previous study were used.

We detect a constitutive interaction between Pol II and ELOF1 (van der Weegen et al., 2021). Moreover, the elongation rate of Pol II is ~20% lower (from 2.0 kb/min to 1.6 kb/min) in ELOF1-deficient cells (Geijer et al., 2021; van der Weegen et al., 2021), suggesting that ELOF1 is part of active elongation complexes. In support of this, our previous mass spectrometry on ELOF1 revealed interaction with Pol II and elongation factors SUPT5H, SUPT6H and SUPT16H (see below; van der Weegen et al., 2021).

7. Would the interactions between UVSSA-Pol II that are stabilized by ELOF1 prevent Pol II from dissociation for TCR to take place?

This is an interesting question that requires further investigation. Since TCR can proceed normally in the presence of such interactions, ELOF1 induced Pol II-UVSSA interactions are not inhibitory for TCR. However, the fate of arrested Pol II (backtracking vs dissociation or combination of both) is yet to be established.

8. For TCR to take place, Pol II is already stalled by a DNA lesion. Why does Pol II need to be further inactivated by UVSSA? Wouldn't inactivation of Pol II by UVSSA in the absence any DNA lesion be toxic to a cell?

Pol II stalling at a DNA lesion results in arrest as well as backtracking (Brueckner et al., 2007; Donahue et al., 1994). TFIIIS will reactivate the backtracked Pol II, which will result in stalling at the same DNA lesions again. This could lead to futile cycles of backtracking, reactivation and stalling. Our data suggest that during TCR, UVSSA prevents TFIIIS-mediated reactivation to prevent this. UVSSA will likely not inactivate Pol II in the absence of any DNA lesion, since the recruitment of UVSSA to Pol II is triggered by DNA damage and mediated by CSB and CSA (van der Weegen et al., 2020).

9. In line 349-351, it is stated that "loss of K1268 ubiquitylation site leads to strongly reduced TFIIH binding to lesion stalled Pol II". Please add a reference or experimental data here. Does this mean that without Ub-K1268, TFIIH is not recruited or doesn't bind stably to Pol II?

We now cite our previous work here showing that RPB1-K1268R knock-in cells that are deficient in Pol II ubiquitylation show strongly reduced TFIIF recruitment (Nakazawa et al., 2020; see below). Moreover, new experiments show that Neddylation inhibitor, MLN4924, also strongly reduces TFIIF recruitment to DNA damage-stalled Pol II (see below).

10. In E32A or dock)? Why?

Please see the quantification in Fig 3c. The ELOF1 3A and 5A mutants both rescue the transcription defect in ELOF1-deficient cells to the same extent (3A = 89%, 5A = 91%, while ELOF1-KO is 39%).

11. In Fig. 3d-e, the 3A mutant ELOF1 appears to be more defective than ELOF1 null. Why?

The 3h time-point after UV should be compared to the 24h time-point to compare how well cells recover (at 24h) from the maximal drop in transcription at 24h. The ELOF1-KO cells show no difference between 3h and 24h, indicative of a complete lack of recovery. The 3A mutant cells show a slight increase from 3h to 24h, but are strongly impaired relative to wild-type cells. Thus, the 3A mutant is certainly not more defective than ELOF1 null cells.

12. In Fig. 3f, labels indicating which form of ELOF1 is added to ELOF1-KO cells are incomplete (3 experiments with 2 labels). Descriptions of the two different panels are also missing.

No panel is missing in this figure. The four conditions (all +/- UV) in Fig 3f are: (1) WT, (2) ELOF1-KO, (3) ELOF1-KO + WT, (4) ELOF1-KO + 3A mutant. This is similar to the conditions in panels Fig 2d, e.

13. Fig. 4a, interaction of Y334 of CSA with the hydrophobic pocket of UVSSA is unclear as Y334 is behind UVSSA. Which residues in UVSSA form the hydrophobic pocket? Is Y334A mutant CSA defective in TCR?

We made a new figure to further clarify this issue. CSA in yellow, UVSSA in purple, Y334 is highlighted in orange and hydrophobic residues in UVSSA surrounding Y334 are in cyan (see below). We further highlighted F137 in UVSSA that stacks against Y334. Side (left) and top (right) views are shown.

We have also generated CSA-KO cells, as well as complemented cells expressing either CSA^{WT}-GFP or CSA^{Y334A}-GFP. While CSA^{WT} rescued the TCR-deficient phenotype of CSA-KO cells in both RRS and trabectedin assay, the CSA^{Y334A} mutant is not functional in TCR at all, even though we can detect expression of the mutant protein at similar levels as CSA^{WT} (see below).

14. The section and figures describing neddylation of CRL4CSA and ubiquitylation of Pol II may be moved to the end of Results after reporting how UVSSA interacts with ELOF1, Pol II and DNA.

We appreciate this suggestion, but we prefer to keep the current order. We feel that figure 4 and 5 are a logical order, after which we characterize different features of UVSSA (the C-terminus in Fig 6, the ZnF in Figure 7).

References

- Anindya, R., O. Aygun, and J.Q. Svejstrup. 2007. Damage-induced ubiquitylation of human RNA polymerase II by the ubiquitin ligase Nedd4, but not Cockayne syndrome proteins or BRCA1. *Mol Cell*. 28:386-397.
- Bacher, S., H. Stekman, C.M. Farah, A. Karger, M. Kracht, and M.L. Schmitz. 2021. MEK1-Dependent Activation of the CRL4 Complex Is Important for DNA Damage-Induced Degradation of p21 and DDB2 and Cell Survival. *Mol Cell Biol*. 41:e0008121.
- Brueckner, F., U. Hennecke, T. Carell, and P. Cramer. 2007. CPD damage recognition by transcribing RNA polymerase II. *Science*. 315:859-862.
- Cavadini, S., E.S. Fischer, R.D. Bunker, A. Potenza, G.M. Lingaraju, K.N. Goldie, W.I. Mohamed, M. Faty, G. Petzold, R.E. Beckwith, R.B. Tichkule, U. Hassiepen, W. Abdulrahman, R.S. Pantelic, S. Matsumoto, K. Sugasawa, H. Stahlberg, and N.H. Thoma. 2016. Cullin-RING ubiquitin E3 ligase regulation by the COP9 signalosome. *Nature*. 531:598-603.
- Coin, F., V. Oksenyich, V. Mocquet, S. Groh, C. Blattner, and J.M. Egly. 2008. Nucleotide excision repair driven by the dissociation of CAK from TFIIH. *Mol Cell*. 31:9-20.
- Compe, E., C.M. Genes, C. Braun, F. Coin, and J.M. Egly. 2019. TFIIIE orchestrates the recruitment of the TFIIH kinase module at promoter before release during transcription. *Nat Commun*. 10:2084.
- Donahue, B.A., S. Yin, J.S. Taylor, D. Reines, and P.C. Hanawalt. 1994. Transcript cleavage by RNA polymerase II arrested by a cyclobutane pyrimidine dimer in the DNA template. *Proc Natl Acad Sci U S A*. 91:8502-8506.
- Ehara, H., T. Kujirai, Y. Fujino, M. Shirouzu, H. Kurumizaka, and S.I. Sekine. 2019. Structural insight into nucleosome transcription by RNA polymerase II with elongation factors. *Science*. 363:744-747.
- Fischer, E.S., A. Scrima, K. Bohm, S. Matsumoto, G.M. Lingaraju, M. Faty, T. Yasuda, S. Cavadini, M. Wakasugi, F. Hanaoka, S. Iwai, H. Gut, K. Sugasawa, and N.H. Thoma. 2011. The molecular basis of CRL4DDB2/CSA ubiquitin ligase architecture, targeting, and activation. *Cell*. 147:1024-1039.
- Geijer, M.E., D. Zhou, K. Selvam, B. Steurer, C. Mukherjee, B. Evers, S. Cugusi, M. van Toorn, M. van der Woude, R.C. Janssens, Y.P. Kok, W. Gong, A. Raams, C.S.Y. Lo, J.H.G. Lebbink, B. Geverts, D.A. Plummer, K. Bezstarosti, A.F. Theil, R. Mitter, A.B. Houtsmuller, W. Vermeulen, J.A.A. Demmers, S. Li, M. van Vugt, H. Lans, R. Bernards, J.Q. Svejstrup, A. Ray Chaudhuri, J.J. Wyrick, and J.A. Marteijn. 2021. Elongation factor ELOF1 drives transcription-coupled repair and prevents genome instability. *Nat Cell Biol*. 23:608-619.
- He, Y., C. Yan, J. Fang, C. Inouye, R. Tjian, I. Ivanov, and E. Nogales. 2016. Near-atomic resolution visualization of human transcription promoter opening. *Nature*. 533:359-365.
- Kokic, G., A. Chernev, D. Tegunov, C. Dienemann, H. Urlaub, and P. Cramer. 2019. Structural basis of TFIIH activation for nucleotide excision repair. *Nat Commun*. 10:2885.
- Kokic, G., F.R. Wagner, A. Chernev, H. Urlaub, and P. Cramer. 2021. Structural basis of human transcription-DNA repair coupling. *Nature*. 598:368-372.
- Lee, K.B., D. Wang, S.J. Lippard, and P.A. Sharp. 2002. Transcription-coupled and DNA damage-dependent ubiquitination of RNA polymerase II in vitro. *Proc Natl Acad Sci U S A*. 99:4239-4244.
- Nakazawa, Y., Y. Hara, Y. Oka, O. Komine, D. van den Heuvel, C. Guo, Y. Daigaku, M. Isono, Y. He, M. Shimada, K. Kato, N. Jia, S. Hashimoto, Y. Kotani, Y. Miyoshi, M. Tanaka, A. Sobue, N. Mitsutake, T. Suganami, A. Masuda, K. Ohno, S. Nakada, T. Mashimo, K. Yamanaka, M.S. Luijsterburg, and T. Ogi. 2020. Ubiquitination of DNA Damage-Stalled RNAPII Promotes Transcription-Coupled Repair. *Cell*. 180:1228-1244 e1224.
- Schwertman, P., A. Lagarou, D.H. Dekkers, A. Raams, A.C. van der Hoek, C. Laffebler, J.H. Hoeijmakers, J.A. Demmers, M. Fousteri, W. Vermeulen, and J.A. Marteijn. 2012. UV-sensitive syndrome protein UVSSA recruits USP7 to regulate transcription-coupled repair. *Nat Genet*. 44:598-602.
- Tufegdžić Vidaković, A., R. Mitter, G.P. Kelly, M. Neumann, M. Harreman, M. Rodriguez-Martinez, A. Herlihy, J.C. Weems, S. Boeing, V. Encheva, L. Gaul, L. Milligan, D. Tollervy, R.C. Conaway, J.W. Conaway, A.P. Srijders, A. Stewart, and J.Q. Svejstrup. 2020. Regulation of the RNAPII Pool Is Integral to the DNA Damage Response. *Cell*. 180:1245-1261 e1221.
- van der Weegen, Y., K. de Lint, D. van den Heuvel, Y. Nakazawa, T.E.T. Mevissen, J.J.M. van Schie, M. San Martin Alonso, D.E.C. Boer, R. Gonzalez-Prieto, I.V. Narayanan, N.H.M. Klaassen, A.P. Wondergem, K. Roohollahi, J.C. Dorsman, Y. Hara, A.C.O. Vertegaal, J. de Lange, J.C. Walter, S.M. Noordermeer, M. Ljungman, T. Ogi, R.M.F. Wolthuis, and M.S. Luijsterburg. 2021. ELOF1 is a transcription-coupled DNA repair factor that directs RNA polymerase II ubiquitylation. *Nat Cell Biol*. 23:595-607.
- van der Weegen, Y., H. Golan-Berman, T.E.T. Mevissen, K. Apelt, R. Gonzalez-Prieto, J. Goedhart, E.E. Heilbrun, A.C.O. Vertegaal, D. van den Heuvel, J.C. Walter, S. Adar, and M.S. Luijsterburg. 2020. The cooperative action of CSB, CSA, and UVSSA target TFIIH to DNA damage-stalled RNA polymerase II. *Nat Commun*. 11:2104.
- Xia, P., S. Wang, Y. Du, Z. Zhao, L. Shi, L. Sun, G. Huang, B. Ye, C. Li, Z. Dai, N. Hou, X. Cheng, Q. Sun, L. Li, X. Yang, and Z. Fan. 2013. WASH inhibits autophagy through suppression of Beclin 1 ubiquitination. *EMBO J*. 32:2685-2696.
- Xu, J., I. Lahiri, W. Wang, A. Wier, M.A. Cianfrocco, J. Chong, A.A. Hare, P.B. Dervan, F. DiMaio, A.E. Leschziner, and D. Wang. 2017. Structural basis for the initiation of eukaryotic transcription-coupled DNA repair. *Nature*. 551:653-657.

Decision Letter, first revision:

Message: Our ref: NSMB-A47916A

7th Nov 2023

Dear Dr. Luijsterburg,

Thank you for submitting your revised manuscript "Structural basis for RNA pol II ubiquitylation and inactivation in transcription-coupled repair" (NSMB-A47916A). It has now been seen by the original referees and their comments are below. One reviewer had already expressed that the manuscript was ready for publication at the previous round. The two remaining reviewers find that the paper has further improved in revision, and therefore we'll be happy to accept it in principle in Nature Structural & Molecular Biology, pending minor revisions to comply with our editorial and formatting guidelines.

We are now performing detailed checks on your paper and will send you a checklist detailing our editorial and formatting requirements in about two weeks. Please do not upload the final materials and make any revisions until you receive this additional information from us.

To facilitate our work at this stage, it is important that we have a copy of the main text as a word file. If you could please send along a word version of this file as soon as possible, we would greatly appreciate it; please make sure to copy the NSMB account (cc'ed above).

Sincerely,

Dimitris Typas
Associate Editor
Nature Structural & Molecular Biology
ORCID: 0000-0002-8737-1319

Reviewer #2 (Remarks to the Author):

My concerns have been addressed. Congratulations to the authors for these beautiful structures and their mechanistic dissection.

Reviewer #3 (Remarks to the Author):

The authors have answered all my questions.

Final Decision Letter:**Message** 21st Dec 2023

:

Dear Dr. Luijsterburg,

We are now happy to accept your revised paper "Structural basis for RNA pol II ubiquitylation and inactivation in transcription-coupled repair" for publication as an Article in Nature Structural & Molecular Biology.

Your paper will be published online soon after we receive proof corrections and will appear in print in the next available issue. You can find out your date of online publication by contacting the production team shortly after sending your proof corrections.

You may wish to make your media relations office aware of your accepted publication, in case they consider it appropriate to organize some internal or external publicity. Once your paper has been scheduled you will receive an email confirming the publication details. This is normally 3-4 working days in advance of publication. If you need additional notice of the date and time of publication, please let the production team know when you receive the proof of your article to ensure there is sufficient time to coordinate. Further information on our embargo policies can be found here:

<https://www.nature.com/authors/policies/embargo.html>

Please note that *Nature Structural & Molecular Biology* is a Transformative Journal (TJ). Authors may publish their research with us through the traditional subscription access route or make their paper immediately open access through payment of an article-processing charge (APC). Authors will not be required to make a final decision about access to their article until it has been accepted. [Find out more about Transformative Journals](https://www.springernature.com/gp/open-research/transformative-journals)

Sincerely,

Dimitris Typas
Associate Editor
Nature Structural & Molecular Biology
ORCID: 0000-0002-8737-1319